# HR3/RORα-mediated cholesterol sensing regulates TOR signaling

Mette Lassen [1], Keith Pardee [2,3], Ivan Bradic[4], Lisa H. Pedersen[1], Olga Kubrak [1], Nadja Ahrentløv [1], Sebastian Clancy[2,3], Takashi Koyama [1], Aleksandar Necakov [5], Suya Liu[6], Arnis Kuksis[7,8], Gilles Lajoie[6], Aled Edwards [7,9], Aurelio A. Teleman[10], Martin R. Larsen [4], Henry M. Krause [7,11], Michael J. Texada [1] & Kim Rewitz [1] ✉

Cells and organisms adjust their growth based on the availability of cholesterol, which is essential for cellular functions. However, the mechanisms by which cells sense cholesterol levels and translate these into growth signals are not fully understood. We report that cholesterol rapidly activates the master growth-regulatory TOR pathway in *Drosophila* tissues. We identify the nuclear receptor HR3, an ortholog of mammalian RORα, as an essential factor in cholesterol-induced TOR activation. We demonstrate that HR3 binds cholesterol and promotes TOR-pathway activation through a non-genomic mechanism acting upstream of the Rag GTPases while also restraining longer-term responses through genomic regulation. We also find that RORα is necessary for cholesterol-mediated TOR activation in human cells, suggesting that HR3/RORα-mediated signaling represents a conserved mechanism for cholesterol sensing that couples cholesterol availability to TOR-pathway activity. These findings advance our understanding of how cholesterol influences cell growth, with implications for cholesterol-related diseases and cancer.

Cholesterol is widely recognized as a fundamental component of cellular membranes. Beyond its structural role as a lipid, it functions as a critical signaling molecule, affecting signaling pathways and physiological processes. Cholesterol can interact with proteins through specific cholesterol-binding domains, thereby influencing various cellular processes such as signal transduction[1–4]. Furthermore, cholesterol is the precursor to other signaling molecules, including steroid hormones, bile acids, and oxysterols, that can activate specific receptors and modulate gene expression[5,6]. Dysregulated cholesterol levels have been implicated in many health disorders ranging from cardiovascular diseases to promoting cancerous growth[7,8]. The deep involvement of cholesterol in physiology highlights the importance of cellular mechanisms that sense cholesterol availability and in response modulate cellular growth.

In mammalian cells, which can synthesize sterols de novo, cholesterol homeostasis is governed by sterol-sensing mechanisms within the endoplasmic reticulum (ER). Through regulation of the transcription factor Sterol Response Element Binding Protein (SREBP),

[1]Department of Biology, University of Copenhagen, Copenhagen, Denmark. [2]Department of Pharmaceutical Sciences, Leslie Dan Faculty of Pharmacy, University of Toronto, Toronto ON, Canada. [3]Department of Mechanical and Industrial Engineering, University of Toronto, Toronto ON, Canada. [4]Department of Biochemistry and Molecular Biology, University of Southern Denmark, Odense, Denmark. [5]Department of Biological Sciences, Brock University, Saint Catharines, Canada. [6]Department of Biochemistry, University of Western Ontario, London Ontario, Canada. [7]Banting and Best Department of Medical Research, Department of Molecular Genetics, University of Toronto, Toronto ON, Canada. [8]Department of Biochemistry, University of Toronto, Toronto ON, Canada. [9]Department of Medical Biophysics, University of Toronto, Toronto, Canada. [10]German Cancer Research Center (DKFZ), Division B140, Heidelberg, Germany. [11]The Donnelly Centre for Cellular & Biomolecular Research, University of Toronto, Toronto ON, Canada. ✉e-mail: Kim.Rewitz@bio.ku.dk

cholesterol biosynthesis and cellular uptake are modulated to maintain lipid homeostasis[9]. The demand for cholesterol increases during rapid cell proliferation – for example, during development and in cancer cells. To meet this demand, upregulation of pathways such as phosphatidylinositol 3-kinase (PI3K)/Protein Kinase B (PKB/AKT) enhances the uptake of exogenous cholesterol[10]. Although cholesterol is necessary for cell-membrane synthesis, increased cholesterol availability also directly drives cell growth and fuels rapid proliferation in cancers. Unlike the well-characterized SREBP pathway, the cellular mechanisms by which cholesterol abundance or deficiency is linked to growth-regulatory pathways are poorly defined.

We recently demonstrated that cholesterol stimulates systemic growth during development through the activation of insulin signaling in *Drosophila*[11]. While our genetic evidence implicated the target of rapamycin (TOR) pathway in this process, the cholesterol-induced TOR responses and the molecular mechanisms linking cholesterol to TOR activation – particularly identifying the components that directly interact with cholesterol to register its availability – remained elusive. Thus, the mechanisms by which cellular cholesterol levels are conveyed to TOR for translation into growth signals are poorly understood. The TOR pathway integrates environmental cues including nutrients, oxygen availability, growth stimuli, and cellular stressors into a single signal that regulates cellular growth[12,13]. Activated TOR promotes cellular growth and protein synthesis by promoting ribosomal biogenesis and enhancing translation while simultaneously inhibiting autophagy and regulating metabolism. In mammalian cell culture, cholesterol has emerged as a key factor that activates the TOR pathway[3,14]. Extracellular cholesterol is taken up through endocytosis and trafficked to the late endosome/lysosome compartment. When cholesterol accumulates within these organelles, the TOR complex is recruited by a large protein assembly to the lysosomal membrane, where it interacts with the small GTPase Rheb, which in turn activates TOR kinase activity. In conditions of nutrient scarcity, including low cholesterol, TOR remains localized to the cytosol. In this compartment, TOR has little access to Rheb and therefore cannot be activated.

A large network of proteins orchestrates the nutrient-responsive localization of TOR to the lysosome (Fig. 1a). The lysosomal recruitment of TOR is ultimately governed by small GTPases of the Rag family[12,15]. These proteins associate with the LAMTOR1 subunit of the Ragulator complex, which anchors them to the lysosomal membrane. In response to nutrient abundance, a diverse array of mechanisms configures the GTP/GDP loading of Rag heterodimers in such way that they bind to RAPTOR, a component of the TOR complex, which maintains TOR in association with the lysosomal surface, where it can be activated by Rheb. Rag heterodimers are active and can interact with RAPTOR when RagA/B is guanosine triphosphate (GTP)-bound and RagC/D is guanosine diphosphate (GDP)-bound[12]. This balance is maintained by GTPase-activating protein (GAP) complexes such as GATOR1 and guanine nucleotide exchange factor (GEF) complexes including Ragulator. In nutrient-depleted states, GATOR1 induces GTP hydrolysis on RagA, leading to loss of TOR activation. This mechanism is influenced by amino acids, which affect the GAP activity of GATOR1. Low amino-acid levels also negatively regulate Ragulator, preventing it from activating the Rag GTPases through its GEF functionality.

In contrast to the level of detail with which amino acid-mediated TOR activation is understood, the cholesterol-sensing mechanisms that are capable of directly binding to or detecting intracellular cholesterol and interacting with the TOR pathway are poorly defined. Multiple intracellular cholesterol pools are present, including those in lysosomes, deposited across the plasma membrane and ER, as well as at membrane contact sites[16]. Given the variability in cholesterol concentration across these distinct cellular compartments, it is likely that several distinct cholesterol-sensing mechanisms exist, each specifically evolved to detect different cholesterol pools and independently modulate the TOR pathway. Recent reports suggest that a mechanism by which lysosomal cholesterol activates the TOR complex in mammalian cells involves LYCHOS, a G protein–coupled receptor (GPCR)-like protein integral to the lysosomal membrane that modulates the Rag nucleotide-binding state in a cholesterol-dependent manner via interaction with GATOR1[14]. Additionally, the mammalian lysosomal transmembrane protein SLC38A9 is important for the activation of TOR in response to both cholesterol and amino acids[3]. In *Drosophila*, dietary cholesterol activates TOR, with increasing levels driving rapid growth and development[11]. Intracellular cholesterol accumulation in lysosomal compartments caused by deficiency in the cholesterol transporter Niemann-Pick Type C1 (NPC1) drives strong activation of the TOR pathway in both *Drosophila* and mammals[3,11]. This common feature suggests that the mechanisms coupling cholesterol levels to the activation of the TOR pathway are evolutionarily ancient and conserved. *Drosophila* possesses an ortholog of LYCHOS, named Anchor, but an SLC38A9-like protein does not appear to exist in flies, implying that other ancient and possibly conserved pathways exist that detect cholesterol and link its availability to the activation of TOR signaling.

Nuclear receptors bind lipophilic molecules and exert their regulatory influence through both direct effects on transcription (genomic effects) and more immediate signaling mechanisms that do not involve changes in gene transcription (non-genomic effects), orchestrating a diverse range of cellular processes[17,18]. DHR96, a *Drosophila* nuclear receptor involved in cholesterol homeostasis, binds cholesterol and regulates the expression of genes involved in cholesterol uptake, metabolism, and transport[1,4]. In mammals, the nuclear receptor Retinoic acid receptor (RAR)-related Orphan Receptor alpha (RORα) influences cholesterol homeostasis by modulating the transcription of genes important for cholesterol synthesis, uptake, and efflux[19]. However, whether nuclear receptors bind cholesterol and transduce levels of this lipid to modulate the TOR pathway is not known.

Here, we demonstrate that dietary cholesterol intake leads to rapid and dynamic activation of the TOR pathway in tissues of *Drosophila*. This response is modulated by the *Drosophila* RORα ortholog, HR3, which – like RORα – binds cholesterol and is activated by this ligand. Although HR3 is known to be transcriptionally upregulated by the ecdysone receptor, EcR, our results reveal that HR3 regulates growth through the TOR pathway in response to cholesterol independently of ecdysone-mediated effects. This regulation involves rapid cholesterol-induced TOR activation that in part is independent of the transcriptional functions of HR3, through an isoform of HR3 that lacks a DNA-binding domain (DBD). Reducing *HR3* levels in cells attenuates the overactivation of TOR caused by the intralysosomal accumulation of cholesterol resulting from depletion of *NPC1* (*Npc1a* in *Drosophila*). This indicates that HR3 is necessary for TOR activation by lysosomal cholesterol. Our findings suggest that HR3 activates the TOR pathway upstream of the Rag proteins. Furthermore, our findings indicate that RORα is involved in cholesterol-induced TOR activation in human cells, suggesting that a conserved function of HR3/RORα is to couple cholesterol levels to TOR-pathway activation.

## Results
### Cholesterol induces TOR-pathway activity
Dietary intake of cholesterol and its subsequent cellular absorption from the bloodstream are key for cellular cholesterol acquisition in both *Drosophila* and mammals. Cholesterol esters carried by lipoprotein particles are internalized from circulating lipoproteins through endocytosis and are transported via the endosomal pathway to lysosomes, where they are hydrolyzed so that free (unesterified) cholesterol can be trafficked between hydrophobic compartments of the cell[9,20]. Unlike mammals, which can also synthesize cholesterol de novo, *Drosophila* (and other insects) lack a functional sterol-biosynthesis pathway. Consequently, cholesterol availability in

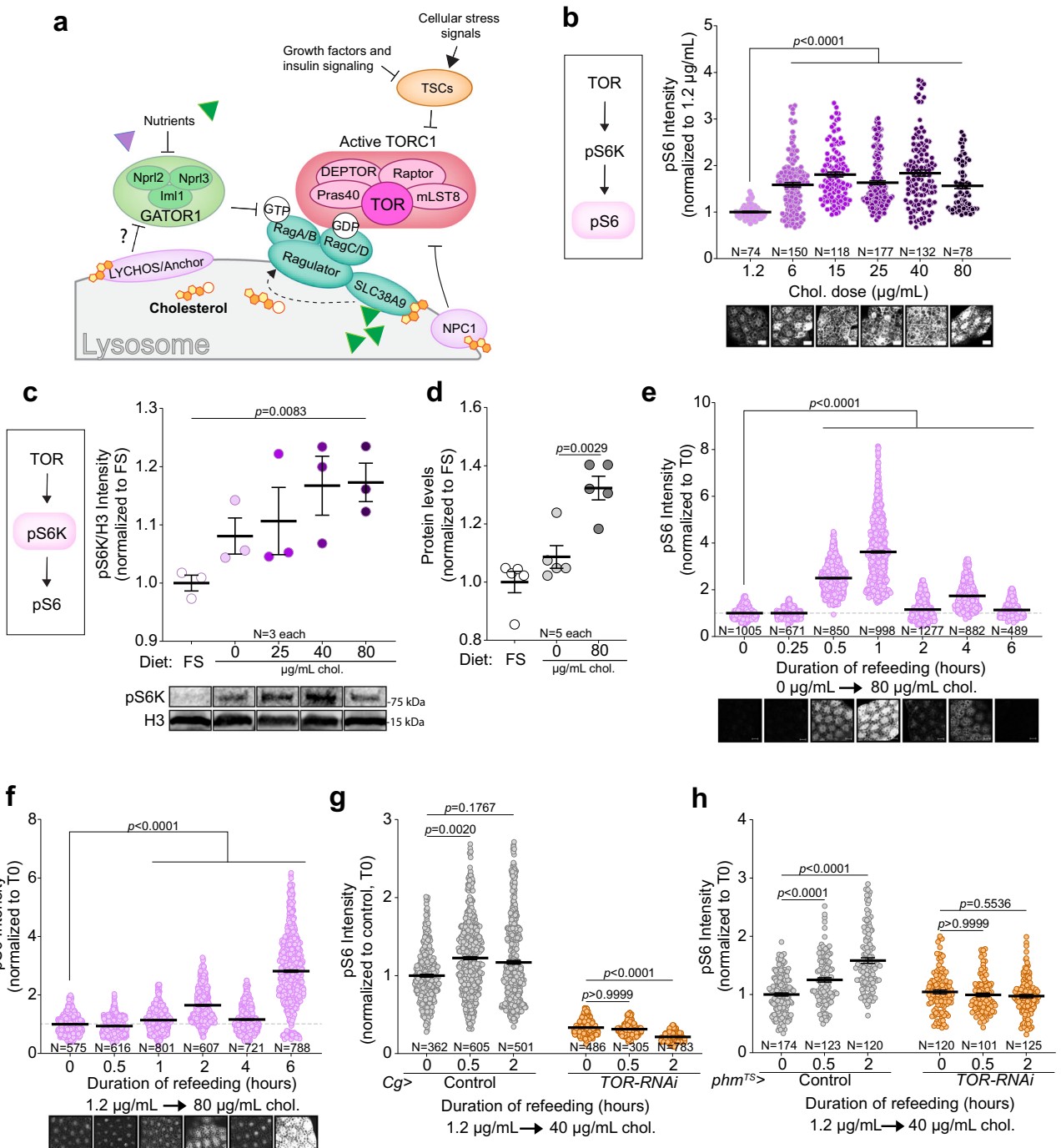

**Fig. 1 | Cholesterol availability alters TOR activity in the fat-body and pro-thoracic gland of *Drosophila*. a** Schematic of the TOR signaling pathway at the lysosome, highlighting canonical regulation by nutrients (via the Rag GTPases/Ragulator and upstream complex GATOR1) and growth factor/insulin signaling (via the TSC complex), as well as proposed cholesterol-dependent inputs based on mammalian studies (including LYCHOS and SLC38A9). **b** Quantification of cholesterol-dose sensitivity of the TOR pathway using pS6 immunohistochemistry staining in larval fat body, with representative images (scale bar 20 μm). **c** Dose response, with representative blot images, of whole-larval pS6, normalized to histone H3, 1 h after refeeding following 10 h cholesterol deprivation. FS, full starvation. Compared samples span two blots. **d** Measurement of whole-larval protein levels by BCA after 4 h refeeding following cholesterol deprivation. FS, full starvation; 0, cholesterol deprivation (0 μg/mL); 80, 80 μg/mL cholesterol. **e** Dynamics of fat-body pS6 staining levels (with representative images, 20-μm scale bar) during

cholesterol refeeding (80 μg/mL) following 10 h cholesterol deprivation. **f** Similar to d except the pre-experiment diet contained 1.2 μg/mL cholesterol. **g** Fat-body pS6 response during cholesterol re-stimulation (40 μg/mL) following 10 h feeding with 1.2 μg/mL cholesterol, in controls and fat-body *TOR* knockdowns. **h** Prothoracic-gland (PG) pS6 response during cholesterol re-feeding (40 μg/mL) following 1.2 μg/mL cholesterol limitation, in controls and with PG-specific *TOR* knockdown for 24 h in the late third instar, see Supplementary Fig. 1d for illustration of knockdown-induction and dietary-manipulation timelines. Statistics:Data are shown as mean ± SEM, normalized to lowest-dose cholesterol or earliest re-feeding timepoint (cholesterol starvation, time 0). In (**b**, **e**–**h**), each point represents the cytoplasmic pS6 intensity of a single cell; in (**c**, **d**), each is a sample of multiple animals. In (**b**–**h**) *p*-values were calculated by Kruskal-Wallis (two-sided) ANOVA test with Dunn's multiple comparisons. Source data and raw blot scans are provided in the Source Data file.

*Drosophila* can be precisely manipulated via dietary modifications[21]. We used a synthetic diet[22] that allows cholesterol supplementation at a range of doses, which promotes larval growth and development[11,23]. We specifically examined the role of dietary cholesterol during the last larval instar, a period characterized by rapid growth driven by nutrient intake. To investigate the dose-dependent effects on TOR-pathway activity, we supplemented the diet with cholesterol at concentrations ranging from 1.2 μg/mL to 80 μg/mL, reflecting ecologically relevant sterol levels[24,25]. As illustrated in Supplementary Fig. 1a, larvae were initially cholesterol-depleted by transferring them to synthetic food containing 0 μg/mL cholesterol for 10 h at 84 h after egg laying (AEL), then re-fed with synthetic food containing different doses of cholesterol for 6 h (until 100 h AEL). Dietary cholesterol dose-dependently activated TOR signaling in fat-body tissue, as indicated by increased phosphorylation of Ribosomal protein S6 (pS6) and of its activator S6 kinase (pS6K), a direct target of TOR, reflecting TOR pathway activity[26] (Fig. 1b, c). Higher cholesterol doses correlated with elevated whole-body protein levels, suggesting that cholesterol promotes organismal protein synthesis and growth (Fig. 1d), consistent with TOR activation.

When larvae were transferred from a cholesterol-free synthetic medium (0 μg/mL) to a cholesterol-replete one (80 μg/mL), we observed a time-dependent activation of the TOR pathway in the fat body (Fig. 1e). Activation occurred within 30 min, peaking at 1 h and subsequently exhibiting a dynamic oscillatory pattern. Taking into account the time required for food consumption and gut-mediated cholesterol distribution, cholesterol effects on TOR are rapid. Considering that natural food sources typically exhibit a range of sterol concentrations, rather than being absolutely sterol free, we investigated whether a transition from a low non-zero concentration to a cholesterol-replete diet would elicit a similar TOR-pathway activation pattern. Larvae were transferred to low-cholesterol food containing 1.2 μg/mL cholesterol for 10 hours at 84 h AEL (Supplementary Fig. 1a). In this condition, the transition to high (80 μg/mL) cholesterol at 94 h also resulted in increased fat-body pS6, but with levels peaking later at 6 h post-shift (Fig. 1e, f). Notably, the kinetics of the pS6 response depended on the pre-feeding cholesterol level. While transfer from cholesterol-free medium (0 μg/mL) to 80 μg/mL elicited an early maximal response (Fig. 1e), refeeding to 80 μg/mL after feeding low-cholesterol medium (1.2 μg/mL) induced a delayed increase in pS6 levels that peaked later (Fig. 1f).

Developmental growth of larval tissues is modulated by the antagonistic actions of the growth-promoting, nutrient-responsive TOR and insulin pathways and the opposing effect of ecdysone signaling, which negatively regulates systemic growth via the receptor EcR[27]. Despite cholesterol's being the precursor for steroid biosynthesis, we have previously demonstrated that cholesterol promotes growth in an ecdysone-independent manner by stimulating insulin secretion, a process dependent on the TOR pathway in the cells of the fat body and the blood-brain barrier[11]. To substantiate this further, we compared the TOR-pathway response in the fat tissue of animals fed cholesterol to the response in those fed ecdysone, using concentrations of ecdysone sufficient to drive developmental progression[28]. Larvae were switched at 84 h AEL from standard diet to a low (1.2 μg/mL)-cholesterol synthetic diet for 10 h, after which they were placed on synthetic diet supplemented with either high cholesterol (80 μg/mL) or ecdysone (Supplementary Fig. 1a). Dietary cholesterol induced a marked increase in fat-body pS6 levels compared to effects of dietary ecdysone, suggesting that cholesterol activates the TOR pathway through a mechanism that does not involve ecdysone (Supplementary Fig. 1b).

To confirm that the observed pS6 elevation in response to cholesterol availability was indeed mediated by the TOR pathway, we performed RNAi-mediated knockdown of *TOR* or the gene encoding its kinase substrate S6K, a key pathway component. Knockdown of *TOR* or *S6K* in the fat body (driven by *Cg-GAL4*, *Cg* > ) abrogated the pS6

increase induced by dietary cholesterol (Fig. 1g and Supplementary Fig. 1c). We also examined the impact of dietary cholesterol on TOR activity in the prothoracic gland (PG), an endocrine tissue that requires substantial cholesterol flux for ecdysone synthesis and thus is particularly sensitive to cholesterol[23]. Cholesterol feeding increased pS6 levels in the PG, an effect that was dependent on TOR signaling, since it was abolished by PG-specific *TOR* knockdown. This was achieved by inducing *TOR* knockdown at 120 h AEL [using temperature-sensitive (TS) *Tub-GAL80^TS* with *phm-GAL4* – together, *phm^TS* >], when larvae were transferred to low-cholesterol food containing 1.2 μg/mL cholesterol for 10 hours before refeeding with cholesterol-replete medium (40 μg/mL; Fig. 1h).

In this system, animals were raised at lower temperatures (18 °C), at which GAL80^TS represses GAL4 activity. When larvae are shifted to 29 °C, GAL80^TS becomes inactive, thereby allowing GAL4 to drive *UAS-TOR-RNAi* expression only after the temperature shift (Supplementary Fig. 1d). This makes it possible to restrict *TOR* knockdown to a defined time window in late larval development. Although the temporal profiles differ modestly between the fat body and the PG, cholesterol increases pS6 in both tissues within the examined time window. Additionally, employing a commercially available intermediate-sterol, cornmeal-based diet (NutriFly, NF)[11], we demonstrated that larvae reared chronically on this diet supplemented with 40 μg/mL cholesterol throughout development exhibited elevated pS6 levels in the PG, indicative of enhanced TOR activity (Supplementary Fig. 1e). This effect was confirmed to be TOR-dependent, as the attenuation of TOR activity via knockdown eliminated the supplementation-induced increase in pS6 levels. To further exclude an ecdysone-mediated mechanism, we used *phm^TS* > to knock down *torso*, the gene encoding the PTTH receptor, which stimulates ecdysone synthesis in the PG, and the biosynthetic enzymes *phantom* (*phm*) and *disembodied* (*dib*), which mediate essential steps in the ecdysone biosynthetic pathway. RNAi induction was performed at 120 h AEL to avoid perturbing the transition from the second to the third larval instar; at 18 °C, larvae had already reached the early third-instar stage by this time. Larvae were then placed on low-cholesterol medium (1.2 μg/mL) for 10 h. Switching these animals to a high-cholesterol diet (80 μg/mL) for 6 h increased fat-body pS6 staining to control-like levels despite impaired ecdysone production (Supplementary Fig. 1f). This suggests that the cholesterol-induced elevation of pS6 occurs independently of ecdysone production, consistent with direct cholesterol-induced activation of TOR (Fig. 1f and Supplementary Fig. 1c, e).

To further confirm the dynamic response of the TOR pathway to cholesterol, we used a transcriptional reporter of TOR function. In this system, expression of Luciferase from regulatory elements of the TOR-inhibited gene *unkempt* (*unk*) is inversely correlated with TOR activity[29]. When animals were transferred from a cholesterol-depleted synthetic medium (0 μg/mL) to a medium containing 80 μg/mL cholesterol, we observed a reduction of Luciferase activity in whole-larval lysates within 30 min, suggesting TOR-pathway activation (Fig. 2a). This pattern mirrored the dynamics observed for the pS6 response (Fig. 1e), and it was different from the systemic response to dietary protein refeeding following complete protein deprivation, which exhibited a gradual decline in reporter activity and thus indicates increasing TOR activity (Fig. 2b).

We next employed an unbiased quantitative mass-spectrometry-based phosphoproteomics approach to further elucidate the signaling response to cholesterol (Fig. 2c). These results show the same rapid and dynamic increase in S6 phosphorylation in response to cholesterol feeding when animals were transferred from a medium devoid of cholesterol (0 μg/mL) to one containing 80 μg/mL of this nutrient (Fig. 2d, left panel, and Supplementary Data 1). In contrast to this dynamic pattern, protein refeeding induced a sustained upregulation of TOR-pathway activity as reflected in pS6 levels (Fig. 2d, right panel, and Supplementary Data 2), recapitulating the results of the

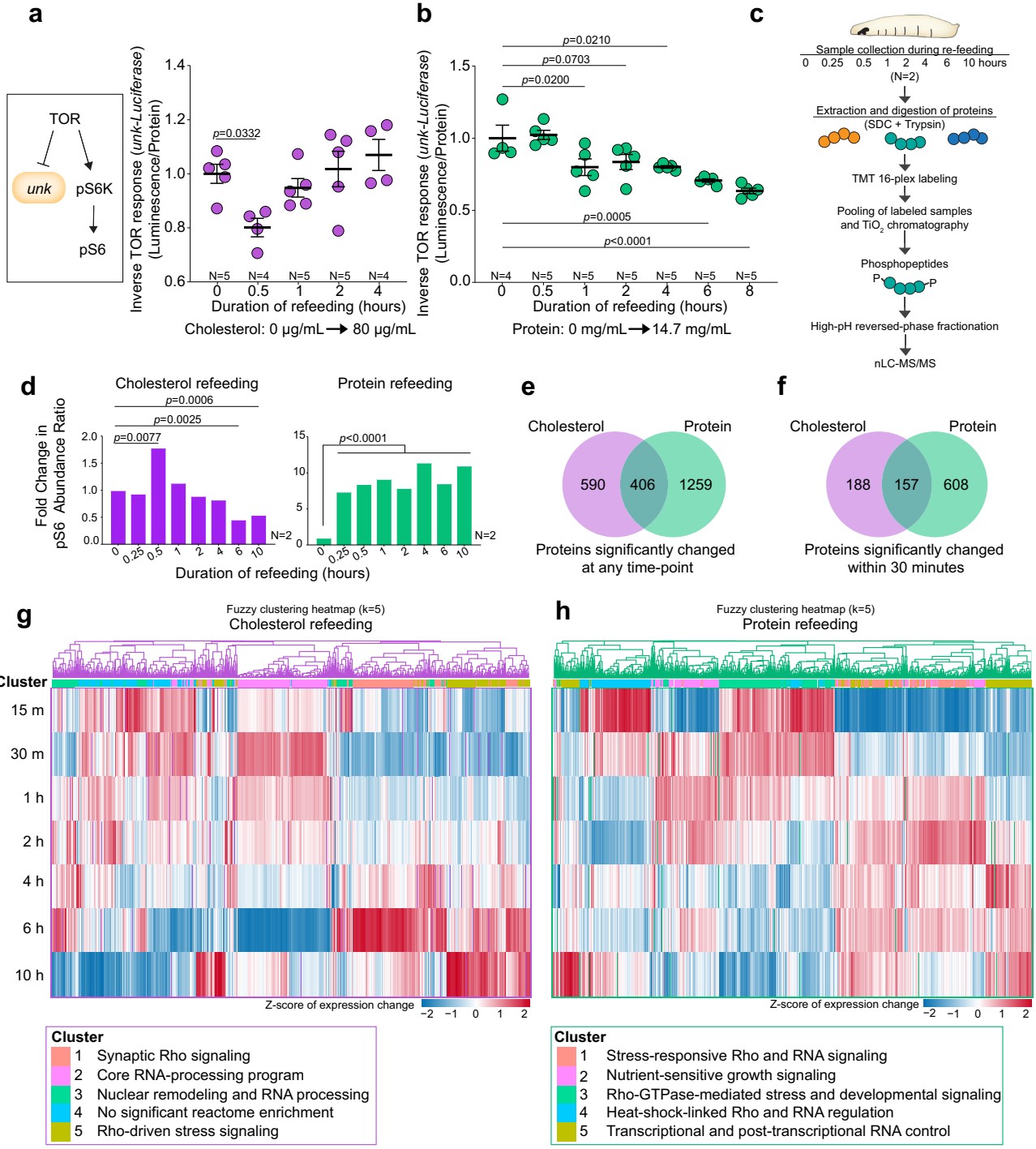

**Fig. 2 | Cholesterol and protein feeding differently activate the TOR pathway and result in phosphorylation changes. a** Whole-animal TOR response during to cholesterol refeeding (80 µg/mL), following 10 h cholesterol deprivation, using *unkempt-Luciferase* assay. Lower values reflect greater TOR activity. **b** Whole-animal TOR response to protein refeeding (14.7 mg/mL casein) following 10 h protein deprivation, using *unkempt-Luciferase* readout. Lower values reflect greater TOR activity. **c** Scheme of phosphoproteomic assays. **d** Mass-spectrometrically measured phosphorylation of ribosomal protein S6 on Ser239 (the phosphorylation site recognized by anti-pS6) over time in whole animals deprived of either protein (left) or cholesterol (right) for 10 h then refed with complete medium. **e** Phosphoproteins significantly changed at any time point during refeeding with either protein or cholesterol – 406 phosphoproteins responded in both assays.

**f** Phosphoproteins significantly changed after 30 min refeeding with protein or cholesterol; 157 phosphoproteins responded in both assays. Heatmap visualization of significantly regulated phosphosites during refeeding with cholesterol (**g**) or protein (**h**) after deprivation of that nutrient alone, grouped by fuzzy *c*-means clustering of Z-score-standardized expression values. Statistics: (**a**, **b**) Luciferase luminescence was normalized to the protein concentration of lysates. Mean ± SEM, normalized to deprivation condition (T0). Two-sided ordinary ANOVA with Dunn's multiple comparisons. **d**–**h** *p*-values were determined using two-sided ANOVA with Benjamini-Hochberg false-discovery procedure in Proteome Discoverer. A phosphorylation change was considered significant with a 30% fold change and *p* < 0.05. **g**, **h** Clustering of significant hits was performed as described in the Methods. Source data are provided in the Source Data file.

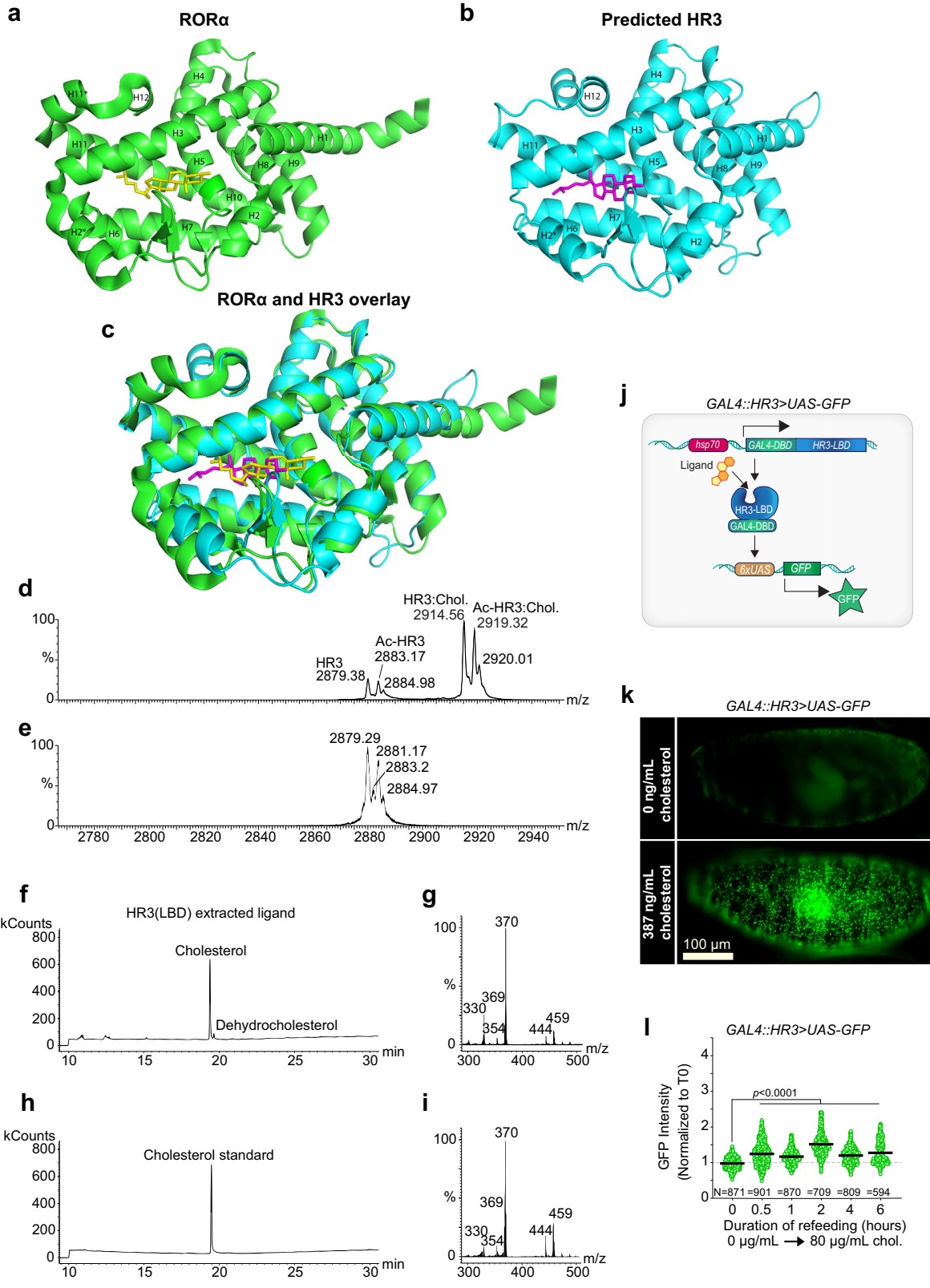

Luciferase-reporter assay. A comparative analysis of proteins demonstrating significant phosphorylation changes (fold change upon transfer from cholesterol- or protein-starvation condition to a complete medium of greater than 30 percent, with a $p$-value less than 0.05) at various refeeding intervals (15 min, 30 min, and 1, 2, 4, 6, and 10 h) revealed 996 proteins that were (de)phosphorylated in response to dietary cholesterol, whereas 1665 proteins showed altered phosphorylation in response to dietary protein (Fig. 2e, f). Ribosomal protein S6 and eukaryotic Initiation Factor 4G1 (eIF4G1), another target of TOR that connects nutrient sensing to translation[30], were among the proteins that were modified within 30 min in both cholesterol and amino-acid response (Supplementary Fig. 2a). This overlap indicates that TOR activity is rapidly promoted by both dietary cholesterol and amino acids.

**Fig. 3 | Cholesterol is a ligand of HR3. a** Structure of the human RORα ligand-binding domain (LBD) with cholesterol shown in the binding pocket. **b** Boltz-2–predicted structure of the *Drosophila* HR3 LBD with cholesterol placed in the predicted ligand-binding pocket. **c** Structural superposition of RORα (green) and the predicted HR3 LBD model (cyan) showing the relative position of cholesterol in each structure (RORα, yellow; HR3 model, magenta). Helices are labeled as indicated. **d** Mass spectrum of purified HR3 under non-denaturing conditions with peaks representing HR3 (2879.38 m/z), acetylated HR3 (2883.17 m/z, Ac-HR3), liganded HR3:cholesterol complex (2914.56 m/z, HR3:Chol.), and acetylated HR3:cholesterol complex (2919.32 m/z, Ac-HR3:Chol.) indicated. **e** Quadrupole time-of-flight collision-induced dissociation MS spectrum of purified HR3 after the loss of the ligand. **f** A gas chromatogram of a derivatized chloroform/methanol extraction of HR3 LBD produces a peak at the indicated time representing the non-ionizing ligand of purified HR3 LBD. **g** The corresponding electron ionization (EI) spectrum of the fraction containing the major peak. **h** Gas chromatogram of a derivatized cholesterol standard. **i** The EI fragment spectrum from the peak

fraction containing the standard. **j** Schematic of the HR3 ligand-sensor system. The construct consists of a GAL4 DNA-binding domain (GAL4-DBD) fused to the HR3 ligand-binding domain (HR3-LBD) under control of the heat-shock promoter (*Hsp70*). In the presence of ligand, HR3-LBD undergoes activation, enabling GAL4-DBD–dependent transcription of a *UAS-eGFP* reporter. Thus, GFP expression serves as a readout of HR3 ligand occupancy in vivo. **k** GFP fluorescence was recorded from permeabilized living embryos expressing the HR3 ligand sensor *GAL4::HR3 > UAS-GFP*, cultured for 1 h either in the absence of cholesterol (top) or with 387 ng/mL (1 μM) cholesterol in the medium (bottom). Scale bar: 100 microns. **l** *Drosophila* larval fat-body GFP signals of *GAL4::HR3 > UAS-GFP* in response to cholesterol re-feeding (80 μg/mL) following 10 h cholesterol deprivation. Statistics: **l** Data are shown as means ± SEM, normalized to T0. Each data point represents a single cell measurement. Significance determined by Kruskal-Wallis nonparametric ANOVA (two-sided) with Dunn's multiple comparisons. Source data are provided in the Source Data file.

The data also show dynamic changes in protein phosphorylation patterns over a 10 h period when animals were refed cholesterol, a response phenomenon not exhibited after protein feeding. To resolve the temporal organization of nutrient-induced signaling, we performed fuzzy *c*-means clustering of phosphorylation dynamics following cholesterol or protein refeeding across the 10 h time course (Fig. 2g, h and Supplementary Figs. 2b–d and 3a, b). This analysis revealed marked qualitative differences in how signaling networks were engaged by the two nutrients. Cholesterol refeeding elicited a highly dynamic and temporally structured phosphorylation program, with sites partitioning into multiple distinct clusters exhibiting different temporal response profiles (Supplementary Fig. 2b, c). Reactome enrichment analysis (Supplementary Fig. 2d) showed that these clusters were differentially associated with functional modules, including RNA metabolism and processing.

In contrast, protein refeeding produced more sustained and comparatively monotonic phosphorylation responses, with fewer pronounced temporal transitions within each cluster (Supplementary Fig. 3a, b). Enrichment analysis also revealed involvement of RNA-processing pathways. Collectively, temporal clustering indicates that cholesterol and protein refeeding differ in the temporal dynamics of signaling network activation, with cholesterol inducing dynamic, phase-specific responses and protein promoting more sustained signaling. Taken together, our data show that dietary cholesterol feeding induces dynamic physiological responses, including TOR activation, in fat-body and PG cells and in whole animals. Furthermore, these changes are distinct in some ways from those that occur in response to amino-acid replenishment.

## HR3 is activated by cholesterol binding

Previous research suggests that HR3 interacts genetically with S6K[31], and the closest human orthologs of HR3, RORα and -β, have been found to bind cholesterol[32]. However, the functional significance of this interaction remains unclear. The sequence similarity between the ligand-binding domains (LBDs) of HR3 and RORα[17] suggests that cholesterol may also be a natural ligand of HR3. To test this hypothesis in silico, we used the molecular modeling package Boltz-2 to generate a structural model of the HR3 ligand-binding domain (LBD) and to assess whether cholesterol would be placed within the predicted HR3 pocket in a pose similar to that observed in the experimentally determined RORα LBD structure (Fig. 3a, b). Here, without intervention, the Boltz-2 docking process placed cholesterol into the HR3 ligand-binding pocket in the same orientation as the cholesterol ligand in RORα (Fig. 3a, b and Supplementary Fig. 4a, b). Consistent with the strong similarity of their primary sequences, we also found close alignment of structural elements between HR3 and RORα (Fig. 3c). As observed in the RORα-cholesterol structure, the 3β-hydroxyl group of cholesterol in the HR3 LBD is aligned with the side chains of two conserved

arginine residues (R328 and R331; Supplementary Fig. 4c, d). In RORα, R367 and R370 (helix 5), along with a hydrogen-bonded water molecule in the pocket, were identified as contributing to interactions with the 3β-hydroxyl group of cholesterol[33]. In the structural alignment of RORα and the predicted HR3 structure, the HR3 residues R328 and R331 align closely with their RORα counterparts R367 and R370 (Supplementary Fig. 4c).

Following this structural prediction, we asked whether the cholesterol-binding pocket is also conserved at the sequence level. A sequence alignment of *Drosophila* HR3 with human RORα showed that many residues implicated in ligand binding are conserved in HR3 (Supplementary Fig. 4d). Several residues experimentally shown to be critical for cholesterol binding due to their close proximity to the ligand ($\leq 4$ Å) are strictly conserved between RORα and HR3 or replaced by residues with similar hydrophobic character[32,33], consistent with tight hydrophobic packing around the cholesterol ligand. Furthermore, hydrogen bonding between H484 and Y507 has been suggested as critical for cholesterol binding in RORα, perhaps by maintaining H484 in an amenable position. Both of these residues are conserved in HR3, underlining the conservation of key elements of the cholesterol-binding pocket[32]. Together, these sequence features complement the docking and structural alignment (Fig. 3a–c and Supplementary Fig. 4a–c) and are consistent with the HR3 LBD's accommodating cholesterol.

To determine whether HR3 indeed binds cholesterol, we expressed the HR3 LBD in Hi5 insect cells and conducted affinity purification of this recombinant protein, followed by electrospray ionization mass-spectrometry (ESI-MS) under non-denaturing (native) conditions to identify any co-purified ligands. In non-denaturing conditions, full-mass-range scans detected peaks suggesting a receptor:ligand complex with an added mass approximately equal to that of cholesterol (386.6 Daltons) (Fig. 3d and Supplementary Fig. 4e). Collision-induced dissociation (CID) of the same native-MS preparation dissociated the ligand and yielded the corresponding apo/modified HR3 species (Fig. 3e). To confirm the identity of the ligand, purified HR3 LBD was extracted with acidified chloroform/methanol to isolate co-purifying small molecules, which were derivatized and analyzed by electron ionization gas chromatography-mass spectrometry (EI-GC/MS). Chromatography of the extract yielded a single major peak at 19 min (Fig. 3f) which, when ionized, produced a spectrum characteristic of derivatized cholesterol (Fig. 3g). A cholesterol standard processed through the same protocol yielded a close match to the spectrum of the putative ligand (Fig. 3h, i). Taken together, evidence from the combination of protein ESI-MS and GC/MS demonstrates that the LBD of HR3 binds cholesterol with a 1:1 stoichiometry.

To assess the binding of cholesterol to HR3 in vivo, along with functional outcomes of this binding, we tested the ability of cholesterol to activate a transgenic HR3 ligand-sensor construct (GAL4::HR3).

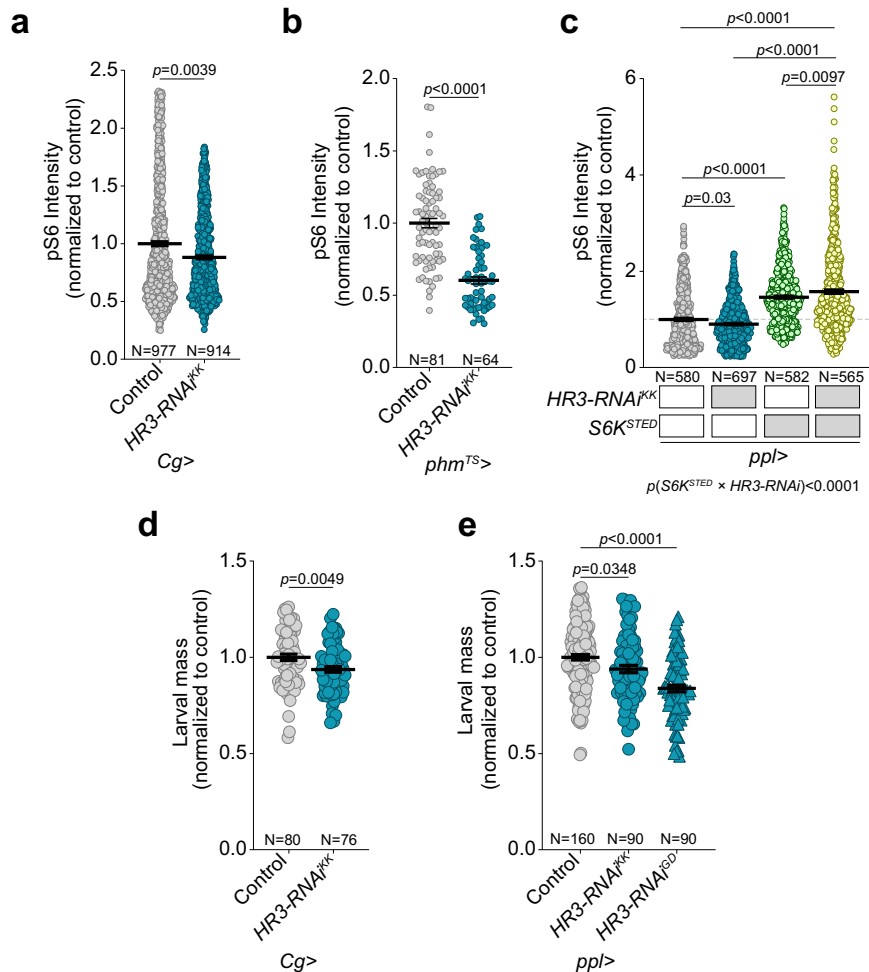

**Fig. 4 | Knockdown of *HR3* decreases systemic growth and TOR activity.**
**a** Quantification of pS6 staining in fat-body cells of controls and animals expressing RNAi against all isoforms of *HR3* in the fat body using *Cg>*. **b** Quantification of pS6 in the cells of the PG in controls and animals expressing *HR3* knockdown using *phm^TS^>*. RNAi-mediated knockdown was induced by temperature shift -24 h prior to sample collection at larval stage L3, as described in Supplementary Fig. 1d. **c** Quantification of pS6 in fat-body cells of control larvae and animals expressing *HR3* RNAi in the fat tissue using *ppl >* , with and without co-expression of

constitutively active S6K (*S6K^STED^*). **d** Body mass of control larvae and larvae expressing fat-body knock-down of *HR3* using *Cg>*. **e** Body mass of control larvae and animals expressing either of two independent HR3-RNAi constructs using a second driver (*ppl>*). Statistics: (**a**, **b**, **c**, **d**, **e**) Graphs plot mean ± SEM.
**a**, **b**, **d** Significance determined using two-tailed unpaired *t*-test. **c**, **e** Kruskal-Wallis ANOVA test (two-sided) with Dunn's multiple comparisons, with two-way ANOVA test for epistasis in **c**. In (**a**, **b**, **c**) each data point reflects a single cell; in (**d**, **e**) each point reflects a single larva. Source data are provided in the Source Data file.

In this system, the LBD of HR3 is fused with the DNA-binding domain (DBD) of GAL4. Ligand binding to the LBD of the fusion protein activates expression of a *UAS-GFP* reporter gene[34] (Fig. 3j). Cholesterol addition to permeabilized living embryos carrying this system induced widespread GFP expression, indicating that exogenous cholesterol can bind and activate the LBD of HR3 (Fig. 3k). We next used this ligand sensor to examine whether the LBD of HR3 can be activated by cholesterol derived from the diet. Cholesterol refeeding of larvae after a deprivation period on synthetic diet lacking cholesterol led to increased GFP staining in the fat body, consistent with binding of cholesterol (or a derivative) obtained through the diet (Fig. 3l). We also confirmed that this cholesterol treatment produces the characteristically dynamic fat-body TOR activation pattern as measured by pS6 staining (Supplementary Fig. 5a). These findings collectively indicate that cholesterol binds to the LBD of HR3 and that this protein functions as a cholesterol-responsive receptor.

## HR3 modulates TOR activity and regulates systemic growth

Given the activation of HR3 by dietary cholesterol, we investigated whether HR3 might influence systemic growth via modulation of the

TOR signaling pathway on standard diet. Knockdown of *HR3* in the fat body or PG using an RNAi construct that targets all of the annotated transcript variants of *HR3* led to reductions in pS6 staining in these tissues (Fig. 4a, b and Supplementary Fig. 1a for fat-body conditions and Supplementary Fig. 1d for PG-specific *GAL80^TS^* induction conditions). This attenuation indicates a regulatory interaction whereby HR3 promotes S6 phosphorylation, perhaps mediated by TOR and S6K. To more directly show that phosphorylation of S6K and S6 occurs downstream of HR3, we simultaneously expressed a constitutively active S6K allele in the fat body, which suppressed the pS6 reduction induced by *HR3* knockdown (Fig. 4c), implying that HR3 modulates S6 phosphorylation via the TOR-S6K signaling pathway. It is well established that reduced fat-body TOR activity attenuates systemic growth[35]. In line with this, we found that *HR3* knockdown in the fat body reduced systemic growth, as reflected in a decrease in larval mass, validated using two distinct fat-body-specific drivers (*Cg>* and *pumpless-GAL4, ppl >* ) and three independent RNAi lines (Fig. 4d, e and Supplementary Fig. 5b).

To distinguish whether the reduced larval mass resulted from slower growth rather than a developmental delay or arrest, we next

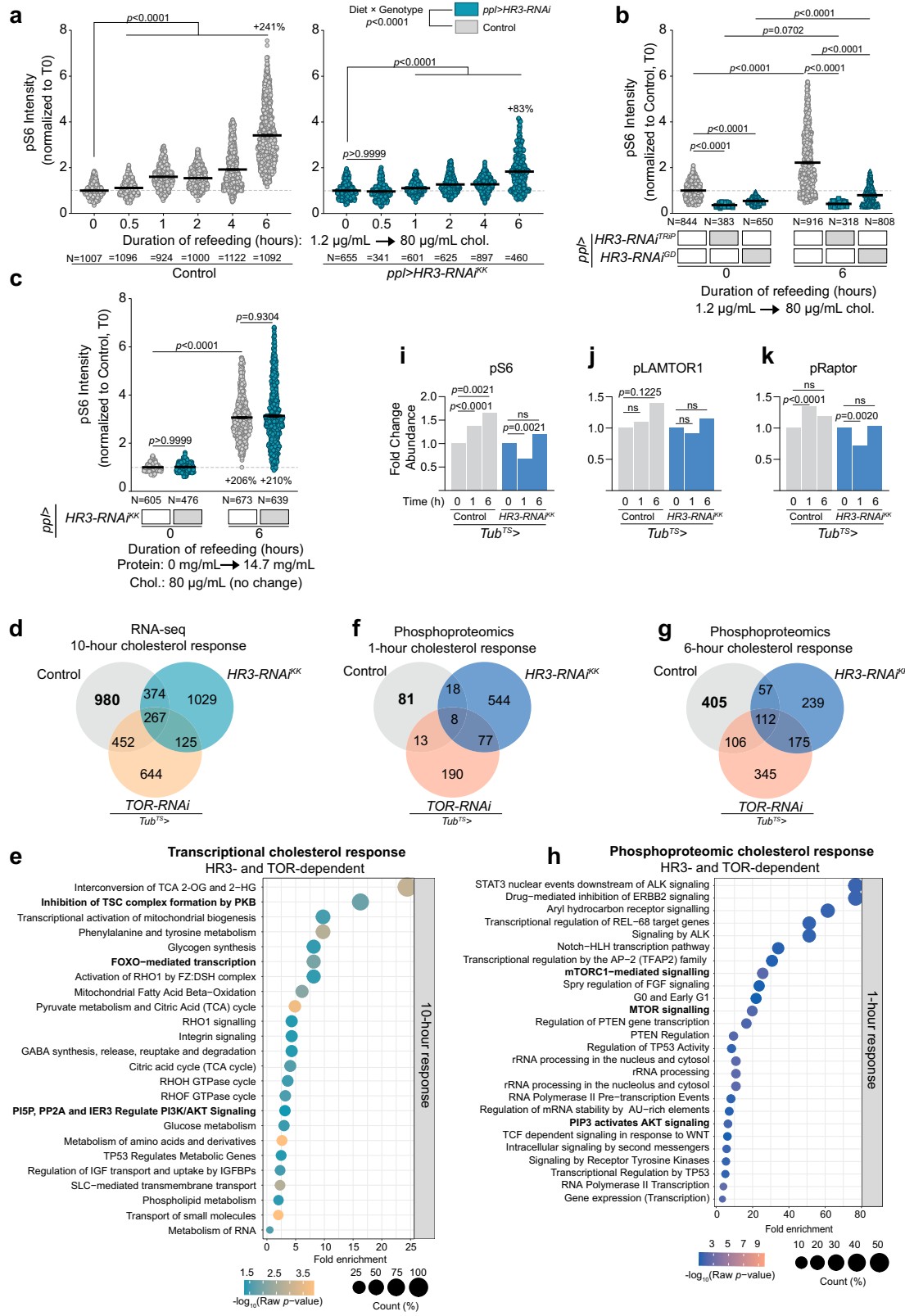

induced *HR3-RNAi* later in development, at the time when larvae transition into the third and final instar. This approach avoids perturbing earlier developmental transitions and allows growth rate to be assessed independently of stage-specific timing effects. Knockdown of *HR3* in the fat body (*Cg-GAL4* with *Tub-GAL80^TS^: Cg^TS^* >) or in the PG (*phm^TS^* >) was induced 120 h AEL by transfer from 18 °C to 29 °C (Supplementary Fig. 1d), and we monitored both larval mass and the

timing of pupariation (Supplementary Fig. 5c, d). Although fat-body-targeted *HR3* knockdown during the third instar significantly reduced larval body size, the timing of pupariation was unchanged (Supplementary Fig. 5c). Because developmental progression remained normal while final size decreased, we conclude that *HR3* depletion in the fat body slowed the rate of growth, consistent with earlier reports showing that reduced fat-body TOR activity diminishes systemic

**Fig. 5 | HR3- and TOR-dependent cholesterol response. a** Phospho-S6 levels in fat-body cells of controls and animals expressing *HR3-RNAi*, after cholesterol refeeding (80 µg/mL) following depletion (1.2 µg/mL in medium). **b** pS6 levels in larval fat-body cells in animals expressing independent *HR3-RNAi* constructs in the fat body, six hours after dietary cholesterol replenishment. **c** pS6 levels in larval fat-body cells of controls and animals with fat-body specific *HR3* knockdown, after amino-acid refeeding (14.7 mg/mL) following 10 h on protein-free medium. **d–k** Transcriptional and proteomic responses to cholesterol refeeding (80 µg/mL) over time in control *Tub-GAL80^TS; Tub-GAL4* larvae and those ubiquitously expressing *HR3-RNAi* or *TOR-RNAi*, following 10 h low-cholesterol (1.2 µg/mL) feeding. At 120 h AEL larvae were transferred to 29 °C to induce RNAi expression for 9 h prior to cholesterol treatments, resulting in at least ~24 h induction before the sample-collection timepoints (see Supplementary Fig. 1d). **d** Number of genes differentially expressed between the three genotypes at the 10 h time point. In bold, "980" reflects genes whose cholesterol response is dependent on both HR3 and TOR. **e, h** Selected pathways enriched in these 980 genes, identified using Panther

and Reactome. See also Supplementary Data 3 and 4. **f, g** Significant phosphorylation changes after 1 and 6 h cholesterol re-feeding. **i, j, k** Phosphorylation of S6, LAMTOR1, or RAPTOR over time after cholesterol re-feeding in controls and *HR3* knockdowns. Statistics: (**a–c**) Graphs plot mean ± SEM, each data point representing a single cell. Data normalized to control at T0. Statistical significance determined using Kruskal-Wallis ANOVA (two-sided) with Dunn's multiple comparisons. Interaction between diet and genotype was determined using two-way ANOVA. **d** *P*-values were computed by Wald test (two-sided) with Benjamini-Hochberg FDR estimation. **f, g** Changes >30% with *p* < 0.05 were taken as significant. *P*-values were determined using two-sided ANOVA with Benjamini-Hochberg false-discovery procedure. **e, h** pathway analysis with two-sided Fisher's *t*-test; only pathways with *p* < 0.05 are included. For panels (**i–k**), phosphosite abundances are shown as fold change relative to time 0 within each genotype, and *p*-values were determined using two-sided ANOVA with Benjamini-Hochberg procedure. Source data are provided in the Source Data file.

growth in *Drosophila*[13,35]. In contrast, the consequences of TOR-pathway manipulation in the PG are more complex, because TOR activity in this endocrine tissue regulates ecdysone production, which in turn sets developmental timing[11,36–38]. Consistent with this established role, *HR3* knockdown induced in the third instar in the PG altered the timing of pupariation (Supplementary Fig. 5d), reflecting the PG's role in controlling the timing of developmental transitions rather than the growth rate.

### HR3 regulates TOR pathway activity in response to cholesterol

Given the role of HR3 in binding cholesterol and modulating TOR pathway activity, we investigated whether HR3 is necessary for the cholesterol-induced activation of the TOR pathway. Larvae, at 84 hours AEL, were first placed on a 1.2 µg/mL-cholesterol diet for 10 h, followed by a shift to a high-cholesterol diet (80 µg/mL). Fat-body-specific knockdown of all *HR3* variants through RNAi resulted in diminished cholesterol-induced activation of the TOR pathway over 6 h, as indicated by reduced pS6 levels (Fig. 5a). Consistent with our previous findings (Fig. 1f), we found that elevated dietary cholesterol rapidly activated TOR signaling in fat tissue within 1 h, with the activation progressively intensifying up to 6 h post-transfer. By contrast, cholesterol replenishment failed to elevate pS6 levels within an hour in animals with fat body-specific *HR3* knockdown, and the long-term activation was attenuated in these animals. These findings were further corroborated using two further independent RNAi constructs also targeting all *HR3* isoforms, showing that fat-body-specific knockdown of *HR3* largely abolished cholesterol-stimulated TOR pathway activation (Fig. 5b). We next explored whether HR3 selectively activates TOR in response to cholesterol. Contrary to the diminished TOR activation by cholesterol, the response of TOR to dietary amino acids remained unperturbed by *HR3* knockdown (Fig. 5c). This indicates that HR3 does not influence the activation of TOR by dietary amino acids, distinguishing its role in cholesterol-mediated TOR activation.

To further examine any potential genetic interaction between HR3 and EcR, we knocked down *EcR* in the fat body and assessed the TOR pathway's response to cholesterol. Given that HR3 is an ecdysone-inducible gene[17], loss of *EcR* would be expected to reduce *HR3* expression, which would in turn dampen the TOR response to cholesterol, similar to the effects of *HR3* knockdown itself. Contrary to this expectation, *EcR* knockdown in the fat body enhanced the response to cholesterol (Supplementary Fig. 5e). EcR suppresses cellular cholesterol uptake[39], so this result is consistent with increased cholesterol uptake that activates TOR. The opposite effects of *HR3* and *EcR* knockdowns suggest that HR3 contributes to cholesterol-induced TOR activation in an EcR-independent manner.

To gain deeper insight into the HR3-mediated mechanisms underlying cholesterol-induced TOR activation, we profiled the genomic and phosphoproteomic responses to cholesterol in animals

ubiquitously expressing knockdown of either *HR3* or *TOR*. Using *Tub-GAL80^TS* in conjunction with the *Tub-GAL4* driver (*Tub^TS* >), RNAi was induced in larvae at 120 h AEL by switching animals from 18 °C to 29 °C (as outlined in Supplementary Fig. 1d). Nine hours later, these animals were switched from a standard diet to a low-cholesterol diet (1.2 µg/mL) for 10 h. Subsequently, larvae were transferred to a high-cholesterol diet (80 µg/mL) for varying lengths of time to assess their response. In our RNA-seq profiling efforts, we identified 2073 genes exhibiting significant regulatory changes (both upregulation and downregulation) in response to a high-cholesterol diet (80 µg/mL) in control larvae, compared against age-matched larvae maintained on a low cholesterol diet (1.2 µg/mL), which served as reference point for all comparisons (Fig. 5d and Supplementary Data 3). The transcriptional response to cholesterol was strongly dependent on both HR3 and TOR, as evidenced by a loss of cholesterol response in 980 (47%) of these genes in animals lacking either *HR3* or *TOR*. This underscores the critical roles of HR3 and TOR in orchestrating the transcriptional machinery responsive to cholesterol. Additionally, other sets of genes became aberrantly responsive to cholesterol upon the knockdown of either *HR3* or *TOR*, with an overlap of 125 genes between these sets, suggesting that these factors normally act to repress such responses. Collectively, these data imply that HR3 and TOR are required to initiate a specific transcriptional response to cholesterol through regulatory mechanisms that activate certain genes while suppressing the response of others to cholesterol. Further investigation was carried out through Reactome pathway analysis applied to the 980 genes that require both HR3 and TOR for proper cholesterol response. This analysis revealed a significant enrichment of genes involved in pathways that include the TSC2 complex and insulin signaling, specifically PI3K/AKT/FOXO-related pathways (Fig. 5e). Given that TSC2 is a critical inhibitor of the TOR signaling pathway, and considering the interplay between TOR and insulin signaling, these findings further support our model in which cholesterol modulates TOR signaling via a mechanism that requires HR3.

We next conducted phosphoproteomic analysis for deeper insights into the HR3- and TOR-dependent pathway responses to cholesterol. All comparisons were made against the baseline established at the zero-hour-refeeding time point of control larvae maintained on a 1.2 µg/mL cholesterol diet for the preceding 10 h, before transfer to cholesterol-replete medium. Our results indicated a rapid (within 1 h) alteration of phosphorylation of 81 proteins (those showing a change of over 30 percent with a *p*-value of less than 0.05) that are HR3- and TOR-dependent (they do not occur in the absence of either protein; Fig. 5f and Supplementary Data 4). Notably, a substantial number of proteins (544) changed phosphorylation in response to cholesterol in *HR3* knockdown animals but not in controls, suggesting that HR3 normally suppresses these changes. Mirroring the pattern seen in fat-body pS6 levels upon transferring animals from

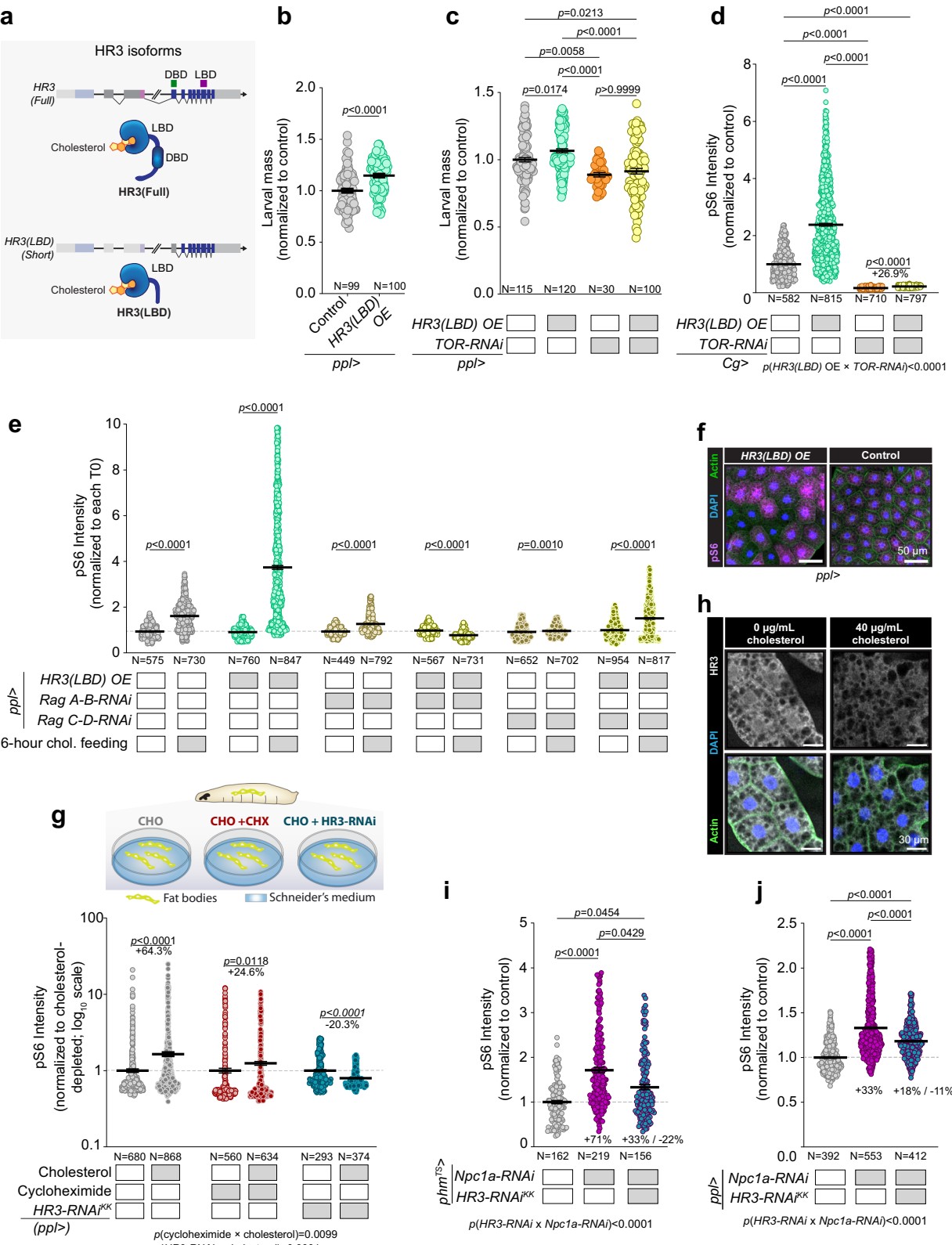

1.2 µg/mL to 80 µg/mL cholesterol (Figs. 1e and 5a), the cholesterol response was greater at 6 h relative to 1 h with 405 proteins exhibiting HR3-and-TOR-dependent phosphorylation changes (Fig. 5g and Supplementary Data 4). Pathway analysis of this HR3-and-TOR-dependent phosphoproteomic response highlighted the TOR pathway, along with ribosome- and translation-related processes, which are regulated by TOR activation (Fig. 5h and Supplementary Fig. 5f). The HR3-

dependent phosphorylations included S6 itself (Fig. 5i), consistent with the HR3-dependent cholesterol-mediated phosphorylation of S6 in the fat body (Fig. 5a, b), as well as LAMTOR and RAPTOR (Fig. 5j, k), two key regulatory components acting upstream of TOR that regulate its activity via interactions with the Rags. This implies HR3 operates at the level of, or upstream of, the LAMTOR (Ragulator) complex and RAPTOR-dependent TOR activation, consistent with insights from

**Fig. 6 | HR3 mechanism of TOR pathway modulation. a** The *HR3* locus gives rise to proteins containing both a DBD and a LBD as well as an unannotated variant lacking the DBD[31]. **b** Larval mass at 96 h AEL of controls and animals overexpressing *HR3(LBD)* in the fat body, grown on standard fly food (STD-FF). **c** Mass of animals like those of panel b, with or without simultaneous *TOR* knockdown. **d** Phospho-S6 staining in 96 h-AEL animals expressing *HR3(LBD)* and *TOR-RNAi*, reared on high-cholesterol diet (NF + 40 µg/mL). **e** Phospho-S6 staining in fat-body cells of animals expressing combinations of HR3(LBD) and *Rag* knockdowns, with and without 6-hour cholesterol refeeding (80 µg/mL) following deprivation (1.2 µg/mL). **f** Anti-pS6 stains of fat-body tissue from controls and animals overexpressing HR3(LBD) after 6 h cholesterol re-feeding (80 µg/mL), representative of at least 30 images per genotype. Scale bars, 50 µm. Supplementary Fig. 6d, e quantify staining variability. **g** Phospho-S6 staining in fat-body tissue dissected from 96 h-AEL larvae (controls and fat-body-specific *HR3* knockdowns) after cholesterol depletion and replenishment, with and without cycloheximide treatment. **h** Immunostaining against HR3 in larval fat-body tissue after cholesterol depletion and replenishment. Full images are shown in Supplementary Fig. 6h. For each, one tissue was imaged, representing ~15 on the slide. **i** pS6 staining intensity in cells of the PG expressing RNAi against *Npc1a* and *HR3*. Knockdown was induced ~24 h prior to sample collection at larval stage L3 (see Supplementary Fig. 1d). **j** pS6 staining intensity in cells of the fat body in controls and animals expressing RNAi against *Npc1a* and *HR3*. Statistics: Data are normalized to controls and graphed as mean ± SEM. **b, e, g** Each point represents a single cell. Data pairs are normalized to each cholesterol-starvation condition or controls. Pairwise *p*-values were calculated using two-sided Mann-Whitney tests, and interaction between variables was assessed by two-way ANOVA. **c, d, h, j** Each point represents a single cell. Significance estimated by Kruskal-Wallis ANOVA (two-sided) with Dunn's multiple comparisons. **c, d, g, i, j** Interactions between variables were assessed by two-way ANOVA. Source data are provided in the Source Data file.

other recent studies in mammalian systems indicating that cholesterol activates the TOR pathway via these components[3,14]. At the 6-hour time point, only 239 cholesterol-induced changes in protein phosphorylation were observed in *HR3* knockdown animals (Fig. 5g), in contrast to the 544 alterations observed at the one-hour time point (Fig. 5f). This pattern suggests that while HR3 normally activates the TOR pathway within 1 h in response to cholesterol, it concurrently modulates a broad spectrum of proteins and blocks their phosphorylation, presumably to maintain a balanced cellular response. This dual role of HR3 suggests a complex regulatory mechanism through which it both stimulates and restrains the phosphorylation of proteins in a time-dependent manner in response to cholesterol levels, ensuring a precisely controlled cellular adaptation.

## Cholesterol-induced TOR activation via HR3 is independent of its DNA-binding function

To delineate how HR3 conveys cholesterol availability to the TOR signaling cascade, we investigated the hierarchical relationship between HR3 and TOR. Given the previously documented genetic interaction between S6K and an HR3 isoform lacking the DNA-binding domain (DBD)[31], we explored whether this HR3(LBD) isoform participates in HR3's mediation of cholesterol's physiological effects (Fig. 6a). Overexpression of this shorter *HR3* isoform containing only the LBD in the fat body led to increased systemic growth, as evidenced by increased larval weight (Fig. 6b) without altering the timing of pupariation (Supplementary Fig. 6a). To ascertain whether the systemic growth effects of HR3(LBD) expression were mediated by the TOR pathway, we simultaneously silenced *TOR* expression in the fat body, which in turn abrogated the growth-promoting effects of the overexpressed *HR3(LBD)* variant (Fig. 6c). This finding indicates that the HR3(LBD) isoform requires TOR function for its growth-promoting effects and suggests that HR3 can modulate TOR-pathway activity through a mechanism that does not require its DBD function. To pursue this notion further, we assessed whether HR3(LBD) is a mediator of cholesterol-stimulated TOR pathway activation. Chronic exposure to a cholesterol-enriched diet of NF supplemented with 40 µg/mL cholesterol combined with overexpression of HR3(LBD) in the fat body significantly enhanced pS6 levels, suggesting that HR3(LBD) promotes cholesterol-induced TOR-pathway activation (Fig. 6d). Aligning with phosphoproteomics findings that potentially position HR3 upstream of the Rag proteins, the augmented pS6 levels caused by HR3(LBD) overexpression were entirely negated by concurrent *TOR* knockdown, demonstrating that HR3(LBD) acts at or above the level of TOR.

Next, we asked whether HR3(LBD) is involved in the acute activation of the TOR pathway in response to cholesterol. Larvae, 84 h AEL, were initially fed a diet containing 1.2 µg/mL cholesterol for 10 h, then transferred to a diet with the higher cholesterol concentration of 80 µg/mL to examine their response. In animals overexpressing *HR3(LBD)* in the fat body and fed 80 µg/mL cholesterol for 1 h, pS6 levels were significantly elevated compared to controls, an effect that was blocked by *TOR* knockdown (Supplementary Fig. 6b). The increase in pS6 was even more pronounced at the 6-hour time point, indicating that the HR3(LBD) isoform is a key mediator of cholesterol-induced TOR-pathway activation (Fig. 6e). This effect was not observed when the diet was supplemented with ecdysone instead of cholesterol, demonstrating that HR3 mediates a cholesterol-specific response (Supplementary Fig. 6c). Notably, this *HR3(LBD)* overexpression also induced significant intercellular variability in pS6 levels after 6 h of cholesterol stimulation (Fig. 6f and Supplementary Fig. 6d, e), suggesting that overexpressing *HR3(LBD)* leads to a progressive destabilization or desynchronization of TOR-pathway dynamics. These findings support the hypothesis that HR3 plays a cell-autonomous role in cholesterol sensing, acting upstream of TOR.

To further dissect how different HR3 domains contribute to TOR regulation, we compared the effects of overexpressing full-length HR3 or a truncated form that retains only the DNA-binding domain, HR3[K243X], with those of HR3(LBD). *HR3[K243X]* carries a premature stop codon at residue 243 and encodes a protein that preserves the DNA-binding domain but lacks the ligand-binding and activation domains and therefore cannot respond to cholesterol or other ligands. Whereas HR3(LBD) overexpression enhanced cholesterol-induced pS6 staining in the fat body, both full-length HR3 and HR3[K243X] blunted the pS6 response, with full-length HR3 causing the stronger attenuation (Supplementary Fig. 6f). Because these constructs maintain DNA-binding capacity, but only full-length HR3 retains an intact LBD, these observations suggest that HR3's DBD-dependent occupancy of target loci generally restrains TOR signaling, and that ligand-bound full-length HR3 can further engage transcriptional programs that dampen cholesterol-stimulated TOR activity, in contrast to the rapid and possibly non-genomic, LBD-dependent activation mediated by HR3(LBD). This dual behavior aligns with our genome-wide transcriptional and phosphoproteomic analyses (Fig. 5d, f, g), which show that HR3 not only engages a large set of cholesterol-responsive genes and phosphoproteins but also suppresses inappropriate or excessive cholesterol responses. consistent with a model in which HR3 provides both activating and feedback-limiting functions to maintain balanced TOR-pathway output.

## HR3 mediates cholesterol-responsive TOR signaling via genomic and non-genomic mechanisms upstream of the Rag GTPases

Nuclear receptors are ligand-regulated transcription factors that canonically exert their effects through transcriptional regulation[17]. However, rapid effects are often mediated by non-genomic actions in which the receptor is localized to membranes or the cytosol and directly influences signal-transduction pathways[40,41]. Given that HR3(LBD) mediates rapid non-genomic effects[31], we asked whether cholesterol-mediated TOR activation might entail both genomic and

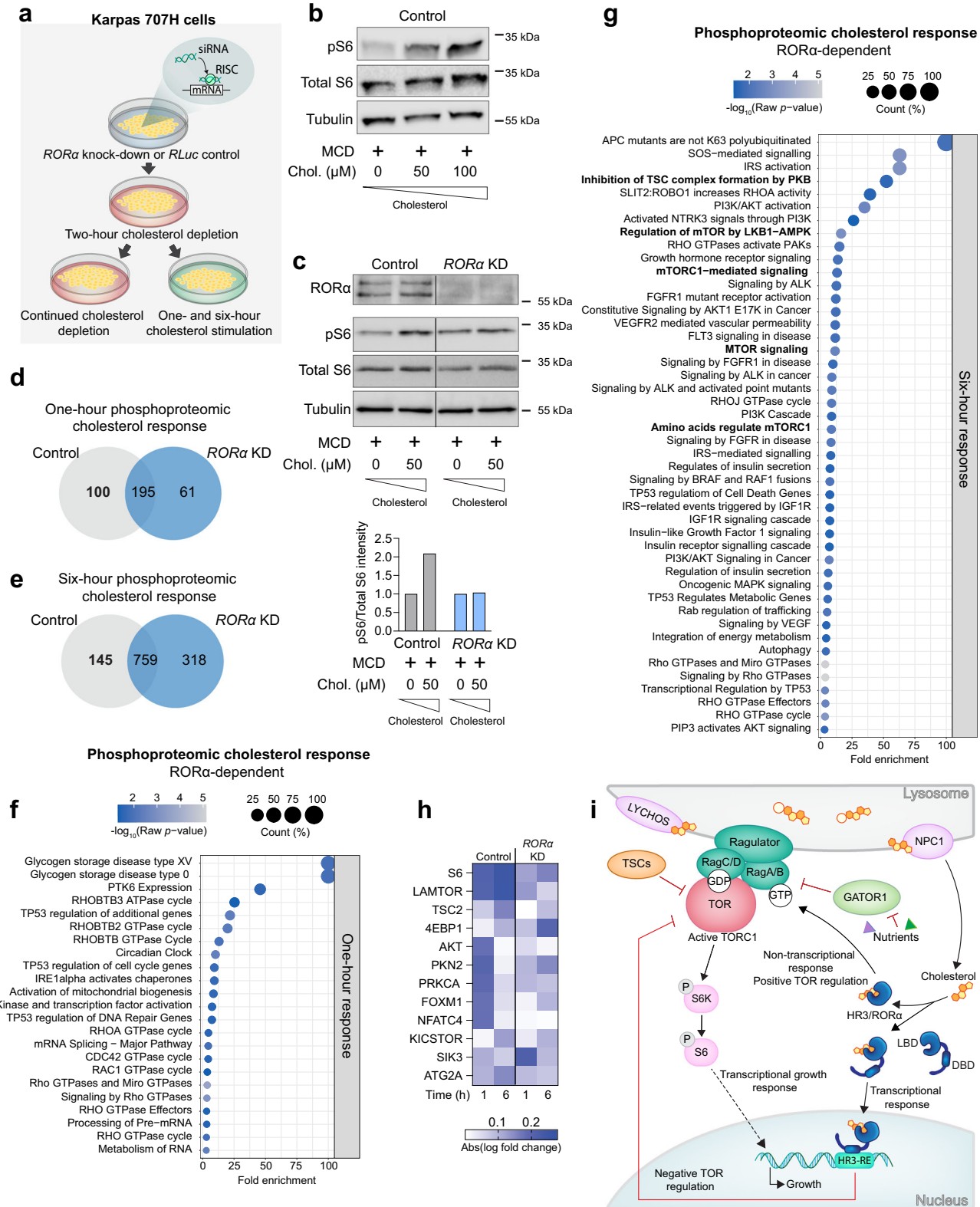

non-genomic HR3 functions. To determine the mechanism by which cholesterol activates TOR, we inhibited translation using cyclohex-imide to suppress any regulation of the TOR pathway by newly expressed proteins. We cultured ex-vivo fat-body tissues from larvae at 96 AEL in Schneider's medium supplemented with lipid-depleted serum. Tissues were depleted of sterols through a 2 h incubation with methyl-beta-cyclodextrin (MCD, 0.75%), after which they were stimu-lated with cholesterol complexed with MCD (0.1%). This ex-vivo

cholesterol stimulation increased fat-body pS6 levels in tissues dis-sected from control animals (Fig. 6g and Supplementary Fig. 6g). Cholesterol stimulation also increased pS6 levels in fat-body tissues treated with cycloheximide, although not as robustly as in controls without cycloheximide. This suggests that a significant portion of the rapid response does not require new protein synthesis. Together this implies that cholesterol activates the TOR pathway through mechan-isms independent of translation, and potentially transcription, as well

**Fig. 7 | Conserved mechanism of cholesterol regulation of the TOR pathway by human RORα in vitro. a** Illustration of in-vitro experiments using the human cell line Karpas 707H, chosen for its high expression of *RORα*. *RORα* was knocked down in the cells using siRNA, and the cells underwent a two-hour cholesterol-depletion treatment using 0.75% methyl-β-cyclodextrin (MCD) and 0.5% lipid-depleted serum (LDS). The cells were replenished with cholesterol using MCD:cholesterol (0.1% MCD:50 μM cholesterol or 0.2% MCD:100 μM cholesterol) with 0.5% LDS or simply changed to new depletion medium and were sampled 1 and 6 hours later. Protocol based on Shin et al.[14] and Castellano et al.[3] **b** Immunoblot against pS6, total S6, and α-Tubulin of control and Karpas-cell extracts after 1 h cholesterol replenishment or mock treatment. **c** Immunoblot against RORα, pS6, total S6, and alpha-Tubulin of extracts from control and *RORα*-siRNA cells, following 1 h cholesterol replenishment or mock treatment. The ratio of pS6 to total S6 staining is quantified, normalized to cholesterol-depleted condition for each genotype.
**d, e** Phosphoproteomic differences between cholesterol-replenished and mock-treated cholesterol-depleted control and *RORα*-siRNA-treated cells after 1 (**d**) and 6 (**e**) hours of treatment. The bolded control number reflects phosphorylation changes that require RORα. **f, g** Selected pathways enriched in proteins exhibiting *RORα*-dependent cholesterol response identified using Panther and Reactome. See also Supplementary Data 5. **h** RORα-dependent phosphorylation response of TOR and insulin related proteins in response to re-stimulation of cholesterol within 1 or 6 h, shown as fold change relative to the cholesterol-depleted condition within each genotype. **i** Model proposed for the activation of TOR signaling mediated by HR3 and RORα in response to cholesterol. Dashed line indicates a possible non-lysosomal pool of cholesterol activating HR3/ RORα. Statistics: (**b** and **c**) Representative of two blots. **d, e** Changes greater than 30% up or down with $p < 0.05$ were considered significant. *P*-values were determined using the two-sided ANOVA test with Benjamini-Hochberg FDR procedure in Proteome Discoverer. **f, g** Pathway analysis with two-sided Fisher's *t*-test; only pathways with $p < 0.05$ are included. Source data and raw blot scans are provided in the Source Data file.

---

as through processes that require new protein synthesis and could involve a transcriptional response.

We then assessed whether HR3 is integral to both processes by examining the pS6 response to cholesterol in fat bodies in which HR3 expression had been silenced. Fat bodies lacking *HR3* exhibited no upregulation of pS6 in response to ex-vivo cholesterol stimulation (Fig. 6g), indicating that HR3 is essential for TOR-pathway activation by cholesterol. This demonstrates that cholesterol activates the TOR pathway through HR3 via tissue-autonomous mechanisms that may involve transcriptional regulation as well as non-genomic actions, such as interacting with signaling proteins upstream of TOR, such as the Rag GTPases. We therefore assessed the intracellular localization of HR3 and discovered that this nuclear receptor is not restricted to the nucleus, but also exhibits a cytosolic or membrane localization pattern (Fig. 6h and Supplementary Fig. 6h). This observation supports the possibility that HR3 may interact with components of the TOR pathway in the cytosol or at the surface of lysosomes. To further this, we explored whether inhibiting the Ragulator-Rag complex would negate the effects of HR3(LBD) overexpression. The TOR activity increase due to *HR3(LBD)* overexpression was completely abolished by *RagA-B* knockdown, whereas *RagC-D* knockdown only diminished the response, indicating that HR3-induced TOR activation is mainly mediated by RagA-B (Fig. 6e). We then induced lysosomal cholesterol accumulation by depleting *Npc1a*, which resulted in TOR activation. By simultaneously reducing *HR3*, we observed a decrease in the hyper-activation of TOR caused by *Npc1a* knockdown (Fig. 6i, j and Supplementary Fig. 6i). This indicates that HR3 is required for TOR activation in response to cholesterol accumulation within lysosomes. Notably, loss of *NPC1* has also been shown to activate TOR in mammalian cells, indicating that lysosomal cholesterol accumulation represents a conserved trigger for TOR activation across species.

To further refine how HR3 interacts with this conserved lysosomal cholesterol-TOR axis, we tested whether the highly conserved protein LYCHOS (known as Anchor in *Drosophila*) contributes to cholesterol-induced TOR activation. In mammalian cells, LYCHOS is thought to be a lysosomal transmembrane protein and has been implicated in promoting cholesterol-dependent TOR activation at lysosomes. LYCHOS is proposed to bind cholesterol at an intramembrane site between its permease-like and GPCR-like domains and, via this interaction, to engage GATOR and relieve its inhibition of the Rag GTPases[14,42]. While transmembrane proteins will make nonspecific hydrophobic contacts with membrane cholesterol, nuclear receptors such as HR3 instead bind ligands with high specificity in buried ligand-binding pockets within their LBDs, and ligand occupancy directly modulates their activity[43]. Despite the strong sequence conservation and one-to-one orthology between *Drosophila* Anchor and mammalian LYCHOS, we were surprised to find that fat-body knockdown of *LYCHOS* did not reduce cholesterol-induced TOR activation (Supplementary Fig. 7a, b). A second independent RNAi line targeting *LYCHOS* produced

comparable cholesterol-induced pS6 responses (Supplementary Fig. 7b, c), further indicating that LYCHOS is dispensable for cholesterol-stimulated TOR activation in the fat body. These findings indicate that, in *Drosophila*, cholesterol activates TOR independently of LYCHOS/Anchor.

We next asked whether GATOR itself is required for cholesterol-induced TOR activation in the fat body. As expected for a negative regulator of the Rag GTPases, knockdown of the GATOR component *Iml1* increased basal pS6 levels in the fat body of cholesterol-depleted animals (Supplementary Fig. 7d). However, cholesterol refeeding elicited similar relative increases in pS6 in control and *Iml1-RNAi* animals, indicating that the cholesterol response does not depend on GATOR activity. Strikingly, simultaneous knockdown of *HR3* strongly attenuated both the elevated basal TOR activity and the cholesterol-induced TOR activation caused by *Iml1* depletion (Supplementary Fig. 7d), showing that HR3 is required for TOR activation even when GATOR1-mediated inhibition is relieved. Although lysosomal cholesterol accumulation associated with *NPC1* loss activates TOR in both *Drosophila* and mammals, the LYCHOS–GATOR module is dispensable for this process in flies. Together, our data identify HR3 as a key cholesterol-responsive component of this conserved axis, linking cholesterol availability to TOR-pathway activation by providing input upstream of the Rag GTPases.

## Human RORα activates the TOR pathway in response to exogenous cholesterol

Considering the functional and sequence conservation between human RORα and its *Drosophila* counterpart, HR3, we investigated the potential role of RORα in modulating mammalian TOR-pathway activity in response to cholesterol. We used human Karpas-707H cells, a multiple myeloma cancer cell line strongly expressing RORα, to assess TOR signaling in response to exogenous cholesterol. In-vitro cholesterol depletion (0.75% MCD for 2 h) of these cells reduced TOR signaling, reflected in decreased pS6 abundance, whereas cholesterol repletion (complexed with 0.1% MCD) enhanced pS6 levels dose-dependently (Fig. 7a, b and Supplementary Fig. 8a, b), establishing a direct correlation between extracellular cholesterol levels and TOR-pathway activation. *RORα* knockdown using synthetic siRNA attenuated the pS6 response to cholesterol (Fig. 7c and Supplementary Fig. 8c), mirroring the *HR3*-knockdown effect observed in *Drosophila* (Fig. 5a, b).

For more comprehensive insights into RORα-mediated TOR-pathway activation by cholesterol, we performed phosphoproteomic analysis of Karpas-707H cells. Comparisons were made against cholesterol-depleted cells, considering a 30% change in phosphorylation with a *p*-value < 0.05 to be significant. Within 1 hour of cholesterol replenishment, 100 significant RORα-dependent protein phosphorylation changes occurred, increasing to 145 changes after 6 hours (Fig. 7d, e and Supplementary Data 5). Pathway analysis showed a

pronounced enrichment in components of the TOR and PI3K/AKT pathways among the RORα-dependent cholesterol-induced phosphorylation changes (Fig. 7f, g). Proteins that were (de)phosphorylated in response to cholesterol in a RORα-dependent manner included S6 and LAMTOR1 (Fig. 7h), aligning with our findings in the fly (Fig. 5i, j), along with other key components of the TOR and insulin-signaling pathways including the TOR inhibitor TSC2, proteins of the KICSTOR complex required for proper Rag modulation by nutrients, SIK3, the direct TOR kinase target 4EBP1, and the insulin/PI3K/PIP3 mediator PKB/AKT. These in-vitro findings also reinforce our in-vivo observations, suggesting cell-autonomous modulation of TOR activity by RORα/HR3.

In results similar to those seen with *HR3* knockdown in the fly, 61 proteins were aberrantly responsive in terms of phosphorylation after 1 h when *RORα* was silenced, indicating that RORα normally restrains these responses (Fig. 7d, e). Following 6 hours of cholesterol stimulation, a larger array of proteins (318) showed cholesterol-induced phosphorylation changes only in RORα-deficient cells. This indicates the necessity of RORα/HR3 in maintaining homeostatic cellular responses to cholesterol over time. Collectively, our findings suggest that RORα and HR3 are essential for translating cholesterol abundance into TOR-pathway activation that governs cellular growth and metabolism (Fig. 7i).

## Discussion

Organisms and cells must finely tune their growth in response to environmental fluctuations[11,27]. Since cholesterol is essential for cellular growth, the availability of cholesterol must be intimately linked with the activity of pathways that control growth. The TOR pathway is a primary growth-regulatory mechanism by which growth is adapted to nutritional cues. Our findings delineate dynamic regulation of the TOR pathway by varying cholesterol levels in *Drosophila* tissues, linking cholesterol availability with growth responses. This dynamic response requires a mechanism that accurately senses cholesterol levels and directly influences the activation of the TOR pathway. Our work suggests the nuclear receptor HR3 in *Drosophila* and its human ortholog RORα mediate this cholesterol-sensing step and translate sterol abundance into growth-regulating TOR activity. Cholesterol is a natural ligand of human RORα[33,44], and our findings now show that cholesterol also acts as a bona fide ligand of its *Drosophila* ortholog, HR3. This represents a significant advance in understanding the sensing mechanism that links cholesterol availability to TOR and thus to cell growth.

Beyond showing that HR3 couples cholesterol levels to TOR regulation and to growth, our work also provides potential mechanistic links between steroid-regulated processes and nutrient-sensing pathways that coordinate growth during development[23,45,46]. In *Drosophila*, the interplay between the TOR/insulin and steroid (ecdysone) signaling pathways controls growth. Ecdysone antagonizes the growth of most larval tissues, but how it interacts with growth-regulatory pathways such as TOR has been an open question. *HR3* expression is induced by ecdysone[17], and our work thus suggests a complex interplay in which ecdysone influences HR3 expression, which in turn adjusts growth according to cholesterol levels via TOR signaling. Furthermore, both HR3 and TOR regulate the biosynthesis of ecdysone[36,38,47], and we recently provided evidence suggesting that cholesterol sensing in the PG is also linked to ecdysone production[11]. The dual function of HR3 in detecting levels of cholesterol – the precursor for ecdysone synthesis – and in stimulating TOR activity in the endocrine cells of the PG that produce ecdysone suggests that HR3 may directly link cholesterol sensing to steroid-hormone production, which governs both juvenile growth and the onset of maturation in animals.

In insects, several dietary sterols besides cholesterol can be converted into functional ecdysteroids equivalent to ecdysone and its more active form, 20-hydroxyecdysone (20E)[48]. Our data argue that HR3-mediated TOR activation reflects a direct sterol-sensing route rather than an ecdysone-driven effect, because (i) the cholesterol response is rapid and partly cycloheximide-insensitive, and (ii) loss of ecdysone signaling following *EcR* knockdown or blockade of ecdysone biosynthesis did not reduce cholesterol-mediated TOR-pathway activation. However, we cannot exclude that HR3 binds a broader set of dietary sterols that can be converted into ecdysteroids. Mammalian receptor RORα activity can be modulated by multiple sterols, including 7-dehydrocholesterol and cholesterol sulfate[32]. 7-dehydrocholesterol is a key sterol intermediate in the ecdysone biosynthetic pathway, raising the possibility that HR3 might also sense this precursor in the PG. Determining the ligand specificity of HR3, including sterols beyond cholesterol, will clarify whether HR3 is cholesterol-specific or a more general sterol sensor, which remains an interesting question for future studies. Because our current experiments were performed in larvae, where growth and biosynthetic activity are high, it will be important to determine whether similar HR3-sterol interactions occur in adults, where metabolic priorities differ. Likewise, whether HR3 responds to sterols in tissues other than the fat body, such as the gut or gonads, remains an open and intriguing question.

HR3 function is also regulated by the nuclear receptor E75, which physically interacts with HR3 and represses its transcriptional activity. This interaction is regulated by the metabolic state of the PG – E75 is a heme-binding protein, and the oxidation state and gas binding of the iron ion govern E75:HR3 binding[49]. Interestingly, this regulatory nuclear-receptor pair not only operates in the fly but is conserved in humans, where their orthologs RORα and Rev-erb respond to the same respective ligands, cholesterol and heme/NO/redox[50–52]. Thus, these receptors may integrate different signaling cues into networks coordinating cellular growth processes in response to intracellular cholesterol levels.

Our results corroborate earlier findings that HR3 genetically interacts with S6 kinase to modulate cell growth[53] and integrate cholesterol sensing into this growth-regulatory system. This regulation is mediated by an HR3 isoform lacking a DNA-binding domain, suggesting possible non-genomic effects. However, cholesterol likely binds to both isoforms of HR3, with and without the DBD.

Indeed, our findings indicate that full-length HR3 exerts repressive functions that depend on its DNA-binding domain and are further enhanced by the presence of the ligand-binding domain. This suggests that cholesterol-dependent activation of full-length HR3 engages transcriptional programs that antagonize or buffer TOR activation. This interpretation is supported by our RNA-seq and phosphoproteomic data, which show that many genes and phosphosites become aberrantly more responsive to cholesterol when HR3 is absent. Taken together, these findings support a model in which HR3 fulfills a dual regulatory role: HR3 promotes TOR activation through rapid, LBD-dependent non-genomic signaling while simultaneously initiating transcriptional feedback programs that restrain, shape, and stabilize the longer-term TOR response. This combined activating and feedback-limiting architecture provides a mechanistic explanation for how cells achieve both sensitivity and robustness in cholesterol-dependent TOR regulation.

Furthermore, these HR3 isoforms likely have overlapping functions, and could conceivably interact with E75 or other transcriptional regulators. Nuclear receptors often mediate effects via both transcriptional and non-genomic pathways, the latter involving direct interactions with signal-transduction pathway components at cytosolic or membrane locations[41]. We present evidence that cholesterol-induced TOR-pathway activation requires both transcriptional and non-transcriptional regulation and depends upon HR3. In humans, RORα regulates Wnt signaling through a non-genomic mechanism that involves Protein Kinase C α (PKCα) phosphorylation[54,55]. RORα is

present in both the nucleus and cytosol, as well as in membrane domains rich in cholesterol. This suggests a potential for direct interaction of membrane-associated RORα with membrane cholesterol. An isoform of RORα lacking a DNA-binding domain has not yet been identified, leaving it an open question whether mammalian cells can produce an isoform similar to HR3 (LBD). By acting via both transcriptional and non-transcriptional routes, HR3 and RORα may precisely regulate cholesterol-mediated growth responses. Intracellular signaling often involves counteracting feedback mechanisms to maintain homeostasis. Positive feedback amplifies signals, while negative feedback modulates these responses, preventing overactivation and allowing the cell to revert to its baseline state. Our results indicate that fat-tissue cells with increased expression of an HR3 isoform lacking a DBD, when stimulated with cholesterol, develop high levels of intercellular variation in pS6, with some cells showing very high TOR pathway activity while neighboring cells exhibit low signaling. This suggests a loss of these cells' ability to control the TOR pathway. An intriguing possibility is that HR3 modulates the TOR pathway through both positive and negative feedback mechanisms that involve rapid non-genomic responses and longer-term transcriptional regulation to maintain balanced cholesterol responses. Recent research has implicated LYCHOS and SLC38A9 in the activation of the TOR pathway by lysosomal cholesterol[3,14]. These factors act upstream of the Rag GTPases but are not reported to be involved in any negative-feedback regulation. Our findings indicate the additional necessity of HR3/RORα for precise modulation of TOR activity, critical not only for immediate responses to cholesterol but also for sustained pathway equilibrium. Thus, HR3/RORα may act as a gatekeeper, normally ensuring that while cholesterol effectively stimulates growth via TOR, it does not induce overactivation that could lead to pathological hyperactivation which could lead to tumorigenesis.

Our study may thus also shed light on the link between high cholesterol and cancer development and progression. Extensive studies have reported an association between high levels of cholesterol in the blood and an increased risk of various cancers[7,56,57]. Although the exact molecular mechanisms connecting elevated cholesterol to cancerous cell proliferation remain elusive, it is known that the enhanced uptake of exogenous cholesterol by cells stimulates oncogenic processes and tumor growth[10]. Our findings show that in response to cellular uptake of exogenous cholesterol, RORα regulates numerous signaling pathways involved in cancer initiation and progression, such as IGF, PI3K/AKT, MAPK, ALK, and SIK3. SIK3 expression is highly elevated in the majority of breast cancer cases[58], which are typically characterized by their dependence on cholesterol[59]. This suggests that RORα and TOR signaling may mechanistically link cholesterol and tumor growth.

## Methods

### *Drosophila* media and husbandry
Flies were kept at 25 °C and 60% humidity with a 12-hour/12-hour daily light cycle. Fly stocks were maintained on standard fly food (STD-FF) medium containing 6% sucrose, 0.8% agar, 3.4% yeast, 8.2% cornmeal, 0.16% Tegosept antifungal agent, and 0.48% propionic acid preservative. Unless a different diet is indicated, this medium was also used for experiments. The fly lines used are listed in Supplementary Table 1, along with other salient reagents.

### Timed egg collections
Virgin females from the driver stocks were mated with males of the RNAi lines in bottles containing STD-FF and additional dry-yeast pellets approximately two days prior to timed egg collections. The day before egg collection began, animals were transferred to egg-laying chambers with apple-juice/agar egg-collection plates supplemented with yeast paste. Egg-collection plates were changed on the following day for egg collection during a 4 h period (ZT1 to ZT5) at 25 °C and 60% humidity.

For genotypes not carrying *GAL80^{TS}*, hatched first-instar larvae were collected after 24 h at 25 °C and transferred to vials containing STD-FF or NutriFly (NF) at 30 larvae per vial containing 10 mL of medium. Offspring carrying *Tub-GAL80^{TS}* were collected after a 48-hour hatching period at 18 °C to prevent GAL4 activity during early larval development.

### Chronic high-cholesterol diet
Bloomington NutriFly (NF) medium (Genesee Scientific, "Bloomington formula", #66-113), which contains a relatively low yeast/sterol content, was cooked according to product guidelines with added 0.16% Tegosept antifungal agent. The medium was supplemented while liquid with cholesterol (from a 8 mg/mL stock of cholesterol in ethanol) to a final added concentration of 40 μg/mL (NF + 40) or 80 μg/mL (NF + 80). Control media were prepared by adding the same amount of 100% ethanol. First-instar larvae were picked from egg collections and developed in these media until experiments were conducted during the third instar.

### Synthetic-food re-feeding experiments: Cholesterol, protein, and ecdysone
The synthetic food (SF) diet is based on the one described in Reis, 2016[22], adjusted to increase its nutritional content and solidity. Casein was used at double the previous concentration, agar was increased to 1% concentration, sucrose was increased to 2.7-fold, and choline chloride was doubled. The final components of 1 L of food are 10 g lipid-depleted agar, 146.6 g lipid-depleted casein, 35.5 g sucrose, 0.64 g choline chloride, 0.85 g inosine, 0.76 g uridine, 133 mL of 10 g/L NaHCO₃ stock, 133 mL of a 37.3 g/L KH₂PO₄ stock), 133 mL of 7.1 g/L K₂HPO₄ stock, 133 mL of 6.2-g/L MgSO₄·7H₂O stock, 133 mL of Vitamin A stock, 13.3 mL of Vitamin B stock, 333 mL deionized water, 13.3 mL of 10% Tegosept in ethanol. Vitamin A stock contained 20 mg thiamine, 100 mg riboflavin, 120 mg nicotinic acid, 160 mg D-pantothenic acid hemicalcium salt, 25 mg pyridoxine, and 2 mg biotin in 1 L deionized water. Vitamin B stock was made of 500 mg folic acid dissolved in 167 mL 20% ethanol. Cholesterol content was adjusted by adding either pure ethanol or 8 mg/mL cholesterol in ethanol to reach final cholesterol concentrations of 0, 1.2, 6, 15, 25, 40, or 80 μg/mL. Casein and agar were lipid-depleted through six chloroform extractions of at least 6 hours each, with stirring, at room temperature, at a 1:5 ratio of solids to chloroform. After each extraction, solid material was collected by filtration (using Whatman paper in a ceramic funnel), and lipid-free material was dried after the last collection by allowing chloroform to evaporate under a fume hood.

Larvae from timed egg collections were developed on STD-FF till early third instar: genotypes without *GAL80^{TS}*, 25 °C until 84 h AEL; those with *GAL80^{TS}*, 18 °C until 120 h AEL. Larvae were then collected by floating in 20% sucrose solution, washed using DI water, and transferred using a paint brush or entomological forceps to synthetic media in a 24-well dish, each well stoppered with a foam plug. For cholesterol- and ecdysone-feeding experiments, animals were transferred at this time to media containing either 0 μg/mL or 1.2 μg/mL cholesterol (with other ingredients constant); for protein-feeding experiments, this medium lacked casein but did contain 80 μg/mL cholesterol. After 10-12 hours on this medium, the larvae were collected using DI water and transferred either to fresh nutrient-dropout medium (0 or 1.2 μg/mL cholesterol or 0 mg/mL protein) or to a re-feeding medium as indicated in each figure (containing 40 or 80 μg/mL cholesterol, 250 μg/mL 20-hydroxyecdysone +1.2 μg/mL cholesterol, or 14.7 mg/mL casein as appropriate) in new 24-well plates. After the indicated time intervals (15 min, 30 min, or 1, 2, 4, 6, or 10 h), the larvae were collected using water washes.

### Larval-mass measurements
First-instar larvae were collected from timed egg-lays, transferred to the indicated diets, and maintained at 25 °C. At 96 h AEL, larvae were

collected and rinsed with deionized water. Batches of larvae were chilled in DI water on ice, and individual animals were blotted dry on a KimWipe and weighed using a Sartorius SE2 Micro Balance.

### Transgene construction

To create the *UAS-HR3* (full-length) construct, the coding sequence of the HR3-PA isoform (FlyBase) was codon-optimized, synthesized as an *EcoRI/XbaI* fragment, cloned into the vector *pUAST-attB* (obtained from the University of Indiana Drosophila Genomics Resource Center, plasmid #1419, donated by J. Bischof and K. Basler[60]), and sequence-verified by GeneArt Services (ThermoFisher). A similar procedure was used to create the *UAS-HR3^{K243X}* DBD-only construct, in which a stop codon was substituted for lysine 243, mimicking the homozygous-lethal *HR3^{K243X}* allele[31]. The constructs were shipped to BestGene, Inc. (Chino Hills, California) for integration into the genome at the *VK00037 attP* site on chromosome arm 2 L. These constructs' DNA sequences are given in Supplementary Table 2.

### Transcript analysis by quantitative PCR

Total RNA was isolated using the NucleoSpin RNA kit (Macherey-Nagel, #740955) according to the manufacturer's instructions. For each genotype or condition, six independent biological replicates were prepared, and each replicate consisted of five whole animals. Samples were homogenized in 2 mL tubes containing RA1 lysis buffer supplemented with 1% β-mercaptoethanol using a TissueLyser LT bead mill (Qiagen) with 5-mm stainless-steel beads (Qiagen #69989). Complementary DNA was generated from purified RNA using the High-Capacity cDNA Reverse Transcription kit (Applied Biosystems, #4368814). Quantitative PCR was performed using RealQ Plus 2× SYBR Green Master Mix (Ampliqon, #A324402) on a QuantStudio 5 Real-Time PCR System (Applied Biosystems). Gene expression was normalized to the *Drosophila* reference gene *Rp49*. Primer sequences are provided in Supplementary Table 1.

### Promega Steady-Glo Luciferase assay

Three feeding late-third-instar *unkempt-Luciferase* larvae for each sample were collected into Glo Lysis Buffer (Promega, E2661) in 2 mL Eppendorf tubes. For normal assays, 200 μL of buffer was used per sample; for large animals or if protein quantification was performed in parallel, 400 μL was used. The samples were homogenized with a 5 mm steel bead in a TissueLyser bead mill (Qiagen; operated at 50 Hz for 30 s). The homogenized samples were incubated at room temperature for 10 minutes for complete cell lysis and then centrifuged at 13,000 x g for 5 min to pellet insoluble material. An aliquot (150 μL) of supernatant was removed to new Eppendorf tubes for protein quantification and placed on ice. For luciferase measurement, 20 μL of each sample was transferred at room temperature to an opaque white 96-well plate (Costar), and 20 μL of Steady-Glo Luciferase Reagent (Promega, E2510) was added to each well, with gentle mixing by pipetting. The plate was centrifuged at 1000 x g to settle the liquids and incubated at room temperature in the dark for 10 min to allow the luciferase reaction to reach a steady state, after which the luminescence was measured using an EnSight multi-mode plate reader (PerkinElmer).

### Protein quantification (Bicinchoninic Acid Method)

The protein concentrations in the aliquots set aside during the luciferase assay and for human western-blot protein-loading calculations were measured using a bicinchoninic acid (BCA)-based method. BCA working reagent was prepared by combining bicinchoninic acid solution (Sigma-Aldrich, #B9643) and 4% cupric sulfate solution (Sigma-Aldrich, #C2284) at a 50:1 ratio. Sample supernatant was diluted by adding of 3 volumes of PBS, and four microliters of the diluted material were pipetted into a 384-well plate. Thirty microliters of BCA working reagent was added to each well, with gentle mixing by pipette, and the

plate was centrifuged at 1000 x g to settle liquids and incubated for 30 min at 37 °C. The absorbance of each sample at 540 nm was measured using an EnSight multi-mode plate reader (PerkinElmer).

### Western blotting of *Drosophila* samples

Three to eight feeding (pre-wandering) late-third-instar larvae for each sample were lysed in ice-cold SDS sample buffer (Bio-Rad, 2x Laemmli Sample Buffer, #1610737), containing 5% beta-mercaptoethanol and protease and phosphatase inhibitors (Roche Complete Mini protease inhibitor, Sigma-Aldrich #11836153001, and Roche Complete Ultra phosphatase inhibitor, Sigma-Aldrich #05892970001), 60 μL per larva, using a TissueLyser bead mill (Qiagen). Samples were denatured at 95 °C for 5 min, and insoluble material was pelleted by centrifugation at 13,000 x g for 5 min. Samples were loaded into precast 4–20% gradient polyacrylamide gels (Bio-Rad, #4561094) and electrophoresed at 150 V for approximately 35 min. Proteins were transferred to 0.2 μm nitrocellulose membrane using a Trans-Blot Turbo Transfer Pack (Bio-Rad, #1704159) and the Bio-Rad dry-transfer apparatus. Membranes were then blocked in Intercept Blocking Buffer (LI-COR, #927-70001) for 1 hour at room temperature with gentle agitation. Phospho-S6K (pS6K) and histone H3 were detected by incubating with rabbit anti-pS6K (Cell Signaling #9209S, diluted 1:1000) and rabbit anti-histone-H3 (Abcam #1791, diluted 1:1000) in Odyssey blocking buffer (LI-COR) + 0.2% Tween-20 (Sigma-Aldrich, #P9416) overnight at 4 °C with gentle agitation. Membranes were washed three times with PBS + 0.1% Tween-20, and secondary staining was performed with IRDye 680RD-labeled goat anti-mouse and IRDye 800CW conjugated goat anti-rabbit (LI-COR, #925-68070 and #925-32210, each diluted 1:10,000 in Intercept Blocking Buffer + 0.2% Tween-20) for 45 min at room temperature with gentle agitation. Membranes were washed three times with PBS + 0.1% Tween-20, followed by a single wash with PBS. Bands were visualized using an Odyssey Fc gel reader (LI-COR) and quantified using the LI-COR Image Studio Gel Reader program.

### Immunostaining, microscopy, and quantification

Larval tissue was dissected in PBS and fixed in freshly prepared 4% paraformaldehyde (EM grade) in PBS at room temperature: for fat-body tissue, 10–20 larvae were inverted and fixed at room temperature for 45 min, whereas for prothoracic-gland (PG) samples, tissues were accumulated in ice-cold 4% PFA during dissection and fixed at room temperature for 70 min. Samples were quickly rinsed in PBST (PBS + 0.1% Triton X-100) and washed three times for 15 min in PBST. The samples were then blocked in PBST + 3% normal goat serum (Sigma) for at least 30 min at room temperature with gentle agitation. Tissues were incubated with primary antibodies diluted in PBST + 3% normal goat serum overnight at 4 °C with gentle agitation. Tissue was then washed three times in PBST. Samples were incubated with secondary antibodies diluted in PBST at 4 °C, in the dark, overnight with gentle agitation. For actin staining, samples were incubated with phalloidin (Alexa Fluor 647 conjugate, ThermoFisher #A22287, or Alexa Fluor 488 conjugate, #A12379, diluted 1:100 in PBST for 1 h), and for nuclear staining, samples were incubated in DAPI (1:500 in PBS, ThermoFisher, #62248) for 0.5–1 h. Samples were washed twice in PBST and once in PBS and mounted on glass slides coated twice with poly-L-lysine (Sigma-Aldrich, #P8920-100ML). The mounted tissue was imaged using a Zeiss LSM 900 confocal microscope using a ×20 objective (NA 0.8). All samples compared within a given figure panel were processed similarly (fixation, staining preparations and imaging hardware settings). In all cases, each experimental sample was prepared simultaneously with its respective control using the same reagent preparations. Data sets from multiple preparations are normalized to controls before comparison.

For quantification of PG cell size and pS6 intensity, a composite of channels was created, and individual cells and their nuclei (based on actin and DAPI staining) were traced at the center of the cell thickness

in the Z-stack, saving this to the region-of-interest (ROI) manager in the image-analysis package FIJI[61]. The Raw Integrated Density (RID) of staining in each channel was quantified within each ROI. The pS6 intensity in the cytoplasm was calculated as (Cell_RID - Nuclear_RID) / (Cell_Area - Nuclear_Area).

For quantification of fat-body pS6 staining, a large number of tissues were mounted on each slide. Using a ×5 overview scan in the DAPI channel on the Zeiss LSM 900 confocal microscope, areas of flat, single-cell-thick tissue were selected for each large piece of fat body and were imaged with the ×20 objective with a 1.5 μm Z spacing. Z-stacks were pre-processed in FIJI using a macro for creating a composite image and Z-projected using "sum". Areas of folded or overlaid tissues were removed by manual segmentation; the channels were split, and a binary mask was created from the DAPI channel. This mask was used to remove the nucleus from the pS6 channel, since nuclear pS6 staining is believed to be spurious[26]. Quantification was automated using CellProfiler, version 3.1.9[62]: each cell's pS6 signal was quantified by segmentation of the tissue into individual cells using the actin (phalloidin staining) channel and the binary mask created using DAPI in order to produce a cytoplasmic pS6 signal. The pS6 signal was measured as the average intensity across each area ("integrated intensity" divided by the cytoplasmic area).

Primary antibodies used were rabbit anti-pS6[26] against the epitope RRR(phospho-S)A(phospho-S)IRE(phospho-S)K, used at 1:500; mouse anti-GFP (clone 3E6, ThermoFisher #A11120, RRID AB:221568, at 1:500); guinea-pig anti-Shroud[63] (a kind gift from R. Niwa, University of Tsukuba; 1:200); and rabbit anti-HR3[31], a generous gift from J. Montagne (Friedrich Miescher Institute for Biomedical Research, Switzerland, 1:250). Secondary antibodies (ThermoFisher, 1:500) were all raised in goats and were cross-adsorbed by the manufacturer to reduce off-target binding. These included anti-rabbit, Alexa Fluor 488 conjugate (#A32731, RRID AB_2633280); anti-rabbit, Alexa Fluor 555 conjugate (#A32732, RRID AB_2633281); anti-rabbit, Alexa Fluor 647 conjugate (#A32733, RRID AB_2633282); anti-guinea-pig, Alexa Fluor 647 conjugate (#A21450, RRID AB_2735091); and anti-mouse, Alexa Fluor 488 conjugate (#A32723, RRID AB_2633275).

### Ex-vivo culture of *Drosophila* fat body

Larvae from a four-hour egg collection were raised on STD-FF at 25 °C. At 96 h AEL, fat-body tissue was dissected in Schneider's insect medium (Gibco, #21720). Tissue was cholesterol-depleted using Schneider's medium containing 10% lipid-depleted-serum (LDS, Sigma-Aldrich, #S5394-50mL) and 0.75% methyl-β-cyclodextrin (MCD, Sigma-Aldrich, #C4555), for 2 hours at room temperature with gentle rocking in a glass staining dish with glass cover (Assistant, #42020010). For experiments including cycloheximide, 15 min prior to cholesterol stimulation, cholesterol-depletion medium was replaced with fresh cholesterol-depletion medium containing 100 μg/mL cycloheximide (Sigma-Aldrich, #239765) from a 100 mg/mL stock solution in DMSO. Tissue was washed twice with room temperature Schneider's medium and then incubated in cholesterol-depletion or cholesterol-stimulation medium for 1 h. Conditions including cycloheximide also contained it during the stimulation period. Cholesterol-stimulation medium consisted of Schneider's medium containing 10% LDS, 0.2% MCD, and 100 μM (39 μg/mL) cholesterol from a 20 mg/mL cholesterol stock in ethanol. After incubations, media were replaced by 4% PFA for 45 min tissue fixation at room temperature. Fixed tissues were then processed for immunostaining as usual.

### Human cell culture

The Karpas 707H cell line was established from the bone marrow of a 53 year old male with multiple myeloma[64]. Cells were obtained from Cambridge University (Cambridge Enterprise Limited) and cultured in StableCell RPMI-1640 culture medium (Sigma-Aldrich, #R2405-500ML) containing 15% fetal bovine serum (Gibco, #10500-064) and

Pen-Strep antibiotic mixture (100 u/mL Penicillin, 100 μg/mL Streptomycin, Gibco, #15140-122). The cultures were maintained at 37 °C and 5% CO$_2$. Karpas cells partly attach to plasticware after 3-4 days of culturing, while some cells remain in suspension. To prevent selective pressure during passaging (cells can be partly released with tapping on cultureware), Karpas cells were passaged by collecting cells in suspension and using Trypsin-EDTA (Gibco, #25200056) to fully float attached cells.

### Cell-culture cholesterol starvation/re-stimulation

The methods used are similar to those described in Shin et al.[14] and Castellano et al.[3] Cells were cholesterol-depleted using cell-culture media supplemented with methyl-β-cyclodextrin (MCD, Sigma-Aldrich, #C4555) and lipid-depleted serum (LDS; lipoprotein-deficient serum from fetal calf, Sigma-Aldrich, #S5394-50mL). Cells were re-stimulated with cholesterol using cell-culture medium supplemented with MCD: cholesterol complex and LDS.

Cells were plated in standard growth medium, RPMI-1640 medium containing 15% fetal bovine serum and Pen-Strep, in 6-well plates (Greiner bio-one, #657160) and allowed to attach for 3-4 days. Cells were then rinsed with serum-free RPMI-1640 medium and then incubated with 0.75% MCD with 0.5% LDS in RMPI-1640 (without antibiotics) for 2 h in order to cholesterol-deplete the cells. Cells were re-stimulated with cholesterol in RPMI-1640 using complexed MCD: cholesterol (1x: 0.1% MCD and 50 μM/ ~ 20 μg/mL cholesterol: 2x dose: 0.2% MCD and 100 μM cholesterol) plus 0.5% LDS. MCD: cholesterol complex was prepared by diluting a 20 mg/mL cholesterol stock in ethanol into a 15 mL Falcon tube containing RMPI-1640, 0.1% MCD, and 0.5% LDS, resulting in a 50 uM final concentration of both cholesterol and MCD. The tube was then vortexed and incubated in a 37 °C water bath for 2 hours before being applied to the cells.

### Knockdown of *RORα* using siRNA

Knockdown of *RORα* in human cells was performed using the siGenome SMARTpool Human *RORA* (#6095) siRNA from the Horizon Discovery "Dharmacon" portfolio (#M-003440-01-005) and Lipofectamine 2000 (Invitrogen, #11668-030). The siRNA stock was diluted to 20 μM in RNase-free water following the product protocols. A siRNA duplex targeting *Renilla Luciferase* (a gene not present in these cells) was used as a control. siRNA and Lipofectamine solutions were prepared according to product specifications. Cells were rinsed with serum-free RPMI-1640 medium before being transfected with 5 μL of siRNA pool (20 μM) and 5 μL Lipofectamine 2000 in a 6-well plate in 1 mL of antibiotic-free Opti-MEM (Gibco, #31985-062) medium plus 2 mL antibiotic-free cell-culture medium. For experiments other than phosphoproteomics, the medium was changed 24 h after treatment to new RPMI-1640 with 15% fetal bovine serum (without antibiotics). Seventy-two hours after knock-down treatment, cells were processed for further experiments (cholesterol treatments or collection of samples). Samples for phosphoproteomics were exposed to the siRNA treatment for the entire 72 h incubation, with additional siRNA and culture medium being added after 24 h for a stronger knock-down.

### Human-cell lysis and immunoblotting

Cells in suspension were collected using RPMI-1640 and centrifuged for 4 min at 500 x *g*, and the supernatant was removed. Lysis buffer was added to still-attached cells in the culture flask and to the pelleted suspension quickly after medium was removed. Cells were lysed in ice-cold RIPA buffer (50 mM Tris-HCl pH 7.5, 150 mM sodium chloride, 0.5% sodium deoxycholate, 1% Nonidet P-40, 0.1% SDS) supplemented with Roche Complete Protease Inhibitor Cocktail with EDTA (Roche, #11836153001), PhosStop (used at 2x, Roche, #PHOSS-RO), 11 mg/mL β-glycerophosphate (from a 110-mg/mL 10x stock solution), ~100 mM sodium fluoride (a saturated solution in RIPA buffer, roughly 1 M, was made as a 10x stock solution), and Benzonase endonuclease (Sigma-

Aldrich, #G9422). Cells were transferred to pre-cooled Eppendorf tubes on ice. The samples were incubated on ice for 10 min after which they were spun down to pellet nuclei and membrane at maximum speed (14,000 rpm) at 4 °C for 15 min. An aliquot of supernatant was reserved for BCA protein measurement. For Western-blot samples, 230 μL of supernatant was mixed with 70 μL 5x Laemmli buffer (4% SDS, 10% β-mercaptoethanol, 20% glycerol, 0.004% bromophenol blue, 0.125 M Tris-HCl, pH 6.8) or 230 μL 2x Laemmli Sample Buffer (Bio-Rad, #1610737), and samples were immediately denatured for 5 min at 95 °C.

Loaded sample sizes were adjusted to equalize total protein based on BCA protein measurements. Protein separation was done using electrophoresis in polyacrylamide gels using 1x Running Buffer (10x running buffer: Tris base 30.2 g, glycine 188 g, 10% SDS solution 100 mL, in deionized water to 1 L) using a BioRad Mini-PROTEAN Tetra Vertical Electrophoresis Cell system. Protein mass ladder PageRuler Plus Prestained Ladder (Thermo Scientific, #26615) or Chameleon Duo Pre-stained Protein Ladder (LI-COR, #928-60000) was used. The gel in Fig. 7b was run on a 12% Mini-Protean TGX precast protein gel (Bio-Rad, #4561034), whereas the one in Fig. 7c was run using hand-cast gels. Gels were run at 20 mA per gel at max voltage (300 V) for approximately 1 h. The blot in Fig. 7b was transferred using semi-dry transfer as described in the *Drosophila* western-blot section, whereas Fig. 7c was transferred to 0.2 μm nitrocellulose membrane (Amersham Proteon Nitrocellulose blotting membrane) using wet transfer: the blotting was performed using 1x Running Buffer with 20% methanol, kept cold with ice packs, run at 100 V for 1 h. Ponceau stain was used to visualize the transferred proteins.

Membranes were blocked for 1 h at room temperature using PBS + 5% nonfat milk powder + 0.1% Tween-20 and stained overnight at 4 °C in primary-antibody mix (diluted in PBS + 0.1% Tween-20 + 5% bovine serum albumin). Rabbit anti-pS6 (Cell Signaling, #4857) was used at 1:1000, mouse anti-S6 (Cell Signaling, #2317) was used at 1:1000, and mouse anti-α-Tubulin (University of Iowa Developmental Studies Hybridoma Bank clone #AA4.3) was diluted 1:5000, rabbit anti-RORα (human) (Cell Signaling Technology, clone E6G51, #34639S) was used at 1:1000, and rabbit anti-human-GAPDH (Cell Signaling Technology, clone 14C10, #2118 L) was used at 1:2500. Membranes were washed three times for 15 minutes in PBS + 0.1% Tween-20 at room temperature before secondary staining. Secondaries (goat anti-rabbit IgG H + L, HRP conjugate (ThermoFisher, #31466, or Jackson ImmunoResearch, #111-035-003) and goat anti-mouse IgG H + L, HRP conjugate (Jackson ImmunoResearch, #115-035-003) were diluted 1:10,000 in PBS + + 5% nonfat milk powder + 0.1% Tween-20 at 1:10,000 and incubated on the blot for 1-2 h at room temperature. Stain was detected using SuperSignal West Femto Maximum Sensitivity Substrate (Thermo Scientific, #34094) or ECL Prime Western Blotting Detection Reagents (Amersham, #RPN2232) chemiluminescence using a Bio-Rad ChemiDoc Touch Imaging System or an Odyssey Fc gel reader (LI-COR).

## RNA sequencing

*TOR* or *HR3* were knocked down in *Drosophila* larvae using the temperature-inducible ubiquitous driver *Tub-GAL80^TS; Tub-GAL4* (*Tub^TS >*). An 8 h egg collection was made at 18 °C, and egg-laying plates were maintained at 18 °C (to prevent activation of the GAL4 system) for an additional 44 hours before first-instar larvae were collected using a metal probe into vials containing standard food (30 larvae per vial). The collected larvae were maintained at 18 °C for a further 72 h before being transferred to 29 °C to induce RNAi expression (at 120 h AEL). After nine hours of RNAi induction on standard food (at 129 h AEL), larvae were transferred to low-cholesterol (1.2 μg/mL) synthetic medium and incubated at 29 °C for 11 h more (until 140 h AEL). Larvae were then transferred to synthetic food containing either 80 μg/mL cholesterol (for replenishment) or 1.2 μg/mL (control) and

incubated at 29 °C for 10 h. At this time (150 h AEL), five feeding late-third-instar larvae were collected for each of five replicates into ice-cold RLT buffer (Qiagen RNeasy Mini kit, #74104) containing 1% β-mercaptoethanol and homogenized using a bead mill (Qiagen TissueLyser LT). RNA was purified using the Qiagen RNeasy Mini kit with DNase (Macherey-Nagel, #740955.250) treatment. Samples were shipped on dry ice to Novogene Europe (UK), where RNA sequencing and bioinformatic analyses were performed. Differentially expressed genes were identified using DESeq2. *P*-values were calculated using Wald tests (two-sided) and adjusted for multiple testing using the Benjamini-Hochberg procedure.

## Phosphoproteomics

*Drosophila:* Cholesterol *vs.* protein refeeding: Refeeding experiments were conducted as described above in the *Drosophila* refeeding method section. Fifteen (cholesterol assays) or twenty (protein assays) feeding late-third-instar larvae for each of two replicates were collected by washing with water, blotted dry on a KimWipe, and snap-frozen in Eppendorf tubes on dry ice. A sample was collected from the nutrient-deprivation condition at the start of refeeding as time-point 0, and further samples were collected at 15 and 30 min and at 1, 2, 4, 6, and 10 h after the start of refeeding. Samples were stored at -80 °C until they were processed. *HR3* and *TOR* knock-down: *TOR* and *HR3* were knocked down in *Drosophila* larvae using the temperature-inducible ubiquitous driver *Tub-GAL80^TS; Tub-GAL4*. After an 8 h egg collection at 18 °C, the plates were maintained at 18 °C for an additional 40 h before first-instar larvae were transferred to vials of STD-FF. These were maintained at 18 °C for an additional 72 h before being transferred to 29 °C to induce RNAi at 120 h AEL. Nine hours later, at 129 h AEL, the animals were transferred to low-cholesterol synthetic medium (1.2 μg/mL cholesterol) and incubated for 11 further hours at 29 °C before being transferred (at 150 h AEL) to fresh synthetic food containing either 1.2 (continued low-cholesterol exposure) or 80 (replenishment) μg/mL cholesterol, still at 29 °C. After one hour and six hours, two samples of 15 larvae each for each treatment were collected, washed with DI water, blotted dry, and snap-frozen on dry ice before storage at -80 °C.

Karpas-707H: Cells treated for knockdown of *RORα* or *Luciferase* (mock knockdown control) were grown in 6-well plates till 80–90% confluence. Cells were cholesterol-depleted with MCD for 2 hours and washed twice with RPMI. Cholesterol-depletion or cholesterol-stimulation medium was added, and both treatments were sampled in triplicate after one hour; an additional triplicate sample of cholesterol-stimulated cells was taken at 6 h post-stimulation. All samples were washed once in RPMI. Cells in suspension were collected in a Falcon tube, and cells attached to the cultureware were gently collected in RPMI medium using a cell scraper and added to the suspended cells. Cells were pelleted at 900 RCF for 4 min, supernatant was removed, and cells were frozen on dry ice and stored at -80 °C.

Proteins from larval samples were extracted in 300 μL 5% sodium deoxycholate (SDC) in 50 mM HEPES (pH 8.5) containing Roche cOmplete protease inhibitor and PhosSTOP phosphatase inhibitors (Sigma) using a FastPrep-24 bead beater (MP Biomedicals) using 25–30 1.4 mm ceramic beads (OMNI international, US). Each tube was subjected to 3 × 45 s bead beating. After treatment the solution was diluted with an additional 300 μL of 50 mM HEPES (pH 8.5) and subjected to a second round of 3 × 45 s bead beating. The samples were centrifuged at 20,000 x *g*, and the supernatant was transferred to a low-binding 1.5 mL Eppendorf tube and subjected to probe sonication for 2 × 20 s at 60% amplitude. After sonication, the sample was denatured at 110 degrees for 5 min and centrifuged for 20 min at 20,000 x *g* to pellet insoluble material. The supernatant was transferred to another tube and the protein concentration was measured using a Nanodrop spectrophotometer. For the Karpas 707H cells, the proteins were extracted from the cell pellets in 300 μL 3% SDC in 50 mM HEPES, pH

8.5, containing cOmplete protease inhibitor and PhosSTOP phosphatase inhibitors (Sigma), using probe sonication for $2 \times 20$ s at 60% amplitude. After sonication, the sample was denatured at 110 degrees for 5 min and subsequently centrifuged for 20 min at 20,000 x $g$ to pellet insoluble material. The supernatant was transferred to another tube and the protein concentration was measured using a Nanodrop N60 spectrophotometer.

A total of 100 μg of protein was taken from each sample and subjected to reduction and alkylation using 10 mM DTT for 20 min followed by 20 mM iodoacetamide for 20 min. Trypsin (5%) was added and the solutions were incubated at 37 °C overnight. After incubation, a further 1% bolus of trypsin was added and the samples were incubated for one further hour at 37 °C. After incubation, the cleaved peptide solutions were labeled with tandem mass tags (TMTpro) 18-plex according to the manufacturer's protocols. After labeling, the 18 samples were combined into one sample containing all the larval peptides and one containing the Karpas 707H material. The SDC was removed by acidification and subsequent centrifugation for 20 minutes at 20,000 x $g$. The supernatant was transferred to a low-binding Eppendorf tube and dried until 150 μL was left. The enrichment of phosphopeptides using titanium dioxide and subsequent high-pH reversed-phase (RP) fractionation were performed as described[65]. In brief, the TMT-labeled peptide mix was incubated with TiO$_2$ beads (0.6 mg of beads per 0.1 mg of protein) in the presence of 5% TFA, 80% acetonitrile, and 1 M glycolic acid. The TiO$_2$ beads were removed by centrifugation, and the supernatant was incubated again with half the initial amount of beads. After incubation, the TiO$_2$ beads were pooled and washed with 200 μL of 1% TFA in 80% acetonitrile, followed by a wash with 100 μL of 0.2% TFA in 10% acetonitrile. The beads were dried, and phosphopeptides were eluted with 1% ammonium hydroxide (pH 11). Finally, the eluted phosphopeptides were dried down and resuspended in 20 mM ammonium formate (pH 9.4) prior to high-pH reversed-phase fractionation.

The phosphopeptide-enriched fraction from the larvae and Karpas 707H cells were analyzed by tandem mass spectrometry using an EASY nanoLC system coupled with a Fusion Lumos Tribrid or an Orbitrap Eclipse Tribrid. Lyophilized peptides from the high-pH RP fractionation (12–20 concatenated fractions) were re-solubilized in 3-5 μL of 0.1% formic acid (FA) and loaded onto a 20 cm analytical column (100 μm inner diameter) packed with ReproSil – Pur C18 AQ 1.9 μm RP material. The peptides were eluted with an organic solvent gradient from 100% phase A (0.1% FA) to 25% phase B (95% ACN, 0.1% FA) for 80-100 minutes (depending on fraction), then from 25% B to 40% B for 10–20 min before the column was washed with 95% B. The flow rate was set to 300 nL/minute during elution. For the two instruments the automatic gain-control target value of $1.5 \times 10^6$ ions in MS and a maximum fill time of 50 ms were used. Each MS scan was acquired at high resolution (120,000 full width half maximum (FWHM)) at m/z 200 in the Orbitrap with a mass range of 350–1400/1500 Da. The instruments were set to select as many precursor ions as possible in 3 s between the MS analyses. Peptide ions were selected from the MS for higher-energy collision-induced dissociation (HCD) fragmentation (collision energy: 34%). Fragment ions were detected in the Orbitrap at high resolution (50,000 FWHM) for a target value of $1.5 \times 10^5$ ions and a maximum injection time of 200 ms using an isolation window of 0.7 Da and a dynamic exclusion of 20–45 s. All raw data were viewed in Xcalibur v3.0 (ThermoFisher Scientific).

**Peptide/protein identification and quantitation.** All LC-MS/MS raw data files from *Drosophila*-larva experiments were searched in Proteome Discoverer (PD) version 2.5.0.400 (ThermoFisher Scientific). The raw data were searched in PD using the SEQUEST HT search algorithm against the Fly Database protein FASTA file. The searches had the following criteria: enzyme, trypsin; maximum missed cleavages, 2; fixed modifications, TMTpro (N-terminal), TMTpro (K) and

Carbamidomethyl (C). Variable modification for the phosphopeptides was Phospho (S/T/Y) and Deamidation (N) whereas no variable modifications were used for the "non-modified peptides". For the results from the Karpas 707H cells the raw data files were searched in PD using first an in-house Mascot server (Version 2.2.04, Matrix Science Ltd., London, UK) against the Swissprot human protein database using the following criteria: enzyme, trypsin; maximum missed cleavages, 2; fixed modifications, TMTpro (N-terminal), TMTpro (K) and Carbamidomethyl (C). Variable modification for the phosphopeptides was Phospho (S/T/Y). The peptide fragment ion spectra that were not identified with a peptide in Mascot with below 1% False Discovery Rate (FDR) was further subjected to database searching using SEQUEST HT in PD against the Human UniProt FASTA database, using the same criteria as above. The TMTpro reporter ion signals were quantified using S/N and they were normalized to the total peptide S/N in the PD program. The two-sided ANOVA test with Benjamini-Hochberg FDR procedure in PD was used to generate $p$-values for all the phosphopeptides and proteins identified in the database searches.

## Data filtering and bioinformatics analysis of phosphoproteomics and RNAseq

Phosphoproteomics data for pathway analysis was filtered for differences >30% in magnitude and $p < 0.05$ between cholesterol starvation and replenishment within each genotype. An Excel macro was used to identify populations of interest. For the RNAseq experiments, gene-expression levels are reported as FPKM (fragments per kilobase of transcript per million bases sequenced), which takes into account both sequencing depth and gene length. Differential expression analysis was conducted by Novogene through read-count normalization, model-dependent $p$-value estimation using a Wald test, and false-discovery-rate estimation using the Benjamini-Hochberg method. Differential expression was further analyzed by comparing genotype-based differences between sets of genes exhibiting cholesterol-induced expression change, resulting in seven categories of genotype-specific cholesterol-induced regulation (with set sizes from Fig. 5d for illustration): (1) genes requiring both HR3 and TOR to exhibit a cholesterol-induced change (occurring only in the control genotype) – 980; (2) those cholesterol-induced changes that require HR3 but are independent of TOR (occurring in controls and *TOR* knockdowns) – 452; (3) cholesterol-induced changes requiring TOR but independent of HR3 (occurring in controls and *HR3* knockdowns) – 374; (4) those changes requiring neither HR3 nor TOR (occurring in all samples) – 267; (5) cholesterol-induced changes repressed by HR3 but not by TOR – novel changes occurring only in *HR3-RNAi* samples – 1029; (6) cholesterol-induced changes repressed by TOR but not by HR3 – novel changes occurring only in *TOR-RNAi* – 644; (7) cholesterol-induced changes repressible by either HR3 or TOR – novel changes that occur in both knockdowns but not in controls – 125. Pathway analysis was carried out on both the phosphoproteomics and RNAseq data using the Panther Classification system (Protein Analysis Through Evolutionary Relationships), with the statistical overrepresentation test using Reactome database (version 85)[66]. In addition to the statistical analysis already conducted during data filtering, a Fisher's $t$-test was applied, and only pathways with a $p < 0.05$ for enrichment were considered. For RNA-seq data, model-dependent $p$-values were used, and for all phosphopeptides, $p$-values were determined using the two-sided ANOVA test with Benjamini-Hochberg FDR procedure in Proteome Discoverer.

Temporal patterns of phosphorylation were analyzed using Fuzzy C-Means (FCM) clustering in the $R$ environment (4.5.0). The analysis employed the *cluster* package for the FCM algorithm, *factoextra* for optimization, *pheatmap* for visualization, and *ggplot2* for plotting. Before clustering, only phosphosites exhibiting significant change at least at one time point were selected. To reduce redundancy, duplicate entries mapping to the same phosphosite were consolidated by

averaging log-2 fold-change values. These unique profiles were then analyzed using FCM clustering (function *fanny*). The optimal number of clusters (k = 5) was determined using the Elbow method (package *factoextra*). Temporal profiles were Z-score-standardized before clustering. Heatmaps were generated using the *pheatmap* package, with visualization performed on Z-score standardized values. To visualize the identified temporal patterns, rows were annotated by their fuzzy cluster assignment and ordered using hierarchical clustering. Column order was fixed to maintain the chronological sequence of experimental time points. The heatmap was then rotated so that columns represented phosphosites and rows represented time points. To characterize the biological functions associated with distinct temporal patterns, functional enrichment analysis was performed for each cluster using the *clusterProfiler* and *ReactomePA* packages. Gene symbols were extracted from the phosphosite identifiers and mapped to Entrez IDs using the *org.Dm.eg.db* annotation database. To ensure a gene-centric analysis, duplicate gene entries resulting from multiple phosphosites on the same protein were removed within each cluster prior to statistical testing. Over-representation analysis (ORA) was conducted against the Reactome database. Statistical significance was assessed using a hypergeometric test. To control for multiple testing, *p*-values were adjusted using the Benjamini-Hochberg (BH) procedure. Pathways were considered significantly enriched when the adjusted *p*-value (*q*-value) was less than 0.15.

### Recombinant HR3 protein expression and purification

For baculovirus-mediated expression in insect cell culture, the *HR3* LBD sequence (I230-T487; GenBank accession NP_788303) was subcloned into a pFASTBAC DUAL vector (Invitrogen) that adds a 6xHis tag and a thrombin cleavage site[67]. The hexahistidine-tagged protein was expressed in Hi5 insect cells (*Trichoplusia ni*) grown in 1 liter of M3 medium (Sigma) supplemented with 10% fetal bovine serum (Wisent). Cells were grown at 26 °C for two days in baffled flasks in a shaking incubator. The cells were harvested by centrifugation at 2,000 x *g*, washed with PBS (20 mL), and centrifuged again before the pellets were frozen in liquid nitrogen and stored at -80 °C. Prior to purification, the cell paste was thawed, resuspended in lysis buffer, and sonicated on ice 3 times for 1 minute each. Lysates were clarified by centrifugation at 13,000 x *g*, and tagged protein was bound using Ni-NTA affinity chromatography (column volume 1 mL). Column-bound protein was washed with 300 mL of wash buffer (30 mM imidazole, 500 mM NaCl, 5% glycerol, 0.5 mM TCEP, and 10 mM Tris; pH 8.2). Protein was eluted from the column using a similar buffer containing 250 mM imidazole. The eluate was concentrated to a volume of 5 mL and loaded onto a size-exclusion-chromatography column (Superdex 16/60, GE Healthcare); HR3 LBD was eluted in a buffer of 150 mM NaCl, 0.5 mM TCEP, and 10 mM Tris (pH 8.2) using an AKTA FPLC apparatus (GE Healthcare). After using SDS-PAGE to confirm the purity of eluted fractions, a 10 mg sample of the receptor protein was diluted in buffer $Q_o$ (30 mM NaCl, 5% propanediol, 5 mM DTT, 20 mM Tris, pH 8.5) and loaded onto a Source 30Q anion-exchange column (GE Healthcare). Using a linear NaCl gradient (30 mM to 2-M NaCl in $Q_o$), the protein was eluted using an AKTA FPLC (GE healthcare) and again checked for purity using SDS-PAGE.

### Non-denaturing mass spectrometry

ES-MS was carried out as previously described[49]. Briefly, mass analysis was done using a quadrupole time-of-flight mass spectrometer (Q-TOF, Micromass) equipped with a nano electrospray (ES) source. Purified HR3 LBD was desalted in four dilution/concentration steps using centrifugal concentration (Millipore) into 20-mM ammonium acetate (pH 6.2) and adjusted to a protein concentration of 1 mg/ml. Protein was loaded in a gold-coated capillary (Protona) with the tip opened to produce an orifice of approximately 10 μm. Positive ESI-MS was performed at a capillary voltage of 1.8 to 2 kV and a cone voltage of 10–50 V. Collision-induced dissociation experiments were carried out with a collision energy of 80–150 V using argon as the collision gas. Average molecular masses were calculated using Maxnet1 in Mass Lynx 4.0 (Micromass).

### Extraction of HR3 and GC/MS analysis

Using solvent-washed glassware, endogenous ligand was extracted from 1 mg of purified HR3 LBD by combining 3 mL of purified receptor with 9 mL of acidified chloroform:methanol (2:1 v/v) and mixing vigorously for 1 min. Organic and aqueous phases were separated by centrifugation (13,000 x *g*), and the organic phase was transferred to a clean tube. The aqueous phase was extracted twice more, and the organic fractions were pooled. The acidified chloroform:methanol extract was repeatedly washed with 0.2 volumes of distilled water and reseparated until the aqueous phase became neutral (pH 7). The organic extract was evaporated to dryness under nitrogen, and residue was dissolved in 50 μL pyridine. A 20 μL aliquot of this sample was combined with 10 μL of $N_2O$-bis-(trimethylsilyl)trifluroacetamide ($N_2O$-BSTFA) and trimethylsilyl-derivatized at 60 °C for 30 min. The same procedure was followed for derivatization of the cholesterol reference[68] (Steraloids).

### Gas chromatography mass spectrometry

The derivatized samples were analyzed using a gas chromatograph (3900 GC, Varian) coupled to an ion-trap mass spectrometer (Saturn 2100 T MS/MS, Varian) equipped with an electron-impact ion source. Samples were chromatographically separated using helium as a carrier gas (1.5 mL/min) on a 30 m x 0.25 mm x 0.25 μm fused-silica column (CP8944, Varian) using a two-segment temperature gradient of 80–150 °C at 20 °C/min and 150–345 °C at 10 °C/min. Eluted analyte from chromatography was ionized in the positive-ion mode with a scan range of 40–650 m/z. Data were analyzed using MS Data Review in MS Workstation (Varian).

### Structure prediction and cholesterol placement

The *Drosophila* HR3 ligand-binding domain (LBD) sequence was obtained from UniProt (A1Z858) and trimmed to residues Ile437–Thr694, corresponding to the region most similar to the human RORα LBD. Boltz-2 (refs. [69,70]) was used to predict the HR3 LBD structure in isolation and in complex with cholesterol. Structure prediction was performed using Boltz-2 default settings, except that the number of recycles was increased to 300 and the number of diffusion samples was increased to 100. Cholesterol was provided to Boltz-2 as a ligand in SMILES format, and no constraints were applied and no binding pocket or docking location was specified. A multiple sequence alignment (MSA) for HR3 was generated using the ColabFold MSA server and provided as input for Boltz-2 predictions. For the HR3:cholesterol complex, Boltz-2 generated 100 independent predictions; models were ranked by the Boltz-2 confidence metric, and the top-ranked model was selected for downstream analysis. The predicted HR3 LBD model was aligned to the experimentally determined RORα LBD structure bound to cholesterol (PDB 1N83) in PyMOL, and ligand poses and selected pocket residues were visualized. The relative orientation of the cholesterol 3β-hydroxyl group with respect to conserved pocket residues was inspected in the aligned structures.

### HR3 Ligand-sensor embryo treatment

Transgenic HR3 ligand-sensor (*GAL4_DBD::HR3_LBD; UAS-GFP*) embryos were permeabilized and treated for 15 min at 25 °C with cholesterol in MBIM culture media as previously described in Palanker et al.[34] The MBIM was then removed, and the embryos covered with halocarbon oil and allowed to develop for a minimum of 2 h prior to observation.

### Statistical analysis

Statistical analysis was computed using the Prism software package (GraphPad, version 10). Fat-body pS6-staining data was tested for

outliers using the ROUT method and a Q of 10 due to the large variation within the tissue. All data sets were assessed for normality prior to statistical analysis, and appropriate parametric or non-parametric tests were selected. Graphs were generally created using the Prism software, showing all data points and mean ± standard error of the mean. Detailed statistical information can be found in each figure's legend. Graphical illustrations of RNAseq and phosphoproteomics data were made using SRplot[70].

## Reporting summary

Further information on research design is available in the Nature Portfolio Reporting Summary linked to this article.

## Data availability

All data generated in this study are available within this manuscript, its figures or its supplementary files or on public databases. Source data are provided with this paper. The transcriptomic data generated for this study have been deposited in the NCBI Gene Expression Omnibus (GEO) repository under the accession code GSE270221. The mass-spectrometry proteomics data generated for this work have been deposited to the ProteomeXchange Consortium[71] via the PRIDE[72] repository under the dataset identifier PXD074014. Request for other data, such as raw imagery, should be directed to the Lead Author. Source data are provided with this paper.

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

## Acknowledgements

This work was supported by funding from the Danish Independent Research Council – Natural Sciences (8021-00055B) and the Danish Independent Research Council – Medical Sciences (1030-00144B) to KR. IB was supported in part by a grant from the Austrian Science Fund (FWF, 10.55776/J4944) to I.B. The Zeiss LSM 900 confocal microscope and the PerkinElmer Ensight plate reader were supported by infrastructure grants from the Carlsberg Foundation (CF19-0353 and CF17-0615) to KR et al.

## Author contributions

M.L., M.J.T., K.P., H.K., and K.R. conceived and designed the study. M.L., K.P., I.B., L.H.P., O.K., N.A., S.C., T.K., S.N., S.L., A.K., G.L., A.E., A.A.T., M.R.L., H.M.K., M.J.T., and K.R. designed, performed, and analyzed experiments. M.L., M.J.T., and K.R. wrote the manuscript.

## Competing interests

The authors declare no competing interests.
