## [Peer Review file · Nature Communications]

HR3/ROR α -mediated cholesterol sensing regulates TOR signaling

Corresponding Author: Professor Kim Rewitz

Version 0:

Reviewer comments:

Reviewer #1

(Remarks to the Author)

In the current manuscript, Lassen and others demonstrate that HR3, an evolutionarily conserved nuclear receptor, mediates stimulatory effects of cholesterol on target of rapamycin (TOR) signaling through a non-genomic pathway in the fruit fly *Drosophila melanogaster*. Moreover, they also show that ROR α , a mammalian ortholog of HR3, plays a similar role in human cells, suggesting that this HR3/ROR α -mediated TOR signaling activation is an evolutionarily conserved signaling pathway.

Researchers including the authors' group have recently revealed the stimulatory effects of cholesterol on TOR signaling in different model organisms. There was also an old study reporting a non-genomic, growth-promoting effect of HR3 in *Drosophila*. But nobody has ever connected these dots to integrate them into a nice story. The current study thus fills a critical knowledge gap in the field and is therefore expected to appeal to many researchers in the relevant field. I have several points for the authors to consider as listed below, hoping that they would help them further improve their manuscript.

Major points:

- 1) To exclude the possibility that the steroid hormone ecdysone is mediating the S6K-stimulating effect of cholesterol feeding (Figure 1), the authors performed ecdysone (20E) feeding experiment in Figure S1 and concluded that cholesterol activates the TOR pathway independently of ecdysone (p. 5, lines 185-187). Although the authors' conclusion is most likely correct, this particular experiment is not ideal, as we can never assume that feeding a certain amount of 20E properly reflects the fluctuation of ecdysone titer in the hemolymph upon cholesterol feeding. As the authors have a tool to transiently suppress gene expression in the prothoracic gland (p h mTS \gg), I would recommend the authors to investigate this possibility further by transiently shutting down ecdysone production in the prothoracic gland (by knocking down a Halloween gene, for example), feeding them with cholesterol, and checking the pS6 levels in the fat body.
- 2) Figure 6C lacks one important experimental group, with Tor-RNAi and no HR3(LBD) OE. This group is critical for evaluating if the growth-stimulating effect of HR3(LBD) OE is indeed canceled by Tor-RNAi.
- 3) Current interpretation of the data presented in Figure 6E is vague and confusing. Although the authors mention that "the TOR activity increase due to HR3(LBD) overexpression was largely dependent on both RagA-B and RagC-D" (p.9, lines 423-424), RagA-B-RNAi and RagC-D-RNAi actually showed the opposite results when combined with HR3(LBD) OE. More specifically, RagA-B-RNAi completely suppressed the effects of HR3(LBD) OE, whereas RagC-D-RNAi did not. It therefore seems to me that the TOR-stimulating effect of HR3 is mainly mediated by RagA-B. These results need to be interpreted more carefully.
- 4) Figure 7B and 7C needs to be repeated multiple times and quantified with appropriate statistical analyses.

Minor points:

- 5) p.4, line 131, "ecdysone steroid hormone EcR": I guess the authors meant to say "ecdysone receptor, EcR."
- 6) In Figure 2, panels A-B and G-H are aligned in the order of cholesterol-protein (purple-green), whereas this order is flipped in panels D, E, and F. It makes it easier for readers to interpret the data if they are aligned the same way throughout the entire figure.
- 7) Figure 3F: It seems that the peak at 330 m/z is mislabeled as 300?
- 8) p.8, lines 348-350, "This implies ..., consistent with insights from other recent studies in mammalian systems.": This sentence reads as if ROR α has already been shown to modulate TOR activities in mammals. Please rephrase.
- 9) It has been shown in *Drosophila* that sterols other than cholesterol can be converted into functional ecdysteroids that are equivalent to ecdysone/20E (e.g., Lavrynenko et al., 2015, PMID: 26395481). It is therefore interesting to discuss whether

HR3 is a specific sensor of cholesterol, or it can sense a broader range of sterols that can be converted into functional ecdysteroids.

Reviewer #2

(Remarks to the Author)

The authors describe several exciting and noteworthy results:

They demonstrate that *Drosophila* DH3 binds cholesterol, identifying for the first time a ligand for the well-described receptor. They further show that DH3 mediates the effect of cholesterol on protein synthesis via Tor and phosphorylation of SK6. It also directs an increase in body mass in larvae in a cholesterol and Tor dependent fashion.

DH3 is a well-characterized transcriptional regulator, but here the authors present data showing that its cholesterol dependent role requires only the ligand binding domain (LBD), and that cholesterol binds to the LBD. This finding significantly extends the role of DH3 beyond its well-described transcriptional role in mediating Ecdysone function and demonstrates a cholesterol-dependent non-genomic function of DH3.

The authors further show that the mammalian DH3 homologue ROR α has a similar function in human Karpas 707H cells, demonstrating a conserved function for these proteins in growth control.

The fly system is uniquely suited for these studies since flies do not produce cholesterol and need to take it up through food. Cholesterol levels in both isolated tissues and whole animals can therefore easily be manipulated in experiments. Furthermore, the available GAL4DH3-LBD reporter is well suited to monitor ligand dependent activation.

This work significantly expands previous findings that DH3LBD form mediates Tor function and links it mechanistically to cholesterol.

The conclusions are well supported through genetics, proteomics, and genomic approaches.

Following are a few questions and suggestions:

1. The potential role of Ecdysone in this process is complex and remains unclear. It is well known that Ecdysone induces DH3, which in turn transcriptionally regulates further mediators of Ecdysone action. To what degree Ecdysone may play a role in the newly described role of DH3 is therefore a complex question. The authors show that addition of Ecdysone does not evoke the same response as cholesterol. As the authors point out, one would not expect the non-protein synthesis dependent role of DH3 to be directly Ecdysone receptor dependent. One of the unanswered questions is whether the LBD-only transcript is Ecdysone dependent. Given the complexity of the Ecdysone/DH3 interaction, placement of the results in the supplemental section is a good idea. Perhaps the discussion of these complexities and the reference to the corresponding figures would be well placed in the discussion?

2. The GAL4 DH3-LBD reporter is a nice and well-characterized in developmental context, and this is the first instance where a ligand for this "orphan receptor" has been identified. The authors may want to expand on the discussion that there might be several ligands for the protein. This might include a notion that the described experiments are done in larvae, where growth is a major aspect, in contrast to adults. This leaves open the question whether similar findings might be observed in adult flies (and tissues other than the fat body).

3. Details in the Methods section are well described. While somewhat redundant, adding some relevant methodical details to figure legends would help readers understand some of the complex figures. For example, in figure 5 I-K where HR3-RNAi is conditionally expressed using Gal80ts, addition of the induction details in the figure legend would be helpful (at what age, for how long). A sentence or two in the text explaining the principle of Gal80ts induction would also help readers who don't know the system. Similarly, a short description of the GAL4DH3-LBD sensor would be helpful.

4. For the majority of experiments, the authors do not use inducible expression. This is not an issue for reporter assays (Observation pS6E, induction of growth in response to cholesterol exposure), but it raises interesting questions when RNAi constructs or the LBD-overexpression construct are assayed. While the assays are often done in larvae by changing cholesterol levels, it is noteworthy that RNAi knockdown or LBD overexpression in these cases have been happening all through development. Have the authors observed changes in developmental speed/ overall growth caused by these manipulations? Have they compared these animals with control animals? If no changes were observed, how do the authors interpret this? A limited overall role of the fat body or PG in these processes? How about compensatory changes of the system (which might be relevant for the interpretation of the cholesterol assays).

5. Have the authors studied the effect of overexpression of the entire protein? A comparison to the effect of LBD-only expression on gene expression and phospho-proteins might identify the contribution of genomic effects.

6. What is known about proposed structures of DH3 or ROR α ? Are there similarities to known cholesterol-binding proteins? Are there DH3 mutants that might have impaired cholesterol binding function?

Reviewer #3

(Remarks to the Author)

Lassen et al. investigated the role of HR3 in cholesterol sensing and the cholesterol-induced activation of the TOR pathway in *Drosophila*. The authors first demonstrated the role of cholesterol in activating the TOR pathway. Moreover, they found that cholesterol can directly bind to HR3, and cholesterol-induced activation of the TOR pathway is HR3-dependent but independent of HR3's DNA-binding function. At the end, the authors showed that human ROR α can also activate the TOR pathway in response to exogenous cholesterol. The overall experimental design is reasonable, and the data are clearly presented, highlighting some very interesting phenomena. However, the mechanistic insights need to be further explored. The authors should address the following questions and make the necessary revisions.

HR3 primarily functions as a transcription factor. It is intriguing to explore which genes are regulated by cholesterol binding to HR3. However, the authors claim that cholesterol-induced TOR activation via HR3 is independent of its DNA-binding function. Thus, it is crucial to address the key mechanistic question of how cholesterol-HR3 regulates the TOR pathway.

In Figure 1 F and G, the TOR pathway responds differently to cholesterol in the fat body and PG. In the fat body, TOR is rapidly activated within 0.5 hours and then decreases at 2 hours. In contrast, TOR activation in the PG continues to increase over 2 hours. What causes these differences?

In Figure 1D, the activation of the TOR pathway peaks at 1 hour, whereas in Figures 2A and 2D (right), it peaks at 0.5 hours, and in Figure 3I, it peaks at 2 hours. How can these variations be explained?

Figure 2C shows that each time point has only two replicates ($n=2$), which is insufficient. Although the authors may want to include all samples in a single TMT experiment, this design does not allow for statistical analysis. It is unclear how the statistical p-values were calculated in Figures 2GH. The number of replicates for other omics experiments should also be specified.

Figure 2C shows that the proteomics data were obtained from the flow-through of TiO₂ enrichment, which introduces some variations. Why didn't the authors take a small portion of the sample before TiO₂ enrichment for bulk proteomics? Additionally, it should be clarified whether the quantification of phosphorylation was normalized based on protein abundance.

Figures 2C-H: The authors have designed a complex omics experiment, but the analysis is too superficial, lacking information on temporal changes. Although the experiment included eight time points, the data in Figures F-H represent changes at a single time point. The authors should provide a comprehensive analysis of phosphorylation and proteome changes over time during cholesterol treatment, distinguishing changes between fast and slow responses.

Other omics experiments have similar issues and need revisions. For example, Figures 5D-G show three different omics datasets at three different time points, making it difficult to correlate the data.

Figures 3A and 3B: Please provide deconvoluted MS data. The legend for Figure 3B describes the spectrum as collision-induced dissociation, while the main text describes it as denatured samples, which are two completely different experiments. The authors need to clarify this discrepancy.

The interaction between cholesterol and the HR3 LBD requires more functional validations. For example, does the deletion of LBD abolish cholesterol-induced TOR activation?

Minor Points:

The authors primarily use pS6 as a marker for TOR pathway activation. Are there other markers, such as phosphorylation of TOR itself, that can be used to validate changes in TOR pathway activity?

Reviewer #4

(Remarks to the Author)

The manuscript of Lassen and coauthors investigates the molecular mechanisms that link dietary intake of cholesterol with the growth regulatory TOR pathway. By using *Drosophila melanogaster* as a model system, they first demonstrate that dietary cholesterol feeding induces TOR activation in fat bodies and PG cells as well as in whole animals. They show that the nuclear receptor HR3, the *Drosophila* ortholog of human ROR, is activated by cholesterol binding and modulates TOR activity independently of ecdysone-mediated effects. They further demonstrate that HR3 can mediate Cholesterol-induced TOR activation through an isoform that lacks DNA-binding domain. Reducing HR3 in cells can mitigate the hyperactivation of TOR and intralysosomal accumulation of cholesterol caused by depleting Npc1a. These results indicate that HR3 is required for TOR activation by lysosomal cholesterol. Finally, they use KARPAS-707H cells, which strongly express human ROR, to explore the potential role of human ROR in modulating the TOR-pathway in response to exogenous cholesterol. Collectively their findings suggest that HR3/ ROR represents a conserved mechanism for coupling cholesterol levels to TOR pathway activation.

Overall, data in the manuscript are solid, providing a significant advance in our understanding of how cholesterol activates cell growth via TOR signaling with potential implications in cancer and cholesterol-related pathologies. The work and data analysis are well conducted and support the conclusions and the research methodologies are sound and innovative.

including phosphoproteomic analysis and mass spectrometry, RNA-seq. Thus, the manuscript is suitable for publication in this journal. I only have minor points:

- 1- Figure 7I illustrates the model for the role of HR3 in cholesterol-mediated TOR activation. However, the model seems relatively simple with respect to the large amounts of data reported in the paper. I would suggest to add a figure (figure 8) reporting the main conclusions of the paper.
- 2- A diagram illustrating the TOR pathway in *Drosophila*, the subcellular localization of the TOR pathway proteins (for example RagA and RagB) could facilitate the reader. This diagram could be shown in Figure 1 and cited in the introduction.
- 3- The subcellular localization of HR3 (and human ROR as well) is a key point for the conclusions in the paper. However, HR3 localization is shown in a supplemental figure S4F. I would suggest to incorporate these data into one of the main figures.
- 4- On page 7, lines 270-271: "Knockdown of HR3 in the fat body or PG using an RNAi construct that targets all of the annotated transcript variants of HR3 led to reductions in pS6 staining in these tissues". The authors should clarify how they demonstrated (or it was previously shown), or where they show in the manuscript, that the RNAi construct targets all of the annotated transcript variants of HR3.

Version 1:

Reviewer comments:

Reviewer #1

(Remarks to the Author)

The authors have thoroughly addressed all the concerns I raised in the first round. I do not have any further comments. Congratulations on this beautiful work!

Reviewer #2

(Remarks to the Author)

I am satisfied with the changes the authors have made in response to my comments and beyond. I commend the authors for their careful extended work and for improvements that have made it easier for readers to fully understand figures. The additional experiments the authors performed add significant new insight into the complexity of this receptor and are appreciated.

Reviewer #3

(Remarks to the Author)

I thank the authors for addressing my previous concerns and for the revisions made accordingly. These changes have clearly improved the overall quality, clarity, and rigor of the manuscript.

I have one remaining technical point regarding the workflow illustrated in Figure 2C, which I believe requires clarification. Related to my previous question: "Why didn't the authors take a small portion of the sample before TiO₂ enrichment for bulk proteomics?" It is not clear whether the bulk (global) proteome analysis was performed on the TiO₂ flow-through fraction or on the input sample prior to TiO₂ enrichment.

As currently shown, Figure 2C suggests that the bulk proteome was analyzed from the TiO₂ flow-through. This approach could potentially introduce substantial bias, as the flow-through is depleted of phosphopeptides and may not accurately represent the original proteome composition.

In addition, the workflow indicates HILIC (hydrophilic interaction liquid chromatography). Since HILIC, similar to TiO₂, is also commonly used for phosphopeptide enrichment, it is unclear whether two different enrichment strategies were applied. However, HILIC is not described in the Methods section. I therefore wonder whether this is a labeling error and whether the authors actually intended to indicate high-pH reversed-phase HPLC (high-pH RP-HPLC) instead.

Reviewer #4

(Remarks to the Author)

The manuscript of Lassen and coauthors investigates the molecular mechanisms that link dietary intake of cholesterol with the growth regulatory TOR pathway. By using *Drosophila melanogaster* as a model system, they first demonstrate that dietary cholesterol feeding induces TOR activation in fat bodies and PG cells as well as in whole animals. They show that the nuclear receptor HR3, the *Drosophila* ortholog of human ROR α , is activated by cholesterol binding and modulates TOR activity independently of ecdysone-mediated effects. They further demonstrate that HR3 can mediate cholesterol-induced TOR activation through an isoform that lacks DNA-binding domain. Reducing HR3 in cells can mitigate the hyperactivation of TOR and intralysosomal accumulation of cholesterol caused by depleting Npc1a. These results indicate that HR3 is required for TOR activation by lysosomal cholesterol. Finally, they use KARPAS-707H cells, which strongly express human ROR α , to explore the potential role of human ROR α in modulating the TOR-pathway in response to exogenous cholesterol. Collectively their findings suggest that HR3/ ROR α represents a conserved mechanism for coupling cholesterol levels to

TOR pathway activation.

The revised version of the manuscript has addressed all the concerns that I raised in my previous review and is now suitable for publication in this journal.

Specifically they have added a new panel to figure 7 (Fig. 7i) showing the main conclusions of their study into a single working model based on their genetic and phosphoproteomic data.

As I suggested, because the subcellular localization of HR3 is an important aspect of the paper, they incorporated HR3 immunostaining images into the main figure (Fig. 6h) showing HR3 localization in the larval fat body under cholesterol-deprived conditions and after cholesterol re-feeding.

Finally they included a new panel in Supplementary Figure 6f, providing schematic representations of the HR3-RNAi constructs that were used in the study and showing that each RNAi line targets distinct regions of the HR3 gene.

Response to reviewers' comments

Reviewer #1 (Remarks to the Author):

In the current manuscript, Lassen and others demonstrate that HR3, an evolutionarily conserved nuclear receptor, mediates stimulatory effects of cholesterol on target of rapamycin (TOR) signaling through a non-genomic pathway in the fruit fly *Drosophila melanogaster*. Moreover, they also show that RORalpha, a mammalian ortholog of HR3, plays a similar role in human cells, suggesting that this HR3/RORalpha-mediated TOR signaling activation is an evolutionarily conserved signaling pathway.

Researchers including the authors' group have recently revealed the stimulatory effects of cholesterol on TOR signaling in different model organisms. There was also an old study reporting a non-genomic, growth-promoting effect of HR3 in *Drosophila*. But nobody has ever connected these dots to integrate them into a nice story. The current study thus fills a critical knowledge gap in the field and is therefore expected to appeal to many researchers in the relevant field. I have several points for the authors to consider as listed below, hoping that they would help them further improve their manuscript.

Author response: We thank the reviewer for the positive and thoughtful assessment of our work and for recognizing the conceptual advance it represents. We appreciate the constructive suggestions and have revised the manuscript accordingly, as detailed in our responses to the individual points below.

Major points:

Reviewer point 1) To exclude the possibility that the steroid hormone ecdysone is mediating the S6K-stimulating effect of cholesterol feeding (Figure 1), the authors performed ecdysone (20E) feeding experiment in Figure S1 and concluded that cholesterol activates the TOR pathway independently of ecdysone (p. 5, lines 185-187). Although the authors' conclusion is most likely correct, this particular experiment is not ideal, as we can never assume that feeding a certain amount of 20E properly reflects the fluctuation of ecdysone titer in the hemolymph upon cholesterol feeding. As the authors have a tool to transiently suppress gene expression in the prothoracic gland (*phm^{TS>}*), I would recommend the authors to investigate this possibility further by transiently shutting down ecdysone production in the prothoracic gland (by knocking down a Halloween gene, for example), feeding them with cholesterol, and checking the pS6 levels in the fat body.

Author response: Thank you for the insightful suggestion. We agree that acute 20E feeding is an imperfect proxy for endogenous ecdysone dynamics, and we have solidified that finding along the suggested line. In a new experiment for the revision, we used *phm^{TS>}* to transiently suppress ecdysone production in the prothoracic gland (PG) by knocking down *torso* (the PTTH receptor), *phantom* (*phm*), or *disembodied* (*dib*), which are all individually essential for ecdysone synthesis. Larvae were unperturbed through development until 120 hours after egg-laying (AEL), at which they were placed on low (1.2 µg/mL)-cholesterol medium and gene knockdown was induced by raising the temperature. After ten hours of knockdown and cholesterol depletion, the animals were then re-fed for 6 h with either 1.2 µg/mL (low)- or 80 µg/mL (high)-cholesterol diet. Despite reduced or absent ecdysone synthesis due to PG-targeted gene knockdown, cholesterol refeeding increased fat-body pS6 to control-like levels, indicating that the response is independent of

ecdysone production and consistent with direct TOR activation. These data and quantification are now included in new Supplementary Figure 1f in the revised manuscript, and the Results and Methods have been updated accordingly.

Reviewer point 2) Figure 6C lacks one important experimental group, with Tor-RNAi and no HR3(LBD) OE. This group is critical for evaluating if the growth-stimulating effect of HR3(LBD) OE is indeed canceled by Tor-RNAi.

Author response: We agree that it is appropriate to add *Tor-RNAi* alone and have included a new figure with *Tor-RNAi* without *HR3(LBD)* overexpression. The revised Figure 6c shows that the growth-promoting effect of *HR3(LBD)* overexpression is abolished when *Tor* is knocked down, indicating that TOR activity is required for HR3(LBD)-dependent growth.

Reviewer point 3) Current interpretation of the data presented in Figure 6E is vague and confusing. Although the authors mention that “the TOR activity increase due to HR3(LBD) overexpression was largely dependent on both RagA-B and RagC-D” (p.9, lines 423-424), RagA-B-RNAi and RagC-D-RNAi actually showed the opposite results when combined with HR3(LBD) OE. More specifically, RagA-B-RNAi completely suppressed the effects of HR3(LBD) OE, whereas RagC-D-RNAi did not. It therefore seems to me that the TOR-stimulating effect of HR3 is mainly mediated by RagA-B. These results need to be interpreted more carefully.

Author response: We thank the reviewer for highlighting this and agree that the results indicate the response is mainly mediated by RagA-B. To make that result clear, we have rephrased our original wording in the Results section (previously: “largely dependent on both RagA-B and RagC-D”). The revised manuscript text now states that the increase in TOR activity due to *HR3(LBD)* overexpression was completely abolished by *RagA-B* knockdown, whereas *RagC-D* knockdown only diminished the response, indicating that HR3-induced TOR activation is mainly mediated by RagA-B. We believe this revision improves the interpretation of the results, and we thank the reviewer for raising this point.

Reviewer point 4) Figure 7B and 7C needs to be repeated multiple times and quantified with appropriate statistical analyses.

Author response: We thank the reviewer for this comment. We have now added quantified data based on independent biological samples for the experiments shown in Figure 7b and 7c in a completely new panel, Supplementary Figure 8a-c. The quantification and statistical analysis of normalized pS6 and ROR α levels confirms the data described in Figure 7b and 7c and supports our conclusion that cholesterol activates TOR signaling through ROR α .

Minor points:

Reviewer point 5) p.4, line 131, “ecdysone steroid hormone EcR”: I guess the authors meant to say “ecdysone receptor, EcR.”

Author response: Thank you for catching this. We have corrected “ecdysone steroid hormone EcR” to “ecdysone receptor, EcR” in the revised manuscript.

Reviewer point 6) In Figure 2, panels A-B and G-H are aligned in the order of cholesterol-protein (purple-green), whereas this order is flipped in panels D, E, and F. It makes it easier for readers to interpret the data if they are aligned the same way throughout the entire figure.

Author response: Thank you for the suggestion. We have flipped panels d-f so the order now matches A-B and G-H. This alignment should make the figure easier to interpret.

Reviewer point 7) Figure 3F: It seems that the peak at 330 m/z is mislabeled as 300?

Author response: Thanks for catching this. We have corrected the label to m/z 330 in the revised Figure 3F.

Reviewer point 8) p.8, lines 348-350, “This implies ..., consistent with insights from other recent studies in mammalian systems.”: This sentence reads as if ROR α has already been shown to modulate TOR activities in mammals. Please rephrase.

Author response: Thank you for the suggestion. We have clarified the sentence to avoid implying prior evidence that ROR α modulates TOR in mammals. The revised text now reads: “*This implies HR3 operates at the level of, or upstream of, the LAMTOR (Ragulator) complex and RAPTOR-dependent TOR activation, consistent with insights from other recent studies in mammalian systems indicating that cholesterol activates the TOR pathway via these components.*” This wording specifies pathway components (LAMTOR/Ragulator and RAPTOR-dependent recruitment) without asserting a direct role for ROR α in mammalian TOR regulation.

Reviewer point 9) It has been shown in *Drosophila* that sterols other than cholesterol can be converted into functional ecdysteroids that are equivalent to ecdysone/20E (e.g., Lavrynenko et al., 2015, PMID: 26395481). It is therefore interesting to discuss whether HR3 is a specific sensor of cholesterol, or it can sense a broader range of sterols that can be converted into functional ecdysteroids.

Author response: To test whether sterols other than cholesterol can activate HR3 is an interesting idea. However, this activation test requires the transgenic HR3 ligand-sensor (*HR3::GAL4; UAS-GFP*). Unfortunately, we had found during the revision that our in-house stock of the *HR3::GAL4 (II); UAS-GFP (III)* began showing variability of the mini-white marker (multiple eye colors), which can arise in homozygous transgenic lines due to, e.g., selection against the presence of the inserted transgene. To eliminate any ambiguity and ensure reproducibility, we therefore tried to re-obtain the line again from the Bloomington *Drosophila* Stock Center. Unfortunately, over several months the stock had to be shipped twice, and both shipments arrived nonviable, which delayed our work and prevented us from performing additional experiments to test other sterol ligands.

We agree that it is very plausible that sterols closely related to cholesterol can ligate and activate HR3. Although our current data specifically show cholesterol-dependent activation of HR3, we have now

included an additional paragraph in the Discussion of the revised manuscript that directly addresses whether HR3 might sense sterols beyond cholesterol that can be converted into functional ecdysteroids, and we cite Lavrynenko *et al.*, 2015, as well as an additional study showing that cholesterol derivatives can activate ROR α ¹. This includes discussion that ROR α can be activated by 7-dehydrocholesterol, which is a key intermediate in the ecdysone-biosynthetic pathway, raising the possibility that HR3 could also be activated by 7-dehydrocholesterol that is converted into ecdysone in the prothoracic gland. We believe that while this remains an important question raised by our study, the new discussion appropriately highlights it as an interesting avenue for future work.

The new discussion paragraph is: “*In insects, several dietary sterols besides cholesterol can be converted into functional ecdysteroids equivalent to ecdysone and the more active form, 20-hydroxyecdysone (20E) 51. Our data argue that HR3-mediated TOR activation reflects a direct sterol-sensing route rather than an ecdysone-driven effect, because (i) the cholesterol response is rapid and partly cycloheximide-insensitive, and (ii) loss of ecdysone signaling by EcR knockdown or blockade of ecdysone biosynthesis did not reduce cholesterol-mediated TOR-pathway activation. However, we cannot exclude that HR3 binds a broader set of dietary sterols that can be converted into ecdysteroids. Mammalian receptor ROR α activity can be modulated by multiple sterols, including 7-dehydrocholesterol and cholesterol sulfate 33. 7-dehydrocholesterol is a key sterol intermediate in the ecdysone biosynthetic pathway, raising the possibility that HR3 might also sense this precursor in the PG. Determining the ligand specificity of HR3, including sterols beyond cholesterol, will clarify whether HR3 is cholesterol-specific or a more general sterol sensor, which remains an interesting question for future studies. Because our current experiments were performed in larvae, where growth and biosynthetic activity are high, it will be important to determine whether similar HR3-sterol interactions occur in adults, where metabolic priorities differ. Likewise, whether HR3 responds to sterols in tissues other than the fat body, such as the gut or gonads, remains an open and intriguing question.*”

If the reviewer believes that this matter should be explored further, we propose that we might pursue computational modeling of the binding of diverse sterols in the predicted HR3 binding pocket, along the lines of what we have done for cholesterol in the revised Fig. 3b,c and Supplemental Fig. 4a-c.

Reviewer #2 (Remarks to the Author):

The authors describe several exciting and noteworthy results:

They demonstrate that *Drosophila* DH3 binds cholesterol, identifying for the first time a ligand for the well-described receptor. They further show that DH3 mediates the effect of cholesterol on protein synthesis via Tor and phosphorylation of SK6. It also directs an increase in body mass in larvae in a cholesterol and Tor dependent fashion.

DH3 is a well-characterized transcriptional regulator, but here the authors present data showing that its cholesterol dependent role requires only the ligand binding domain (LBD), and that cholesterol binds to the LBD. This finding significantly extends the role of DH3 beyond its well-described transcriptional role in mediating Ecdysone function and demonstrates a cholesterol-dependent non-genomic function of DH3.

The authors further show that the mammalian DH3 homologue ROR α has a similar function in human Karpas 707H cells, demonstrating a conserved function for these proteins in growth control.

The fly system is uniquely suited for these studies since flies do not produce cholesterol and need to take it up through food. Cholesterol levels in both isolated tissues and whole animals can therefore easily be manipulated in experiments. Furthermore, the available GAL4DH3-LBD reporter is well suited to monitor ligand dependent activation.

This work significantly expands previous findings that DH3LBD form mediates Tor function and links it mechanistically to cholesterol.

The conclusions are well supported through genetics, proteomics, and genomic approaches.

Following are a few questions and suggestions:

Author response: We are grateful for the reviewer's supportive assessment, and we appreciate the recognition of our key findings identifying cholesterol as an HR3 ligand and delineating its non-genomic role in TOR/S6K-mediated growth control, as well as the results indicating that this is a conserved role for ROR α in human cells. The reviewer also makes specific helpful comments, which we address in our responses below.

Reviewer point 1) The potential role of Ecdysone in this process is complex and remains unclear. It is well known that Ecdysone induces DH3, which in turn transcriptionally regulates further mediators of Ecdysone action. To what degree Ecdysone may play a role in the newly described role of DH3 is therefore a complex question. The authors show that addition of Ecdysone does not evoke the same response as cholesterol. As the authors point out, one would not expect the non-protein synthesis dependent role of DH3 to be directly Ecdysone receptor dependent. One of the unanswered questions is whether the LBD-only transcript is Ecdysone dependent. Given the complexity of the Ecdysone/DH3 interaction, placement of the results in the supplemental section is a good idea. Perhaps the discussion of these complexities and the reference to the corresponding figures would be well placed in the discussion?

Author response: We agree that the ecdysone-HR3 relationship is complex, and the reviewer brings up a good point. However, to what extent the "long" transcripts that encode the DBD::LBD protein and the "short" variant that encodes only the LBD-only transcript are regulated independently is difficult to answer. According to 5'-RACE experiments performed by Montagne et al. (2010)² to identify the beginning of the novel short "RS" transcript variant, this mRNA is initiated at a genomic site that is transcribed within one of the exons of the longer transcript forms, and further transcription and splicing mimic those of the longer forms. Translation from these shorter messages makes use of a more distal initiator codon, giving rise to the LBD-only protein. Therefore, there is no sequence that is specific to the short transcript, a circumstance that prevents our measuring its abundance by, for example, qPCR. We therefore addressed the point functionally by adding further experiments showing that cholesterol-mediated TOR activation is independent of ecdysone. In the revised manuscript, we show that cholesterol still drives TOR signaling (pS6) when the ecdysone pathway is impaired by knockdown of the PTTH-receptor gene *torso* in the

prothoracic gland (PG) or by silencing the ecdysone-biosynthetic genes *phantom* and *disembodied* in the PG, each of which is required for ecdysone production (new Supplementary Fig. 1f). These manipulations did not diminish the TOR response to cholesterol. Together with our finding that *Ecdysone Receptor (EcR)* knockdown does not reduce the cholesterol-induced TOR response (Supplementary Fig. 5e) and our evidence that HR3 promotes TOR in part through non-genomic mechanisms (Fig. 6g), these results support the ecdysone-independence of cholesterol:HR3-mediated TOR activation. In the Discussion, we further note that ecdysone induces *HR3* expression, suggesting a temporal interplay in which ecdysone sets *HR3* expression levels, while HR3, in turn, modulates TOR-dependent growth according to cholesterol availability.

Reviewer point 2) The GAL4 DH3-LBD reporter is a nice and well- characterized in developmental context, and this is the first instance where a ligand for this “orphan receptor” has been identified. The authors may want to expand on the discussion that there might be several ligands for the protein. This might include a notion that the described experiments are done in larvae, where growth is a major aspect, in contrast to adults. This leaves open the question whether similar findings might be observed in adult flies (and tissues other than the fat body).

Author response: We appreciate this insightful suggestion and agree that HR3 may respond to multiple sterols depending on developmental stage and tissue context. In the revised manuscript, we have expanded the Discussion to address this point. We now note that while our data indicate a direct sterol-sensing function of HR3, we cannot exclude the possibility that HR3 binds a broader set of dietary sterols that can be converted into ecdysteroids. We further discuss that our experiments were performed in larvae, in which growth is paramount, and that it will be important to determine whether similar HR3-sterol interactions occur in adults or in other tissues such as the discs, gut, or gonads. The new discussion paragraph is: “*In insects, several dietary sterols besides cholesterol can be converted into functional ecdysteroids equivalent to ecdysone and the more active form, 20-hydroxyecdysone (20E) 51. Our data argue that HR3-mediated TOR activation reflects a direct sterol-sensing route rather than an ecdysone-driven effect, because (i) the cholesterol response is rapid and partly cycloheximide-insensitive, and (ii) loss of ecdysone signaling by EcR knockdown or blockade of ecdysone biosynthesis did not reduce cholesterol-mediated TOR-pathway activation. However, we cannot exclude that HR3 binds a broader set of dietary sterols that can be converted into ecdysteroids. Mammalian receptor RORα activity can be modulated by multiple sterols, including 7-dehydrocholesterol and cholesterol sulfate 33. 7-dehydrocholesterol is a key sterol intermediate in the ecdysone biosynthetic pathway, raising the possibility that HR3 might also sense this precursor in the PG. Determining the ligand specificity of HR3, including sterols beyond cholesterol, will clarify whether HR3 is cholesterol-specific or a more general sterol sensor, which remains an interesting question for future studies. Because our current experiments were performed in larvae, where growth and biosynthetic activity are high, it will be important to determine whether similar HR3-sterol interactions occur in adults, where metabolic priorities differ. Likewise, whether HR3 responds to sterols in tissues other than the fat body, such as the gut or gonads, remains an open and intriguing question.*”

Reviewer point 3) Details in the Methods section are well described. While somewhat redundant, adding some relevant methodical details to figure legends would help readers understand some of the complex figures. For example, in figure 5 I-K where HR3-RNAi is conditionally expressed using Gal80ts, addition of the induction details in the figure legend would be helpful (at what age, for how long). A sentence or two

in the text explaining the principle of Gal80ts induction would also help readers who don't know the system. Similarly, a short description of the GAL4DH3-LBD sensor would be helpful.

Author response: We thank the reviewer for this helpful suggestion. In the revised manuscript, we have incorporated all of the requested clarifications directly into the relevant figure legends, added brief explanatory passages in the appropriate places in the Results section, and included three new schematic overview panels that clearly explain the feeding and induction conditions as well as the HR3 ligand-sensor system. These additions improve the accessibility of the material for readers who may be less familiar with *Drosophila* genetic tools and the logic of our experimental setups.

To clarify how *GAL80^{TS}*-mediated temporal control was used in the conditional RNAi experiments, we added a short explanation to the Results text describing the principle of the system - specifically, that *GAL80^{TS}* blocks GAL4 at low temperature and becomes inactive upon shifting larvae to 29 °C, thereby permitting induction of RNAi expression within a defined developmental window. We have also added explicit induction details in the legend and two new schematic overview panels in Supplementary Fig. 1a and Supplementary Fig. 1d, which visually summarize the full feeding and induction paradigms used, including the developmental stage at which animals were shifted to 29 °C, how long they remained at this temperature, and when cholesterol manipulations were initiated. These schematics also anchor the timing and rationale of the *GAL80^{TS}* induction relative to larval age and cholesterol treatments. We refer to these figures again later when *HR3-RNAi* is conditionally expressed, as the reviewer suggested, to clarify the exact conditions relevant to Fig 5i-k.

In addition, to address the reviewer's request regarding the GAL4-HR3-LBD cholesterol sensor, we expanded the legend of Fig. 3 to include a concise description of how the ligand sensor functions, and we added a new panel (Fig. 3j) with an illustration of the ligand-sensor design. This schematic explains that the HR3 ligand-binding domain fused to the GAL4 DNA-binding domain drives *UAS-GFP* expression only when cholesterol binds to HR3, thereby providing a readout of ligand-dependent activation in embryos and larval tissues.

Together, these updates ensure that readers can easily follow the experimental logic, understand the function of the genetic tools, and interpret the timing and conditions used in the key cholesterol- and HR3-dependent assays. We believe that these additions substantially improve the clarity and accessibility of the manuscript.

Reviewer point 4) For the majority of experiments, the authors do not use inducible expression. This is not an issue for reporter assays (Observation pS6E, induction of growth in response to cholesterol exposure), but it raises interesting questions when RNAi constructs or the LBD-overexpression construct are assayed. While the assays are often done in larvae by changing cholesterol levels, it is noteworthy that RNAi knockdown or LBD overexpression in these cases have been happening all through development. Have the authors observed changes in developmental speed/ overall growth caused by these manipulations? Have they compared these animals with control animals? If no changes were observed, how do the authors interpret this? A limited overall role of the fat body or PG in these processes? How about compensatory changes of the system (which might be relevant for the interpretation of the cholesterol assays).

Author response: The reviewer raises the important question regarding potential developmental effects of constitutive RNAi and *HR3(LBD)* overexpression. In the revised manuscript, we have now directly addressed this point by generating new data sets for temporally induced manipulations using *GAL80^{TS}* drivers and by adding new data showing developmental timing for the *HR3(LBD)* overexpression line.

First, to separate effects on growth rate from potential changes in developmental timing, we performed new experiments in which *HR3-RNAi* was induced at the transition to the third instar. Using temperature-sensitive fat-body (*Cg^{TS}>*) and prothoracic-gland (*phm^{TS}>*) drivers, RNAi was activated at the L2-L3 transition (120 h AEL at 18 °C), which avoids perturbing earlier developmental transitions and allows us to examine effects on growth and timing during the third (L3) instar independently of earlier stages. As shown in the new Supplementary Fig. 5c, fat-body-specific *HR3* knockdown using this approach reduced larval size without affecting the timing of pupariation, demonstrating that the decreased mass results from a slower growth rate without impacting developmental progression. Since HR3 affects TOR signaling, this finding is consistent with established roles of fat-body TOR activity in regulating systemic growth rate. In contrast, because TOR signaling in the prothoracic gland directly influences ecdysone production and thereby developmental timing, PG-specific *HR3* knockdown induced at the same stage altered pupariation timing, as shown in a new Supplementary Fig. 5d. This tissue-specific difference supports the conclusion that the fat body and PG contribute differently to systemic growth and timing, and that constitutive manipulations may have distinct developmental consequences depending on the tissue involved.

Second, as suggested by the reviewer, we also examined whether overexpression of the *HR3(LBD)* isoform, which enhances cholesterol-induced TOR activation, affects overall developmental timing. The newly added data (shown in Supplementary Fig. 6a) demonstrate that *HR3(LBD)* overexpression in the fat body, which increases larval body mass, does not alter the timing of pupariation, indicating that this manipulation enhances growth rate without disturbing developmental timing. These results argue against widespread compensatory or developmental buffering effects and instead support a model in which HR3(LBD) primarily affects TOR-mediated growth responses. Together, these new temporally controlled experiments show that constitutive knockdown or overexpression does not generally cause confounding alterations in developmental timing in the fat body, whereas manipulations in the PG behave as expected for a tissue that controls systemic timing through ecdysone. We believe these additions and new experiments address the reviewer's point.

Reviewer point 5) Have the authors studied the effect of overexpression of the entire protein? A comparison to the effect of LBD-only expression on gene expression and phospho-proteins might identify the contribution of genomic effects.

Author response: We thank the reviewer for this insightful suggestion. In response, we generated new transgenic UAS lines expressing either full-length *HR3* or a truncated allele (*HR3^{K243X}*) that produces only the DNA-binding domain due to a premature “stop” mutation. We expressed these constructs in the fat body, mirroring our previous experiments with the “short” HR3(LBD) isoform, which contains only the ligand-binding domain and enhances cholesterol-induced TOR activation.

These new experiments indicate clear functional differences between HR3 isoforms. Whereas overexpression of HR3(LBD), which lacks a DBD, potentiates cholesterol-induced pS6 activation, full-length HR3, including the DBD, strongly *suppresses* this response. The *HR3^{K243X}* DBD-only variant

displays an intermediate phenotype, producing detectable but weaker repression of cholesterol responses than the full-length HR3. This effect is consistent with a model in which the DNA-binding domain mediates transcriptional repression of TOR signaling, with the presence of the ligand-binding domain in the full-length receptor enhancing this feedback-limiting genomic activity.

These findings integrate well with our genome-wide RNA-seq and phosphoproteomic datasets. In animals expressing *HR3* knockdown, numerous genes and phosphosites become aberrantly responsive to cholesterol (Fig. 5d,f,g), indicating that HR3 normally buffers or restrains excessive transcriptional and signaling outputs. The strong repression of cholesterol response observed with full-length *HR3* overexpression is therefore consistent with HR3's acting through its genomic functions to shape and restrict the longer-term TOR response to cholesterol.

Together, these additions indicate that HR3 possesses dual mechanisms of regulatory action. HR3(LBD) mediates rapid, non-genomic activation of TOR, whereas full-length HR3 engages ligand-enhanced transcriptional programs that provide negative feedback and dampen TOR activation. Collectively, the new experiments strengthen the mechanistic distinction between genomic and non-genomic HR3 functions and address the reviewer's request. We are grateful to the reviewer for encouraging us to pursue this line of investigation, which we believe has added an important additional layer of mechanistic insight into how HR3 regulates TOR-pathway activation. We have now added the resulting data as a new panel (Supplementary Fig. 6f) and incorporated the corresponding description into the Results and Discussion sections of the revised manuscript.

Reviewer point 6) What is known about proposed structures of DH3 or ROR α ? Are there similarities to known cholesterol-binding proteins? Are there DH3 mutants that might have impaired cholesterol binding function?

Author response: This is an interesting question. High-resolution crystal structures of the human HR3 ortholog ROR α have been reported, in which cholesterol is bound in the canonical nuclear-receptor ligand-binding pocket, providing a structural framework for cholesterol recognition by this receptor family. In contrast, an experimental structure of *Drosophila* HR3 has not been reported. To address this gap and the question raised by the reviewer, we added a structure-based analysis in the revised manuscript produced with the Boltz-2 package to model the HR3 ligand-binding domain (LBD) structure and to assess cholesterol placement in the pocket. Strikingly, the unsupervised Boltz-2 docking algorithm placed cholesterol within the predicted HR3 LBD pocket in an orientation very similar to that observed for cholesterol in ROR α , and the overall protein fold and key helices aligned well between HR3 and ROR α (new Fig. 3a-c and supplementary Fig. 4a-c). These results support the idea that ROR α and HR3 share a conserved, receptor-type cholesterol-binding architecture, rather than interacting with cholesterol through broadly hydrophobic patches in the manner of membrane-integral proteins such as those reported previously³.

This structural analysis also suggests specific candidate residues that may contribute to cholesterol recognition by HR3. In the predicted HR3 pocket, the cholesterol 3 β -hydroxyl is positioned near two conserved arginine side chains (HR3 R₃₂₈XXR₃₃₁), orthologous with residues R₃₆₇XXR₃₇₀ of ROR α that

have been implicated in stabilizing the 3 β -hydroxyl via polar interactions (Supplementary Fig. 4c). In addition, we now provide sequence-level support for conservation of the ligand-binding pocket. A sequence alignment of *Drosophila* HR3 and human ROR α shows that multiple residues implicated in sterol binding are conserved in HR3, including residues experimentally reported to contact cholesterol in ROR α (within 4 Å) or substituted with similar hydrophobic residues, consistent with conserved hydrophobic packing in the pocket (new Supplementary Fig. 4d). We also note the conservation of residues proposed to support cholesterol binding in ROR α through intrapocket interactions (H484 and Y507), both of which are conserved in HR3 (H451 and Y471: Supplementary Fig. 4d).

We believe these residues are strong candidates for structure-guided functional tests, and targeted substitutions at these positions would be expected to interfere with cholesterol binding and provide a direct way to genetically separate ligand binding from downstream signaling outputs. While we believe generating and validating such binding-pocket mutants is beyond the scope of the current revision, we consider this a clear next step enabled by our combined structural predictions and biochemical evidence for HR3-cholesterol binding.

Reviewer #3 (Remarks to the Author):

Lassen et al. investigated the role of HR3 in cholesterol sensing and the cholesterol-induced activation of the TOR pathway in *Drosophila*. The authors first demonstrated the role of cholesterol in activating the TOR pathway. Moreover, they found that cholesterol can directly bind to HR3, and cholesterol-induced activation of the TOR pathway is HR3-dependent but independent of HR3's DNA-binding function. At the end, the authors showed that human ROR α can also activate the TOR pathway in response to exogenous cholesterol. The overall experimental design is reasonable, and the data are clearly presented, highlighting some very interesting phenomena. However, the mechanistic insights need to be further explored. The authors should address the following questions and make the necessary revisions.

Author response: We thank the reviewer for taking time to provide thoughtful and positive evaluation of our work. We are pleased that the reviewer finds the overall design reasonable, the data clearly presented, and the phenomena highlighted by our study interesting.

We would like to briefly clarify the scope of our project in relation to the reviewer's request for additional mechanistic exploration beyond what is already presented in the manuscript. When the manuscript was originally submitted to *Nature Cell Biology*, we were offered the choice of either performing additional work to pinpoint the exact molecular node at which HR3 intersects with the TOR pathway through, for example, proximity labeling or pull-down assays, or transferring the manuscript to *Nature Communications*, where the scope could remain more focused and would not require fully defining this precise mechanistic intersection. This is without doubt an interesting question, and the additional mechanistic detail obtained will be useful, but addressing the matter would require a substantial body of additional work, better suited for future independent studies that go well beyond the scope of the present manuscript. We believe that our work, which comprises an extensive set of experiments spanning quantitative phosphoproteomic profiling to genetic, biochemical, and functional analyses in both *Drosophila* and human cells, firmly establishes the role of HR3 in the TOR system for the first time and thus establishes the foundation on which that future work can be built.

Thus, we fully appreciate the reviewer's request for deeper mechanistic insight and agree that this is an important point for future studies. Despite the understanding, outlined above, that the transfer to *Nature Communications* would not require us to define the precise mechanistic intersection, we have nonetheless made a substantial effort in the revised manuscript to further narrow down the mechanism and provide more detailed mechanistic understanding. We have added experimental results that characterize in greater detail how HR3 regulates TOR signaling. For this purpose, we generated new transgenic lines to enable tissue-specific overexpression of full-length HR3 (containing both the ligand-binding domain, LBD, and the DNA-binding domain, DBD) as well as a construct containing only the DBD, allowing us to dissect the contribution of the ligand-binding versus DNA-binding functions to TOR activation (new Supplementary Fig. 6f). In addition, we performed further epistasis analysis between HR3 and TOR-pathway components, which indicate that HR3 acts downstream of GATOR1 and upstream of the Rags (RagA/B in particular) in the TOR-activation pathway (new Supplementary Fig. 7d). We believe these additions, as well as other revisions made in response to the points raised by the reviewer, address the reviewer's comments, as detailed in our point-by-point responses below.

Reviewer point 1) HR3 primarily functions as a transcription factor. It is intriguing to explore which genes are regulated by cholesterol binding to HR3. However, the authors claim that cholesterol-induced TOR activation via HR3 is independent of its DNA-binding function. Thus, it is crucial to address the key mechanistic question of how cholesterol-HR3 regulates the TOR pathway.

Author response: The reviewer raises the question about how HR3 mechanistically regulates the TOR pathway. We agree that understanding how cholesterol-bound HR3 regulates TOR is interesting, given the proviso above, and we have provided additional data in the revised manuscript to address this.

To dissect the contribution of HR3's DNA-binding versus non-genomic functions mentioned by the reviewer, we generated new UAS-driven transgenic lines expressing either full-length HR3 or a truncated HR3^{K243X} allele that produces a version of HR3 containing only the DNA-binding domain (DBD) due to a premature stop codon. We expressed these constructs in the fat body to assess their effects on TOR-pathway activity. Viewed together with data from experiments using the HR3(LBD) isoform, which contains only the ligand-binding domain (LBD), the new data (now included as new Supplementary Fig. 6f and described in the Results and Discussion) clearly indicate functional division between the transcriptional effects mediated by HR3 and the non-genomic (protein-interaction) effects mediated by the HR3(LBD) isoform that cannot bind DNA. These data show that whereas overexpression of HR3(LBD) potentiates cholesterol-induced pS6 activation, full-length HR3 strongly suppresses this response. The DBD-only HR3^{K243X} variant produces an intermediate phenotype, causing detectable but weaker repression of cholesterol-induced TOR responses than full-length HR3. Because both the full-length and DBD-only HR3^{K243X} variants maintain DNA-binding capacity, but only full-length HR3 retains an intact LBD, these observations suggest that HR3's DBD-dependent occupancy of target loci generally restrains TOR signaling, and that ligand-bound full-length HR3 can further engage transcriptional programs that dampen cholesterol-stimulated TOR activity. Together these findings support a model in which HR3 has dual variant-specific roles. The LBD-only isoform mediates rapid, non-genomic activation of TOR in response to cholesterol, consistent with our conclusion that the acute cholesterol-HR3-TOR link does not require DNA binding. In contrast, the DBD (and hence full-length HR3) engages genomic programs that restrain TOR signaling over longer timescales. This interpretation is reinforced by our genome-wide RNA-seq and phosphoproteomic analyses, which

show that loss of *HR3* allows numerous genes and phosphosites to become aberrantly responsive to cholesterol (Fig. 5d,f,g), indicating that HR3 normally buffers excessive transcriptional and signaling outputs.

In addition, we added new genetic data that further pinpoint the level of the lysosomal TOR-activation pathway at which cholesterol-bound HR3 acts. In mammals, lysosomal cholesterol has been proposed to activate TOR through multiple routes. One of these passes through a LYCHOS-GATOR module that relieves inhibition of the Rag GTPases³. We tested whether the corresponding pathway operates in flies. Fat-body knockdown of the single, highly conserved *Drosophila* LYCHOS ortholog Anchor using two independent RNAi lines did not reduce cholesterol-stimulated Ribosomal protein S6 phosphorylation, our downstream indicator of TOR activity (Supplementary Fig. 7a,b), even though the knockdown efficiency was up to ~80% (Supp. Fig. 7c), indicating that cholesterol-induced TOR activation in this tissue is LYCHOS/Anchor-independent.

We next asked whether the GATOR1 complex itself is required. GATOR1 acts as a GTPase-activating protein (GAP) for RagA/B and thus has the effect of inhibiting TOR. As expected, knockdown of the GATOR1 component *Iml1* increased basal TOR activity (pS6) by relieving this inhibition. However, cholesterol refeeding produced a similar relative increase in pS6 in control and *Iml1-RNAi* animals, indicating that the cholesterol response does not require GATOR1 (new Supplementary Fig. 7d). Importantly, simultaneous *HR3* knockdown strongly attenuated both the elevated basal TOR activity and the cholesterol-induced TOR activation caused by *Iml1* depletion (new Supplementary Fig. 7d). Thus, HR3 is required for normal cholesterol-induced TOR activation even when GATOR-mediated inhibition is relieved. Together, our findings suggest that, although NPC1-dependent lysosomal cholesterol accumulation activates TOR in both *Drosophila* and mammals, the LYCHOS-GATOR module is dispensable for this process in flies, and HR3 provides a critical cholesterol-responsive input to TOR signaling upstream of the Rag GTPases. Because we observe a similar requirement for the conserved human HR3 ortholog ROR α in cholesterol-induced TOR activation, our data support the idea that an HR3/ROR α -based cholesterol-sensing module represents an evolutionarily conserved route linking sterol availability to TOR signaling. Moreover, unlike transmembrane proteins such as LYCHOS, which may interact with membrane cholesterol through broadly hydrophobic contacts, HR3 and ROR α are receptors with defined ligand-binding pockets in which ligand occupancy can directly modulate receptor activity, providing a mechanistically specific basis for cholesterol sensing.

We have revised the Results and Discussion sections to accommodate these new findings, making this dual genomic/non-genomic regulatory architecture explicit and considering how it explains the apparent DNA-binding independence of acute TOR activation while integrating HR3's canonical function as a transcription factor. In the revised manuscript, we also include and discuss new data supporting the positioning of HR3 within the lysosomal cholesterol-dependent TOR pathway as an essential input that operates independently of LYCHOS-GATOR and acts upstream of the Rag GTPases. We believe these additions provide substantially deeper mechanistic insight into how cholesterol-HR3 signaling regulates the TOR pathway and thus address the reviewer's concern.

Reviewer point 2) In Figure 1 F and G, the TOR pathway responds differently to cholesterol in the fat body

and PG. In the fat body, TOR is rapidly activated within 0.5 hours and then decreases at 2 hours. In contrast, TOR activation in the PG continues to increase over 2 hours. What causes these differences?

Author response: It is indeed interesting that the kinetics of the pS6 response are not identical in the fat body and the prothoracic gland (PG) in Fig. 1f,g (Fig. 1g,h in the revised manuscript). The differing physiological roles of these tissues require distinct baseline signaling states and place different demands on intracellular trafficking systems. These differences may be especially strong with regard to lipid handling, since the PG requires large amounts of cholesterol for cell-autonomous use in ecdysone synthesis but does not otherwise store unusual amounts of lipid, whereas the fat body stores large amounts of lipids, including cholesterol, specifically for use by other tissues. Therefore the intracellular disposition of cholesterol stores, and the appropriate cellular response to the level of these stores, likely differs between tissues – perhaps in the fat tissue, cholesterol is more rapidly shuttled into inert storage compartments inaccessible to the TOR system.

Whatever the tissue-specific nuances may be, in both tissues cholesterol rapidly increases TOR-pathway activity within 0.5 hours, and this activity remains elevated through the 2-hour time point, and in both cases the response is abolished by tissue-specific *TOR* knockdown, indicating that the effect is TOR-dependent. We therefore interpret the modest differences in the time courses as reflecting tissue-specific kinetics rather than fundamentally different mechanisms. We have clarified this point in the revised Results section to avoid overinterpreting small kinetic differences between tissues.

Reviewer point 3) In Figure 1D, the activation of the TOR pathway peaks at 1 hour, whereas in Figures 2A and 2D (right), it peaks at 0.5 hours, and in Figure 3I, it peaks at 2 hours. How can these variations be explained?

Author response: We agree that the apparent timing of peak activation varies across experiments. A clear major factor in this difference is the starting cholesterol condition. In Fig. 1d (Fig. 1e in the revised manuscript) and in the cholesterol-induced phosphoproteomics time course in Fig. 2d,e, larvae were shifted from a fully cholesterol-free diet (0 $\mu\text{g/mL}$) to cholesterol-replete food (80 $\mu\text{g/mL}$), which elicited an early maximal response that then changed dynamically over time. By contrast, when larvae were shifted from low but non-zero cholesterol (1.2 $\mu\text{g/mL}$) to 80 $\mu\text{g/mL}$ (Fig. 1e, and 3i, now Fig. 1f and supplementary Fig. 5a in the revised manuscript), the increase in pS6 was delayed and peaked later. In addition, the different readouts (pS6 immunostaining, *unkempt-Luciferase*, and phosphoproteomics) capture TOR activity with distinct temporal integration and sampling times, which can further influence the apparent timing of the maximum. We have clarified in the revised Results which time courses begin from 0 versus 1.2 $\mu\text{g/mL}$ cholesterol and noted that these differing starting conditions alter the timing of the pS6 response, and we avoid overinterpreting small differences in peak timing across assay modalities and experimental contexts.

Reviewer point 4) Figure 2C shows that each time point has only two replicates ($n=2$), which is insufficient. Although the authors may want to include all samples in a single TMT experiment, this design does not allow for statistical analysis. It is unclear how the statistical p-values were calculated in Figures 2GH. The number of replicates for other omics experiments should also be specified.

Author response: We agree with the reviewer that two biological replicates per time point ($n = 2$) provides limited statistical power, and we have made sure the manuscript to (i) clearly state replicate numbers for all omics experiments and (ii) clarify that p-values were calculated in Fig. 2g,h and throughout the proteomics analyses using the ANOVA test built into the *Proteome Discoverer* software package. For the cholesterol-repletion time course in Fig. 2, we were constrained by the capacity of the TMTpro 16-plex format, which limited the number of biological replicates that could be included while still covering the full set of time points and conditions within a single multiplexed experiment.

Despite this limitation, statistical inference is still possible for TMT-based phosphoproteomics with $n = 2$ by using the ANOVA-based models implemented in *Proteome Discoverer*, which estimate variance across reporter-ion intensities and conditions within the multiplexed sample and test for condition-dependent changes. Phosphorylation changes were considered significant using the predefined criteria of $P < 0.05$ and a minimum fold-change threshold of 30%, as stated in the figure legends and the Methods section. These statistical procedures are standard for multiplexed TMT-based phosphoproteomics and allow inference across conditions despite the limited number of biological replicates per time point. In addition, it is worth noting that standard statistical tests (including ANOVA) are mathematically valid for $n = 2$ (although less well-powered), as also described by GraphPad Prism's guidance on statistics with low n (see text below from GraphPad Prism, source: <https://www.graphpad.com/support/faqid/591/>)

KNOWLEDGEBASE - ARTICLE #591

Statistics with $n=2$

Which statistical calculations are valid when you only have two values in each group?

Is valid to calculate the SD or SEM or CI of two values?

It seems to be common lab folklore that the calculations of SD or SEM are not valid for $n=2$. This folklore is wrong. The equations that calculate the SD, SEM and CI all work just fine when you have only duplicate ($N=2$) data.

Is it valid to compute a t test or ANOVA with only two replicates in each group?

Sure. You get more power with more data. But $n=2$ is enough for the results to be valid. (But of course t tests and ANOVA cannot be done with $n=1$.)

Importantly, our conclusions from Fig. 2 do not rely solely on this $n = 2$ time course. The key biological inferences, namely that dietary cholesterol rapidly activates TOR signaling and drives a distinctive dynamic phosphorylation response, are supported by multiple orthogonal datasets and independent experiments. These include pS6 immunostaining time courses in the fat body (Fig. 1e,f), whole-animal TOR-reporter measurements using the *unkempt-Luciferase* assay (Fig. 2a), and independent genetic and mechanistic experiments demonstrating the dependence of the cholesterol response on HR3 and TOR (Figs. 5-6 and associated Supplementary Figures). In addition, later phosphoproteomic experiments in the manuscript that test HR3 or TOR dependence were performed with increased replication ($n = 3$ independent biological replicates per condition), and these experiments recapitulate the central conclusions from the initial time-

course analysis (Fig. 5f,g and associated Supplementary Tables), further strengthening the evidence base. Finally, genetic experiments (Fig. 5a-c; Fig. 6e,i,j) provide functional validation of the signaling relationships inferred from the omics data.

In the revised manuscript, we have ensured that the n numbers and the statistical analyses are presented clearly and consistently across the Results, Figures, and Methods.

Reviewer point 5) Figure 2C shows that the proteomics data were obtained from the flow-through of TiO₂ enrichment, which introduces some variations. Why didn't the authors take a small portion of the sample before TiO₂ enrichment for bulk proteomics? Additionally, it should be clarified whether the quantification of phosphorylation was normalized based on protein abundance.

Author response: The reviewer is raising a point regarding normalization and the use of the TiO₂ flow-through. In large-scale phosphoproteomics experiments, phosphorylation changes are most commonly quantified without normalization to total protein abundance. There are two main reasons for this. First, after TiO₂ enrichment, the corresponding non-phosphorylated peptide or protein is not consistently detected for all phosphosites in the flow-through. Normalizing phosphosite intensities to total protein abundance would therefore require exclusion of a substantial fraction of regulated phosphosites, reducing coverage and potentially biasing the analysis toward highly abundant proteins. Second, normalization to total protein introduces additional statistical noise, as both the phosphopeptide and the corresponding non-phosphopeptide carry their own variance. Propagating these independent sources of variability can reduce statistical power rather than improve it, which is why many phosphoproteomics studies report phosphorylation changes without protein-level normalization when global protein abundance changes are not the primary focus. For these reasons, we did not routinely normalize phosphosite abundances to total protein levels in the global analyses shown in Fig. 2 and related datasets. Instead, phosphosite quantification was based on TMT reporter ion intensities normalized to total peptide signal, which corrects for sample loading and labeling efficiency and represents standard practice for time-resolved phosphoproteomics.

However, to address the reviewer's concern, we also analyzed the TiO₂ flow-through to detect non-phosphorylated peptides, and we normalized selected phosphosites in key TOR-pathway proteins to the matched total (non-phosphopeptide) abundance when this information was available. Specifically, this protein-level normalization was applied to TOR-related proteins in Fig. 5i-k and Fig. 7h, where corresponding non-phosphorylated peptides were reliably detected. This has now been explicitly clarified in the Methods and Figure legends. We believe this added clarification in the revised manuscript and targeted normalization address the reviewer's concerns.

Reviewer point 6) Figures 2C-H: The authors have designed a complex omics experiment, but the analysis is too superficial, lacking information on temporal changes. Although the experiment included eight time points, the data in Figures F-H represent changes at a single time point. The authors should provide a comprehensive analysis of phosphorylation and proteome changes over time during cholesterol treatment, distinguishing changes between fast and slow responses.

Author response: We have now explicitly addressed this by performing and presenting a time-resolved analysis of the phosphoproteomic data. To capture temporal dynamics rather than single time-point effects,

we applied fuzzy c-means clustering to the phosphorylation profiles measured across all eight time points (15 minutes to 10 hours) following cholesterol or protein refeeding and included these data in new Fig. 2g,h and Supplementary Figs. 2b-d and 3a,b. This approach groups phosphosites based on shared temporal trajectories and allows individual sites to partially belong to multiple clusters, which is well suited for nutrient-responsive signaling programs.

This analysis revealed marked temporal differences in how signaling networks are engaged by the two nutrients. Cholesterol refeeding elicited a highly dynamic and temporally structured phosphorylation program, with sites partitioning into multiple distinct clusters exhibiting different temporal response profiles (Supplementary Fig. 2b-d). In contrast, protein refeeding produced a more sustained and comparatively monotonic phosphorylation response, with fewer pronounced temporal transitions within the clusters (Supplementary Fig. 3a,b). Thus, the dataset captures distinct temporal architectures rather than effects at a single time point. We further performed Reactome enrichment (ORA) for each temporal cluster, demonstrating that clusters are differentially associated with functional modules. In particular, cholesterol-responsive clusters showed differential association with pathways including RNA metabolism and processing, whereas enrichment analysis of protein-responsive clusters also revealed involvement of RNA processing pathways (Supplementary Figs. 2 and 3). Collectively, this temporal clustering analysis shows that cholesterol induces dynamic, phase-specific responses, whereas protein promotes sustained signaling, directly addressing the reviewer's request for a comprehensive time-resolved analysis distinguishing fast and slow responses. We have revised the Results text to emphasize these observations.

Reviewer point 7) Other omics experiments have similar issues and need revisions. For example, Figures 5D-G show three different omics datasets at three different time points, making it difficult to correlate the data.

Author response: We agree that correlating multi-omics datasets measured at a limited number of time points is inherently challenging. To address this, we explored whether a similar temporal clustering approach to that used for the phosphoproteomic dataset in Fig. 2 could be applied to the datasets shown in Fig. 5. Specifically, we performed fuzzy c-means clustering on the Fig. 5 phosphoproteomic data (three time points: 0, 1, and 6 hours). However, with only three time points, the analysis did not produce distinct temporal clusters that yielded robust or interpretable pathway enrichment, and the resulting patterns were not informative beyond what is already captured by the direct comparisons shown in the figure. For this reason, we decided not to include this analysis in the manuscript. For the same reason, we did also not apply temporal clustering to the phosphoproteomic data shown in Fig. 7, which is also limited to three time points.

In contrast, the dataset in Fig. 2 includes eight time points across a 10-hour window, providing sufficient temporal resolution to distinguish fast, delayed, and sustained responses and enabling meaningful clustering and pathway-enrichment analyses. We believe that applying identical temporal analyses to datasets with fundamentally different temporal resolutions would not be informative. Nevertheless, we are of course happy to include clustering analyses for Fig. 5 or Fig. 7 in the manuscript if the reviewer feels this would be valuable.

Reviewer point 8) Figures 3A and 3B: Please provide deconvoluted MS data. The legend for Figure 3B

describes the spectrum as collision-induced dissociation, while the main text describes it as denatured samples, which are two completely different experiments. The authors need to clarify this discrepancy.

Author response: We thank the reviewer for bringing this discrepancy to our attention. We agree that the wording in the original text and figure legend could be interpreted as describing two different MS experiments. We have corrected this for clarity in the revised manuscript.

Figure 3a (3d in the revised manuscript) shows native (non-denaturing) ESI-MS of purified HR3 LBD, revealing apo HR3 and a liganded HR3:cholesterol complex. Figure 3b (3e in the revised manuscript) shows the same native MS preparation subjected to collision-induced dissociation (CID), which promotes loss of the bound ligand and yields the corresponding apo HR3 species. Thus, panel e is not a separate denaturing-MS experiment, but a CID spectrum acquired from the native complex to demonstrate ligand dissociation. We have revised the Results text to reflect this accurately.

Regarding the request for deconvoluted MS data, these data were obtained some time ago, and we no longer have access to the original files or deconvolution outputs beyond the spectra shown in the figure. To address this transparently, we have added a supplementary Fig. 4e reporting the dominant charge state and calculated neutral masses corresponding to the labeled m/z peaks in Fig. 3d,e. These calculations confirm that the mass difference between apo HR3 and the liganded species is consistent with cholesterol.

In addition, to further strengthen the interpretation of cholesterol binding, we have included new structure-based analyses in the revised manuscript. Using the modeling package Boltz-2, we generated a predicted HR3 ligand-binding domain structure and found that cholesterol is placed into the predicted HR3 pocket in a configuration closely resembling that observed in the crystal structure of cholesterol-bound ROR α (new Fig. 3a-c; Supplementary Fig. 4a-c). We have also added a new HR3–ROR α sequence alignment highlighting conservation of residues implicated in ligand recognition, including key arginines near the 3 β -hydroxyl group and multiple pocket residues reported to contact cholesterol in ROR α , supporting conservation of the cholesterol-binding pocket in HR3 (Supplementary Fig. 4d). Together, these additions clarify the MS methodology and provide complementary structural support for cholesterol binding by HR3.

Reviewer point 9) The interaction between cholesterol and the HR3 LBD requires more functional validations. For example, does the deletion of LBD abolish cholesterol-induced TOR activation?

Author response: We agree that validation of the HR3-cholesterol interaction is important. In the revised manuscript, we therefore provide structural, biochemical, and functional evidence supporting that cholesterol acts through the HR3 ligand-binding domain (LBD) to regulate TOR signaling.

First, we added *in silico* structural modeling showing that Boltz-2 places cholesterol into the predicted HR3 LBD pocket without manual constraints, in a pose highly similar to the cholesterol ligand observed in the experimentally solved ROR α LBD structure (Fig. 3a,b). The predicted HR3 LBD also aligns well with ROR α (Fig. 3c), and the cholesterol 3 β -hydroxyl group is positioned near a pair of conserved arginine residues that correspond to the pocket residues implicated in sterol coordination in ROR α (Supplementary Fig. 4a-d).

Second, we previously demonstrated direct cholesterol binding to the purified HR3 LBD by native MS and GC/MS, and we showed cholesterol-dependent activation of an HR3 LBD-based *in vivo* ligand sensor (Fig. 3d-l).

Third, to address the reviewer's requested functional test of LBD necessity, we generated new domain-specific HR3 transgenes that allowed us to directly compare an HR3 variant that retains the DNA-binding domain (DBD) but lacks the LBD versus the variant that contains only the LBD (versus the isoform with both domains). Specifically, we produced a full-length *HR3* construct and an LBD-deleted truncation (*HR3*^{K243X}), which retains the DNA-binding domain (DBD) but lacks the ligand-binding domain (LBD), and we compared these to the HR3(LBD) isoform that lacks the DBD and promotes cholesterol-induced TOR activation (new Supplementary Fig. 6f). Using these constructs, we find that *HR3(LBD)* overexpression enhances cholesterol-induced TOR activation (pS6) (Fig. 6d,e), whereas *HR3*^{K243X} (DBD-only; lacking the LBD) does not promote cholesterol-induced TOR activation and instead blunts the response, similar to full-length HR3, with full-length HR3 producing the strongest attenuation (Supplementary Fig. 6f). These results provide the requested functional validation and support that the LBD is required for cholesterol-dependent TOR activation, while the isolated LBD is sufficient to drive the cholesterol-induced TOR response.

Because full-length HR3 and *HR3*^{K243X} both maintain DNA-binding capacity, but only full-length HR3 retains an intact LBD, these observations suggest that DBD-dependent occupancy of target loci generally restrains TOR signaling, and that ligand-bound full-length HR3 can further engage transcriptional programs that dampen cholesterol-stimulated TOR activity, in contrast to the rapid, potentially non-genomic, LBD-dependent activation mediated by HR3(LBD). This dual behavior aligns with our genome-wide transcriptional and phosphoproteomic analyses (Fig. 5d,f,g), which show that HR3 not only enables a large set of cholesterol-responsive genes and phosphoproteins, but also suppresses inappropriate or excessive cholesterol responses, consistent with a model in which HR3 provides both activating and feedback-limiting functions to maintain balanced TOR-pathway output. We have included text in the Results section of the revised manuscript that explains these new domain-specific transgene data and highlights how they support a model in which the HR3 LBD is the key determinant of cholesterol-dependent TOR activation, while the DBD-dependent functions primarily impose transcriptional restraint and feedback control over TOR output.

Together, these additions provide further mechanistic support that cholesterol engages HR3 via its LBD and that this interaction is functionally important for cholesterol-induced TOR signaling.

Minor Points:

Reviewer point 10) The authors primarily use pS6 as a marker for TOR pathway activation. Are there other markers, such as phosphorylation of TOR itself, that can be used to validate changes in TOR pathway activity?

Author response: We agree that it is best practice to support TOR activity changes with more than one readout. In the manuscript, we already include multiple independent TOR-pathway activity measures beyond pS6: (i) pS6K (a direct TOR target) in whole-larval immunoblotting (Fig. 1c), (ii) the *unkempt-Luciferase* TOR reporter, which provides an orthogonal functional readout of TOR activity *in vivo* (Fig.

2a,b), and (iii) our phosphoproteomics datasets, which capture coordinated TOR-pathway-associated phosphorylation responses, including S6 and upstream TOR-pathway regulatory components (for example RAPTOR and LAMTOR/Ragulator-associated factors; Fig. 5i–k) and, in human cells, additional established TOR-pathway nodes including 4EBP1 and AKT/TSC2 axis components (Fig. 7h).

Regarding phosphorylation of TOR itself, we did not use this as a primary validation readout because TOR phosphorylation sites are not always a reliable proxy for TOR kinase output across contexts, and in *Drosophila* in particular the availability and performance of phospho-TOR antibodies suitable for quantitative tissue assays can be limiting.

Reviewer #4 (Remarks to the Author):

The manuscript of Lassen and coauthors investigates the molecular mechanisms that link dietary intake of cholesterol with the growth regulatory TOR pathway. By using *Drosophila melanogaster* as a model system, they first demonstrate that dietary cholesterol feeding induces TOR activation in fat bodies and PG cells as well as in whole animals. They show that the nuclear receptor HR3, the *Drosophila* ortholog of human ROR α , is activated by cholesterol binding and modulates TOR activity independently of ecdysone-mediated effects. They further demonstrate that HR3 can mediate cholesterol-induced TOR activation through an isoform that lacks DNA-binding domain. Reducing HR3 in cells can mitigate the hyperactivation of TOR and intralysosomal accumulation of cholesterol caused by depleting Npc1a. These results indicate that HR3 is required for TOR activation by lysosomal cholesterol. Finally, they use KARPAS-707H cells, which strongly express human ROR α , to explore the potential role of human ROR α in modulating the TOR-pathway in response to exogenous cholesterol. Collectively their findings suggest that HR3/ ROR α represents a conserved mechanism for coupling cholesterol levels to TOR pathway activation.

Overall, data in the manuscript are solid, providing a significant advance in our understanding of how cholesterol activates cell growth via TOR signaling with potential implications in cancer and cholesterol-related pathologies. The work and data analysis are well conducted and support the conclusions and the research methodologies are sound and innovative including phosphoproteomic analysis and mass spectrometry, RNA-seq. Thus, the manuscript is suitable for publication in this journal. I only have minor points:

Author response: We appreciate the reviewer's careful reading of our manuscript and the constructive feedback. We are grateful for the positive evaluation of the significance, rigor, and methodological innovation of our work. We have revised the manuscript to address all the minor points raised, which are discussed below.

Reviewer point 1) Figure 7I illustrates the model for the role of HR3 in cholesterol-mediated TOR activation. However, the model seems relatively simple with respect to the large amounts of data reported in the paper. I would suggest to add a figure (figure 8) reporting the main conclusions of the paper.

Author response: We agree that the original model schematic in Fig. 7i was too simplified relative to the breadth of data presented in the manuscript. To address this, we have added a new summary figure (Fig. 7i) that consolidates the main conclusions of the study into a single working model. The new figure provides more mechanistic detail on where HR3 intersects with the TOR pathway, based on our genetic and phosphoproteomic data. In particular, the schematic places HR3 upstream of the RagA/B module and incorporates the evidence that cholesterol-induced TOR activation in the fat body is independent of the LYCHOS/Anchor and GATOR modules while still requiring HR3. The figure also emphasizes the dual role of HR3 in TOR regulation: the protein induces a rapid, non-genomic, LBD-dependent activation of TOR signaling, alongside a DBD-dependent genomic program that restrains and shapes the longer-term response.

Reviewer point 2) A diagram illustrating the TOR pathway in *Drosophila*, the subcellular localization of the TOR pathway proteins (for example RagA and RagB) could facilitate the reader. This diagram could be shown in Figure 1 and cited in the introduction.

Author response: We agree that a schematic overview of the TOR pathway and the subcellular localization of key components would facilitate the reader, and we appreciate this helpful suggestion. We have therefore added a new panel (Fig. 1a) as suggested illustrating the TOR pathway, including the localization of the Rag module and associated regulators in the context of lysosomal TOR activation.

Reviewer point 3) The subcellular localization of HR3 (and human ROR α as well) is a key point for the conclusions in the paper. However, HR3 localization is shown in a supplemental figure S4F. I would suggest to incorporate these data into one of the main figures.

Author response: We agree that the subcellular localization of HR3 is an important aspect of our conclusions. To address this point, we have now incorporated representative HR3 immunostaining images into the main figure as suggested by the reviewer (Fig. 6h), showing HR3 localization in the larval fat body under cholesterol-deprived conditions and after cholesterol re-feeding.

Reviewer point 4) On page 7, lines 270-271: “Knockdown of HR3 in the fat body or PG using an RNAi construct that targets all of the annotated transcript variants of HR3 led to reductions in pS6 staining in these tissues”. The authors should clarify how they demonstrated (or it was previously shown), or where they show in the manuscript, that the RNAi construct targets all of the annotated transcript variants of HR3.

Author response: We appreciate the reviewer highlighting this point. In the revised manuscript, we have included new data to clarify how each of the three independent *HR3-RNAi* lines used in this study targets *HR3*. As shown in the new panel in Supplementary Figure 6f, we have provided schematic representations of the *HR3-RNAi* constructs used, indicating that each RNAi line targets distinct regions of the *HR3* gene. This revision clarifies how the RNAi approach targets all annotated *HR3* transcript variants. Thank you for helping us improve the clarity of the manuscript.

References

1. Kallen, J.A. *et al.* X-ray structure of the hROR α LBD at 1.63 Å: structural and functional data that cholesterol or a cholesterol derivative is the natural ligand of ROR α . *Structure* **10**, 1697–1707 (2002).

2. Montagne, J. *et al.* The nuclear receptor DHR3 modulates dS6 kinase-dependent growth in *Drosophila*. *PLoS Genet* **6**, e1000937 (2010).
3. Shin, H.R. *et al.* Lysosomal GPCR-like protein LYCHOS signals cholesterol sufficiency to mTORC1. *Science* **377**, 1290–1298 (2022).

February 10, 2026.

Reviewer #1

Reviewer: The authors have thoroughly addressed all the concerns I raised in the first round. I do not have any further comments. Congratulations on this beautiful work!

Author response: We thank the reviewer for his or her very kind words and comments that helped us improve our work.

Reviewer #2

Reviewer: I am satisfied with the changes the authors have made in response to my comments and beyond. I commend the authors for their careful extended work and for improvements that have made it easier for readers to fully understand figures. The additional experiments the authors performed add significant new insight into the complexity of this receptor and are appreciated.

Author response: We are happy to hear that our revisions have improved our manuscript to meet the reviewer's expectations, and we thank the reviewer for his or her kind words. The reviewer's suggestions have certainly helped us to improve our work, and we have enjoyed working together with the reviewer along the way to our finished product. Thanks!

Reviewer #3

Reviewer: I thank the authors for addressing my previous concerns and for the revisions made accordingly. These changes have clearly improved the overall quality, clarity, and rigor of the manuscript.

Author response: We are glad to hear that our revised manuscript meets the reviewer's expectations, and we appreciate his or her kind words. We thank the reviewer for working with us to improve our manuscript.

Reviewer; I have one remaining technical point regarding the workflow illustrated in Figure 2C, which I believe requires clarification. Related to my previous question: "Why didn't the authors take a small portion of the sample before TiO₂ enrichment for bulk proteomics?" It is not clear whether the bulk (global) proteome analysis was performed on the TiO₂ flow-through fraction or on the input sample prior to TiO₂ enrichment. As currently shown, Figure 2C suggests that the bulk proteome was analyzed from the TiO₂ flow-through. This approach could potentially introduce substantial bias, as the flow-through is depleted of phosphopeptides and may not accurately

represent the original proteome composition. In addition, the workflow indicates HILIC (hydrophilic interaction liquid chromatography). Since HILIC, similar to TiO₂, is also commonly used for phosphopeptide enrichment, it is unclear whether two different enrichment strategies were applied. However, HILIC is not described in the Methods section. I therefore wonder whether this is a labeling error and whether the authors actually intended to indicate high-pH reversed-phase HPLC (high-pH RP-HPLC) instead.

Author response: We thank the reviewer for raising this important technical point regarding the workflow illustrated in Figure 2C. In this study, global proteome analysis was not performed, as our experimental design and downstream analyses were focused exclusively on phosphoproteomic signaling changes. Therefore, no bulk proteomics dataset derived from either the TiO₂ flow-through or the pre-enrichment input sample was generated or used in this manuscript.

We also appreciate the reviewer's astute observation regarding the fractionation step. The previous reference to HILIC was made in error, and we have corrected the figure and its accompanying description to accurately indicate that high-pH reverse-phase HPLC, rather than HILIC, was used for fractionation (see updated figure). We hope these revisions address the reviewer's concerns satisfactorily.

Reviewer #4

Reviewer: The revised version of the manuscript has addressed all the concerns that I raised in my previous review and is now suitable for publication in this journal.

Author response: We are happy to hear that our revised work meets all the reviewer's expectations. We truly appreciate his or her taking the time to help us improve the strength of our work, and we feel that each reviewer's input has indeed helped us to create a more solid contribution.

Notes to the Editors

In this final stage of the revision process, we have made the changes detailed above in our reply to Reviewer #3's comments, and we have also made several small changes to our manuscript to respond to points raised in the Editorial Checklist. These amendments include shortening over-long figure legends and adding missing details concerning statistical tests. We have also corrected a small number of typographical errors that we noticed during the process. Changed text in the manuscript is set in blue. In addition to the meaningful corrections made to Fig. 2c in response to the reviewer's comment and the addition of a scale bar to Fig. 3k as requested in the Checklist, we have also made minor purely cosmetic changes to figures to improve their clarity (standardizing capitalization and italics, clarifying axis labels, adding descriptions of color scales). None of these changes affects the data or their interpretation in any way.

We are excited to submit our final manuscript for publication in your fine journal. Once again, thank you for all your assistance, and if you have any further questions or comments, please do not hesitate to contact me.

Sincerely,

Kim Rewitz